# Spike-timing-dependent Hebbian learning as noisy gradient descent

**Niklas Dexheimer**[1]* **Sascha Gaudlitz**[2]* **Johannes Schmidt-Hieber**[1]

[1]University of Twente  [2]Humboldt-Universität zu Berlin

{n.dexheimer, a.j.schmidt-hieber}@utwente.nl  sascha.gaudlitz@hu-berlin.de

## Abstract

Hebbian learning is a key principle underlying learning in biological neural networks. We relate a Hebbian spike-timing-dependent plasticity rule to noisy gradient descent with respect to a non-convex loss function on the probability simplex. Despite the constant injection of noise and the non-convexity of the underlying optimization problem, one can rigorously prove that the considered Hebbian learning dynamic identifies the presynaptic neuron with the highest activity and that the convergence is exponentially fast in the number of iterations. This is non-standard and surprising as typically noisy gradient descent with fixed noise level only converges to a stationary regime where the noise causes the dynamic to fluctuate around a minimiser.

## 1 Introduction

Hebbian learning is a fundamental concept in computational neuroscience, dating back to Hebb [16]. In this work, we provide a rigorous analysis of a Hebbian *spike-timing-dependent plasticity (STDP)* rule. Those are learning rules for the synaptic strength parameters that only depend on the spike times of the involved neurons. More precisely, we consider a neural network composed of $d$ presynaptic/input neurons, which are connected to one postsynaptic/output neuron. The presynaptic neurons communicate with the postsynaptic neuron by sending spike sequences, the so-called spike-trains. Reweighted by synaptic strength parameters $w_1, \ldots, w_d \geq 0$, they contribute to the postsynaptic membrane potential. Whenever the postsynaptic membrane potential exceeds a threshold, the postsynaptic neuron emits a spike, and the membrane potential is reset to zero. Experiments have shown the following stylised facts, which lie at the core of Hebbian learning based on spikes: (1) *Locality:* The change of the synaptic weight $w_i$ depends only on the spike-train of neuron $i$ and the postsynaptic spike-train. (2) *Spike-timing:* The change of the synaptic weight $w_i$ depends on the relative timing of presynaptic spikes of neuron $i$ and

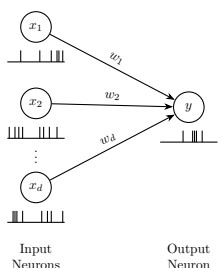

Figure 1: Neural network with a single output neuron

of the postsynaptic neuron. More precisely, a pre-post spike sequence tends to increase $w_i$, whereas a post-pre sequence tends to decrease $w_i$. We refer to Morrison et al. [28] for a more comprehensive list of experimental results on STDP rules.

Hebbian learning rules are well-studied if instead of the precise timings of pre- and postsynaptic spikes, only the mean firing rates are taken into account. These *rate-based* models exhibit many desirable properties, including performing streaming PCA [23, 19] and receptive field development [13, Section 11.1.4]. Much less is known if the precise timing of pre- and postsynaptic spikes is considered, since the intrinsic randomness of the dynamics complicates the mathematical analysis.

---

*These authors contributed equally to this work.

39th Conference on Neural Information Processing Systems (NeurIPS 2025).

Our main contribution lies in connecting STDP to noisy gradient descent and providing a rigorous convergence analysis of the noisy learning scheme. To this end, we introduce a learning rule for the weights $w_1, \ldots, w_d$, which captures the locality and spike-time dependence of Hebbian STDP. We rewrite the learning rule as a noisy gradient descent scheme with respect to a suitable loss function. The connection to noisy gradient descent and stochastic approximation [24, 34] paves the way for applying mathematical tools from stochastic process theory to analyse the STDP rule. Our analysis of STDP is inspired by the work on noisy gradient descent for non-convex loss functions of Mertikopoulos et al. [27]. By refining their arguments and carefully tracking the error terms, we show an exponentially fast alignment of the output neuron with the input neuron of the highest mean firing rate on an event of high probability. The specialisation of the output neuron to the input neuron of the highest intensity is related to the winner-take-all mechanism in decision making [12, 49, 31, 25, 40]. The competitive nature of Hebbian STDP has been observed by [39, 38, 15] and the specialisation to few input neurons is important for receptive field development [9]. By connecting Hebbian STDP to noisy gradient descent, we are able to provide a mathematical analysis beyond ensemble averages and to quantify the speed of convergence.

Taking into account the intrinsic geometry of the probability simplex, we also relate our learning rule to noisy mirror descent, more precisely to noisy entropic gradient descent, which has been proposed for brain-like learning by Neumann et al. [30], Cornford et al. [10].

The key contributions are:

1. **STDP as noisy gradient descent.** We deduce a new framework, in which Hebbian STDP is interpreted as noisy gradient descent. This connection allows us to employ powerful tools from the theory of stochastic processes for analysing Hebbian STDP.

2. **Linear convergence.** We prove the alignment of the output neuron with the input neuron of highest intensity at exponential rate on an event of high probability.

3. **Connection to noisy mirror descent.** We relate our learning rule to noisy mirror descent, more specifically to entropic gradient descent. This connection facilitates the integration of techniques from both areas, potentially leading to future synergistic effects.

**Related literature**

Common approaches to understanding STDP restrict to the mean behaviour after taking the ensemble average, e.g. [13, 22, 15], or compute the full distribution using the master equation of the Markov process [13, Section 11.2.4]. Unfortunately, the latter is only feasible in specific scenarios. In [21], the authors consider a general noisy spike-time dependent dynamic which is transformed into a deterministic ODE by imposing a slow learning rate and using the self-averaging effect of the system. A stability analysis reveals structure formation and output stabilisation. One major difference to our work is the influence of the noise. In [21], the variance of the weights grows linearly and a careful comparison of time scales is required. In our work, despite a constant injection of noise into the system, the dynamic for the spike-triggering probabilities converges to a deterministic limit. Secondly, the use of recent ideas from the analysis of noisy SGD allows us to track the influence of the realised noise in every step. A considerable number of previous works derived STDP rules based on the minimisation of a loss function, typically corresponding to the minimization/maximization of some notion of energy or information, see [8, 6, 7, 44, 42, 32, 43, 36]. While this approach is appealing, the mathematical analysis of these learning rules is challenging due to the modifications required to achieve biological plausibility. In contrast, we start with a biologically plausible learning rule and utilise the arising loss function to derive mathematical convergence guarantees of the learning rule. The importance of the choice of a suitable metric for the derivation of the learning rule is laid out in [41]. We refer to [1, 48, 37, 14, 45] and the references therein for further results on STDP. In [20], a related learning rule is mathematically analyzed by relating it to expert aggregation.

## 1.1 Notation

**Linear algebra.** For a positive integer $d$, we write $[d] := \{1, \ldots, d\}$ and $\mathbf{1} := (1, \ldots, 1)^\top \in \mathbb{R}^d$. For $i \in [d]$ we denote by $\mathbf{e}_i$ the $i^{\text{th}}$ standard basis vector of $\mathbb{R}^d$. The Hadamard product between two vectors $\mathbf{a}, \mathbf{b} \in \mathbb{R}^d$ is denoted by $\mathbf{a} \odot \mathbf{b} := (a_1 b_1, \ldots, a_d b_d)^\top \in \mathbb{R}^d$. We write $\mathbb{I} \in \mathbb{R}^{d \times d}$ for the identity matrix on $\mathbb{R}^d$ and $\|\mathbf{u}\|^2 = \sum_{i=1}^d u_i^2$ for the squared Euclidean norm of a vector $\mathbf{u} \in \mathbb{R}^d$.

**Probability.** $M(1, \mathbf{p})$ denotes the multinomial distribution with one trial ($n = 1$) and probability vector $\mathbf{p} = (p_1, \ldots, p_d)^\top$, that is $\xi \sim M(1, \mathbf{p})$ if only only if $\mathbb{P}(\xi = i) = p_i$ for any $i \in [d]$. We denote by

$$\mathfrak{P} := \left\{ \mathbf{p} \in \mathbb{R}^d : \ p_i \geq 0 \, \forall i \in [d], \sum_{i=1}^{d} p_i = 1 \right\}$$

the probability simplex in $\mathbb{R}^d$. We denote by $\mathbb{1}_{\mathcal{A}}$ the indicator function of a set $\mathcal{A}$.

## 1.2 Hebbian inspired learning rule

Inspired by Hebbian learning, we consider an unsupervised learning dynamic with $d$ input (or presynaptic) neurons and one output (or postsynaptic) neuron. The $i^{\text{th}}$ input neuron has a *mean firing rate* $\lambda_i > 0$ describing the expected number of spikes per time unit. The vector $\boldsymbol{\lambda} = (\lambda_1, \ldots, \lambda_d)$ contains the $d$ mean firing rates. The strength of the connection between the $i^{\text{th}}$ input neuron and the output neuron is modulated by the weight parameter $w_i \geq 0$, and changes to encode the information of the input firing rates.

We introduce a Hebbian STDP rule in Subsection 2.3 and show that, under some assumptions on the spike-trains, it is equivalent to the following dynamics. If $\mathbf{w}(0) = (w_1(0), \ldots, w_d(0))^\top$ are the $d$ weights at initialisation, the updating rule from $\mathbf{w}(k)$ to $\mathbf{w}(k+1)$ is given by

$$\mathbf{w}(k+1) = \mathbf{w}(k) \odot \left( \mathbf{1} + \alpha \left( \mathbf{B}(k) + \mathbf{Z}(k) \right) \right), \tag{1}$$

where $\alpha > 0$ is the learning rate and $k = 0, 1, \ldots$ denotes the postsynaptic spike time. The $d$-dimensional vector $\mathbf{B}(k)$ is the standard basis vector pointing to the presynaptic neuron, which triggered the $(k+1)^{\text{st}}$ postsynaptic spike. It is given as $\mathbf{B}(k) = \sum_{i=1}^{d} \mathbb{1}_{\zeta_k = i} \mathbf{e}_i$, the one-hot encoding of independent multinomial random variables $\zeta_k \sim M(1, \mathbf{p}(k))$, with $k$-dependent probability vector

$$\mathbf{p}(k) = \frac{\boldsymbol{\lambda} \odot \mathbf{w}(k)}{\boldsymbol{\lambda}^\top \mathbf{w}(k)} \in \mathbb{R}^d, \quad k = 0, 1, \ldots \tag{2}$$

Since the probabilities $p_i(k)$ model the probability that the $(k+1)^{\text{st}}$ postsynaptic spike is triggered by neuron $i = 1, \ldots, d$, we call them *(postsynaptic) spike-triggering-probabilities*. The i.i.d. $d$-dimensional vectors $\mathbf{Z}(k)$, $k = 0, 1, \ldots$ model the contribution of presynaptic spikes, which did not trigger the $(k+1)^{\text{st}}$ postsynaptic spike, to the weight change. They are modelled to have i.i.d. components $Z_1(k), \ldots, Z_d(k)$, which are supported in $[-(Q-1), (Q-1)]$, for some $Q > 1$, and centred such that $\mathbb{E}[\mathbf{Z}(k)] = 0$.

In the remainder of the paper, we analyse the long-run behaviour of $\mathbf{p}(k)$ as $k \to \infty$ under the learning rule Eq. (3). We say that the output neuron aligns with the $j^{\text{th}}$ input neuron if $p_j(k) \to 1$ as $k \to \infty$. Since the input intensities $\lambda_1, \ldots, \lambda_d > 0$ are fixed throughout the dynamic, this condition is equivalent to $w_j(k) / \sum_{i=1}^{d} w_i(k) \to 1$ as $k \to \infty$. Figure 1 visualises the learning rule Eq. (2).

## 2 Representation as noisy gradient descent

We continue by relating the learning rule Eq. (1) to noisy gradient descent. For notational simplicity, define $\mathbf{Y}(k) := \mathbf{B}(k) + \mathbf{Z}(k)$ for $k = 0, 1, \ldots$. Combining the weight updates Eq. (1) with the formula for the probabilities $\mathbf{p}$ from Eq. (2), we find

$$\mathbf{p}(k+1) = \frac{\boldsymbol{\lambda} \odot (\mathbf{w}(k) \odot (\mathbf{1} + \alpha \mathbf{Y}(k)))}{\boldsymbol{\lambda}^\top (\mathbf{w}(k) \odot (\mathbf{1} + \alpha \mathbf{Y}(k)))} = \frac{\mathbf{p}(k) \odot (\mathbf{1} + \alpha \mathbf{Y}(k))}{\mathbf{p}(k)^\top (\mathbf{1} + \alpha \mathbf{Y}(k))}, \quad k = 0, 1, \ldots. \tag{3}$$

The normalisation in the denominator and the multiplicative nature of the update ensures that the dynamic of $\mathbf{p}(k)$ is restricted to the probability simplex. By a Taylor expansion around $\alpha = 0$, we find

$$\mathbf{p}(k+1) = \mathbf{p}(k) \odot \left( \mathbf{1} + \alpha \left( \mathbf{Y}(k) - \mathbf{p}(k)^\top \mathbf{Y}(k) \mathbf{1} \right) \right) + \mathcal{O}(\alpha^2). \tag{4}$$

Since

$$\mathbb{E}\left[ \mathbf{Y}(k) - \mathbf{p}(k)^\top \mathbf{Y}(k) \mathbf{1} \mid \mathbf{p}(k) \right] = \mathbf{p}(k) - \|\mathbf{p}(k)\|^2 \mathbf{1},$$

the random vectors

$$\boldsymbol{\xi}(k) := \mathbf{p}(k) \odot \left( \mathbf{Y}(k) - \mathbf{p}(k)^\top \mathbf{Y}(k) \mathbf{1} - \mathbf{p}(k) + \|\mathbf{p}(k)\|^2 \mathbf{1} \right), \quad k = 0, 1, \ldots.$$

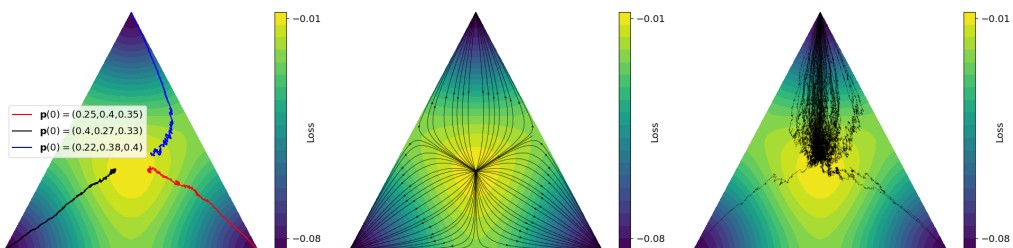

Figure 2: Contour plot of the loss function $L$ from Eq. (6) on the probability simplex $\mathfrak{P}$ for $d = 3$ with different overlays. Left: Three sample trajectories of Eq. (3) with different initial configurations $\mathbf{p}(0)$. Middle: Stream plot of the gradient field given by Eq. (7). Right: 100 sample trajectories of Eq. (3) with $\mathbf{p}(0) = (0.3, 0.3, 0.4)^\top$. All trajectories are simulated with 2000 iteration steps, learning rate $\alpha = 0.01$ and $\mathbf{Z}(k) \sim \text{Unif}([-1, 1]^d)$.

are centred. The distribution of $\boldsymbol{\xi}(k)$ depends on $\mathbf{w}(k)$ and $\mathbf{p}(k)$. Up to $\mathcal{O}(\alpha^2)$-terms, we can write the learning rule Eq. (3) as a noisy gradient descent scheme

$$\begin{aligned} \mathbf{p}(k+1) &= \mathbf{p}(k) \odot (\mathbf{1} + \alpha(\mathbf{p}(k) - \|\mathbf{p}(k)\|^2 \mathbf{1})) + \alpha \boldsymbol{\xi}(k) \\ &= \mathbf{p}(k) - \alpha \nabla L(\mathbf{p}(k)) + \alpha \boldsymbol{\xi}(k), \quad k = 0, 1, \dots \end{aligned} \tag{5}$$

for the loss function

$$L(\mathbf{p}) := -\frac{1}{3} \sum_{i=1}^d p_i^3 + \frac{1}{4} \left( \sum_{i=1}^d p_i^2 \right)^2 = -\frac{1}{3} \mathbf{p}^\top (\mathbf{p} \odot \mathbf{p}) + \frac{1}{4} \|\mathbf{p}\|^4, \quad \mathbf{p} \in \mathbb{R}^d \tag{6}$$

with gradient

$$\nabla L(\mathbf{p}) = -\mathbf{p} \odot (\mathbf{p} - \|\mathbf{p}\|^2 \mathbf{1}) \in \mathbb{R}^d, \quad \mathbf{p} \in \mathbb{R}^d. \tag{7}$$

Dropping $\mathcal{O}(\alpha^2)$ terms is only done for illustrative purposes. Our main result (Theorem 2.2) applies to the original learning rule Eq. (3). The subsequent lemma summarises the key properties of the loss function $L$ from Eq. (6). For $d = 3$, Figure 2 visualises the loss function $L$ and the learning dynamics Eq. (3).

**Lemma 2.1.** *All critical points of the loss function Eq. (6) can be written as $\mathbf{p}^* = \frac{1}{|S|} \sum_{j \in S} \mathbf{e}_j$ for some $S \subseteq [d]$. Every critical point with $|S| \geq 2$ is a saddle point. The local minima of the loss function $L$ from Eq. (6) are the standard basis vectors $\{\mathbf{e}_1, \dots, \mathbf{e}_d\}$. Furthermore, every local minimum of $L$ is also a global minimum.*

## 2.1 Linear convergence of the learning rule

We state the convergence guarantee for the learning rule Eq. (3). Renaming the indices, we can assume that $p_1(0)$ is the largest initial probability. Provided that $p_1(0)$ is strictly larger than each other component of $\mathbf{p}(0)$, the following theorem shows linear convergence of the first component to 1 on an event $\Theta$ in expectation. The probability of $\Theta$ can be chosen arbitrarily close to 1 by reducing the learning rate $\alpha$.

**Theorem 2.2.** *Given $\varepsilon \in (0, 1)$, assume*

$$\Delta := p_1(0) - \max_{i=2,\dots,d} p_i(0) > 0 \quad \text{and} \quad 0 < \alpha \leq \frac{\Delta^2}{16Q^2} \left( (1 - Q\alpha)^3 \wedge \frac{1}{256(1 - p_1(0))} \left( 4\frac{\Delta}{d} + \Delta^2 \right) \varepsilon \right).$$

*Then there exists an event $\Theta$ with probability $\geq 1 - \varepsilon/2$ such that*

$$\mathbb{E}\left[ \|\mathbf{p}(k) - \mathbf{e}_1\|_1 \mathbb{1}_\Theta \right] \leq 2(1 - p_1(0)) \exp\left( -\frac{\alpha}{16} \left( 4\frac{\Delta}{d} + \Delta^2 \right) k \right), \quad \text{for all } k = 0, 1, \dots$$

*Consequently, given $\delta > 0$,*

$$\mathbb{P}\left( \|\mathbf{p}(k) - \mathbf{e}_1\|_1 \geq \delta \right) \leq \varepsilon \quad \text{for all} \quad k \geq \frac{16d}{\alpha \Delta(4 + d\Delta)} \log\left( \frac{4(1 - p_1(0))}{\varepsilon \delta} \right).$$

*Remark* 2.3. If all weights are equal at the starting point of the learning algorithm, the assumption $p_1(0) - \max_{i=2,\dots,d} p_i(0) > 0$, is equivalent to requiring $\lambda_1(0) - \max_{i=2,\dots,d} \lambda_i(0) > 0$. In this case, the convergence of $\mathbf{p}(k)$ to $\mathbf{e}_1$ corresponds to the network performing a winner-take-all mechanism [12, 49, 31, 40, 25]. The competitive selection of input neurons in Hebbian STDP has been observed by [39, 38, 15], among others. Our results extend existing findings by going beyond ensemble averages and also provide a rate of convergence for the random dynamics.

The convergence on an event of high probability is in line with other recent results on noisy/stochastic gradient descent for non-convex loss functions, see e.g. [27, 46] or [11, Theorem 2.5]. Contrary to these results, we can choose a constant learning rate and obtain linear convergence. To illustrate the reason for this, we give a brief overview of the proof of Theorem 2.2. The full proof can be found in Section A.1.

1. We restrict the analysis to the event $\Theta$ on which

$$p_1(k) - \max_{i=2,\dots,d} p_i(k) \geq c > 0,$$

   holds for all iterates $k$. On this event, the derivative of the first component can be bounded from below by

$$p_1(k)(p_1(k) - \|\mathbf{p}(k)\|^2) \gtrsim 1 - p_1(k). \tag{8}$$

2. As described in Eq. (3), we apply a Taylor approximation to the original dynamics. We bound the error term for the $i^{\text{th}}$ component $p_i(k+1)$ by the order $\alpha^2 p_i(k)(1 - p_i(k))$. The approximation error is dominated by the gradient update on $\Theta$, if the learning rate is small enough (see Eq. (8)).

3. Similarly as in Eq. (5), we restate the learning increments of the dynamics as the sum of the true gradient and a centred noise vector $\boldsymbol{\xi}(k)$. By Eq. (8), this decomposition yields linear convergence of $p_1(k) \to 1$ on $\Theta$.

4. To find a lower bound for the probability of the chosen event $\Theta$, we employ a similar strategy as Mertikopoulos et al. [27]. Through the representation Eq. (5), we can show that $\Theta$ occurs, as soon as $\mathbf{M}(k) := \alpha \sum_{i=1}^{k} \boldsymbol{\xi}(i)$ is uniformly bounded by some sufficiently small constant. As $(\mathbf{M}(k))_{k \in \mathbb{N}}$ is a martingale, the probability of the latter event can be controlled through Doob's submartingale inequality (see Eq. (27)).

5. To apply Doob's submartingale inequality, we bound the second moment of $\mathbf{M}(k)$. Since the variance of the components of $\boldsymbol{\xi}(k)$ is also dominated by $1 - p_1(k)$, we achieve a bound of the order $\alpha^2 \sum_{i=1}^{\infty}(1 - p_1(i)) \mathbb{1}_\Theta$. This series is summable as we have linear convergence to 0, which allows us to choose a constant learning rate $\alpha$.

## 2.2 Associated gradient flow

In this subsection, we consider the associated gradient flow of probabilities $\mathbf{p}(t)$ as a vector-valued function, which solves the ODE

$$\frac{\mathrm{d}}{\mathrm{d}t}\mathbf{p}(t) = \mathbf{p}(t) \odot \big(\mathbf{p}(t) - \|\mathbf{p}(t)\|^2 \mathbf{1}\big) = -\nabla L\big(\mathbf{p}(t)\big), \quad t \geq 0 \tag{9}$$

and is initialized by the probability vector $\mathbf{p}(0)$. By definition, $\frac{\mathrm{d}}{\mathrm{d}t}\sum_{i=1}^{d} p_i(t) = 0$, such that $\sum_{i=1}^{d} p_i(t) = \sum_{i=1}^{d} p_i(0) = 1$. Since the updating rule is multiplicative, the gradient flow produces for all $t \geq 0$ a probability vector. The gradient flow Eq. (9) also occurs as a specific *replicator equation* in evolutionary game theory, see Hofbauer and Sigmund [17, Chapter 7]. Elementary properties and an explicit solution for the gradient flow with $d = 2$ are derived in Section 2.2. Although the loss function $L(\mathbf{p})$ in Eq. (6) does not satisfy a global Polyak-Łojasiewicz condition, in particular it is not globally convex, we can deduce the following convergence for the ODE Eq. (9).

**Theorem 2.4.** *Assume*

$$p_1(0) \geq \max_{i=2,\dots,d} p_i(0) + \Delta,$$

*for some $\Delta > 0$. Then*

$$\|\mathbf{e}_1 - \mathbf{p}(t)\|_1 \leq 2\big(1 - p_1(0)\big) \exp\Big(-\frac{\Delta}{d}(1 + (d-1)\Delta)t\Big),$$

*that is linear convergence of $\mathbf{p}(t) \to \mathbf{e}_1$ as $t \to \infty$.*

## 2.3 Biological plausibility of the proposed learning rule

We study a biological neural network consisting of $d$ input (or presynaptic) neurons, which are connected to one output (or postsynaptic) neuron. For the subsequent argument, we assume that the spike times of the $d$ input neurons are given by the corresponding jump times of $d$ independent Poisson processes $(X_t^{(1)})_{t\geq 0}, \ldots, (X_t^{(d)})_{t\geq 0}$ with respective intensities $\lambda_1, \ldots, \lambda_d$. All neurons are excitatory and each connection between an input neuron $j \in [d]$ and the output neuron has a time varying and non-negative synaptic strength parameter, which we denote by $w_j(t) \geq 0$.

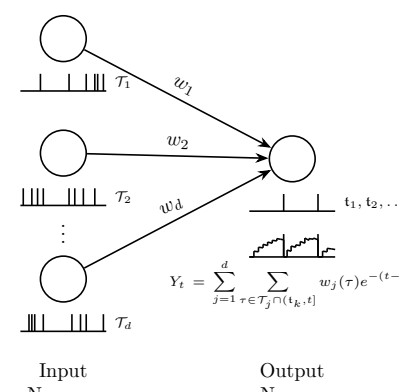

An idealized model is that a spike of the $j^{\text{th}}$ input neuron at time $\tau$ causes an exponentially decaying contribution to the postsynaptic membrane potential of the form $t \mapsto w_j(\tau)Ce^{-c(t-\tau)}\mathbb{1}_{t\geq\tau}$. We set the parameters $c, C > 0$ to one, as this can always be achieved by a time change $t \mapsto tc$ and a change of units of the voltage in the membrane potential.

Figure 3: Considered biological neural network with spike trains and membrane potential $Y_t$ of the postsynaptic neuron.

If $\mathcal{T}_j$ denotes the spike times of neuron $j \in [d]$, the postsynaptic membrane potential $(Y_t)_{t\geq 0}$ is given by $Y_0 = 0$ and $Y_t = \sum_{j=1}^d \sum_{\tau\in\mathcal{T}_j, \tau\leq t} w_j(\tau)e^{-(t-\tau)}$ for all $t \geq 0$ until $Y_t \geq S$, where $S > 0$ is a given threshold value. Once the threshold $S$ is surpassed, the postsynaptic neuron emits a spike and its membrane potential is reset to its rest value, which we assume to be 0. Afterwards, the incoming spikes will contribute to rebuilding the postsynaptic membrane potential. If $\mathsf{t}_0 := 0 < \mathsf{t}_1 < \mathsf{t}_2 < \ldots$ denote the postsynaptic spike times, the membrane potential at arbitrary time is therefore given by

$$Y_t = \sum_{j=1}^d \sum_{\tau\in\mathcal{T}_j\cap(\mathsf{t}_k, t]} w_j(\tau)e^{-(t-\tau)} \quad \text{for all } \mathsf{t}_k < t \leq \mathsf{t}_{k+1}. \tag{10}$$

We consider the following pair-based spike-timing-dependent plasticity (STDP) rule ([14, Section 19.2.2]): A spike of the $j^{\text{th}}$ presynaptic neuron at time $\tau$ causes the weight parameter function $t \mapsto w_j(t)$ to decrease at $\tau$ by $\alpha e^{-(\tau-\mathsf{t}_-)}$, where $\mathsf{t}_-$ is the last postsynaptic spike time before $\tau$ and to increase at any postsynaptic spike time $\mathsf{t}_k$ by $\alpha \sum_{\tau\in\mathcal{T}_j\cap(\mathsf{t}_k,\mathsf{t}_{k+1}]} e^{-(\tau-\mathsf{t}_k)}$, with $\alpha > 0$ the learning rate. As common in the literature, spike times that occurred before $\mathsf{t}_k$ only have a minor influence and are neglected in the updating of the weights after $\mathsf{t}_k$. The term $\sum_{\tau\in\mathcal{T}_j\cap(\mathsf{t}_k,\mathsf{t}_{k+1}]} e^{-(\tau-\mathsf{t}_k)}$ is then the trace ([14, Equation (19.12)]) of the $j^{\text{th}}$ presynaptic neuron at time $\mathsf{t}_{k+1}$.

For mathematical convenience, we will assume that all weight-updates in $(\mathsf{t}_k, \mathsf{t}_{k+1}]$ are delayed to the postsynaptic spike times $\mathsf{t}_k$ in the sense that the learning rule becomes

$$w_j(\mathsf{t}_{k+1}) = w_j(\mathsf{t}_k)\left(1 + \alpha\left(\sum_{\tau\in\mathcal{T}_j\cap(\mathsf{t}_k,\mathsf{t}_{k+1}]} e^{-(\mathsf{t}_{k+1}-\tau)} - e^{-(\tau-\mathsf{t}_k)}\right)\right), \quad k = 0, 1, \ldots \tag{11}$$

The postsynaptic spike times $\mathsf{t}_k$ are the moments at which the postsynaptic membrane potential $Y_t$ reaches the threshold $S$. They depend on the presynaptic spike times, however, the exact dependence is hard to characterise in the assumed model. For mathematical tractability, we will instead work with an adjusted rule to select the postsynaptic spike times $\mathsf{t}_1, \mathsf{t}_2, \ldots$ Since $Y_t$ only increases at the presynaptic spike times, $\mathsf{t}_{k+1}$ has to happen at a presynaptic spike time. Denote by $\tau_{j1}, \tau_{j2}, \ldots$ the spike times of the $j$-th presynaptic neuron after the previous postsynaptic spike time $\mathsf{t}_k$ in increasing order. The distribution of $\mathsf{t}_{k+1}|\mathsf{t}_k$ is completely determined by the probabilities

$$\mathbb{P}\big(\mathsf{t}_{k+1} = \tau_{j\ell}\big) = \mathbb{P}\big(\mathsf{t}_{k+1} = \tau_{j\ell}\big|\mathsf{t}_{k+1} \in (\tau_{jm})_{m\geq 1}\big)\mathbb{P}\big(\mathsf{t}_{k+1} \in (\tau_{jm})_{m\geq 1}\big), \; j = 1, \ldots, d, \; \ell = 1, \ldots$$

Based on probabilistic arguments related to the underlying Poisson processes that we outline in Section A.3, we replace the probabilities $P\big(\mathsf{t}_{k+1} \in (\tau_{jm})_{m\geq 1}\big)$ by the probabilities

$$\frac{\lambda_j w_j(\mathsf{t}_k)}{\sum_{\ell=1}^d \lambda_\ell w_\ell(\mathsf{t}_k)}. \tag{12}$$

Working with Eq. (12) instead of $\mathbb{P}(t_{k+1} \in (\tau_{jm})_{m \geq 1})$ results in an approximation of the distribution of $t_{k+1}$. Lemma A.7 describes a setting, where Eq. (12) is exact. If all weights are much larger than the threshold $S$, every presynaptic spike causes a postsynaptic spike. The proof of Lemma A.7 can be adapted to this case to show that the probability that the $j^{\text{th}}$ neuron emits the first spike is $\lambda_j / \sum_{\ell=1}^{d} \lambda_\ell$. Since Hebbian learning is intrinsically unstable, we argue that the proposed approximation describes the dynamic at the beginning of the learning process. This view is corroborated by experimental results, see point (vi) of Morrison et al. [28, Section 2.1].

Compared to the original definition of $t_{k+1}$, the proposed sampling scheme has the advantage that the presynaptic spike times, which were not selected as postsynaptic spike time, add centred noise to the updates. More precisely, one can show that by the construction of $t_k$ and the properties of the underlying Poisson processes, the conditional distribution $\tau | \{ \tau \in (t_k, t_{k+1}) \}$ is uniformly distributed on $(t_k, t_{k+1})$. By the symmetry relation $e^{-(b-u)} - e^{-(u-a)} = -(e^{-(b-v)} - e^{-(v-a)}) \in [-1, 1]$, which holds for all real numbers $a \leq u \leq b$ with $v = b + a - u \in [a, b]$, this implies that conditionally on $\tau \in (t_k, t_{k+1})$, the random variable $e^{-(t_{k+1} - \tau)} - e^{-(\tau - t_k)}$ is centred and supported on $[-1, 1]$. The update rule Eq. (11) then becomes

$$w_j(t_{k+1}) = w_j(t_k)\left(1 + \alpha \mathbb{1}_{\{j = j^*(k+1)\}}\left(1 - e^{t_k - t_{k+1}}\right) + \alpha \sum_{\tau \in \mathcal{T}_j \cap (t_k, t_{k+1})} Z(\tau, j)\right), \qquad (13)$$

with centred random variables $Z(\tau, j)$ satisfying $|Z(\tau, j)| \leq 1$. Assuming that the postsynaptic firing rate is slow compared to the learning dynamic, we discard the term $e^{t_k - t_{k+1}} \ll 1$. Since $j^*(k+1)$ follows a multinomial distribution with parameters $\lambda_j w_j(t_k)/(\sum_{\ell=1}^{d} \lambda_\ell w_\ell(t_k))$, the term $\mathbb{1}_{\{j=j^*(k+1)\}}$ corresponds to the $j^{\text{th}}$ component of $\mathbf{B}(k)$ in Eq. (1). This motivates the learning rule Eq. (1). Additional details on the derivation are given in Subsection A.3 of the supplementary material.

## 3 A mirror descent perspective

In this section, we rewrite the gradient flow Eq. (9) as natural gradient descent on the probability simplex and relate the discrete-time learning rule Eq. (3) for the probabilities $\mathbf{p}$ to noisy mirror gradient descent.

Recall from Eq. (5) that the learning rule Eq. (3) can be interpreted as noisy gradient descent with respect to the loss function $L$ from Eq. (6) in the Euclidean geometry. As we consider a flow on probability vectors, a different perspective is to use the natural geometry of the probability simplex. To this end, we consider the interior of the probability simplex $\mathcal{M} := \text{int}(\mathfrak{P})$ as a Riemannian manifold with tangent space $\mathcal{T}_{\mathbf{p}}\mathcal{M} = \{\mathbf{x} \in \mathbb{R} : \mathbf{1}^\top \mathbf{x} = 0\}$ for every $\mathbf{p} \in \mathcal{M}$. A natural metric on $\mathcal{M}$ is given by the *Fisher information metric / Shahshahani metric* [4, 17], which is induced by the metric tensor $d_{\mathbf{p}} : \mathcal{T}_{\mathbf{p}}\mathcal{M} \times \mathcal{T}_{\mathbf{p}}\mathcal{M} \to \mathbb{R}, (\mathbf{u}, \mathbf{v}) \mapsto \mathbf{u}^\top \text{diag}(\mathbf{p})^{-1}\mathbf{v}$ at $\mathbf{p} \in \mathcal{M}$. Here, $\text{diag}(\mathbf{p}) \in \mathbb{R}^{d \times d}$ is the diagonal matrix with diagonal entries given by $\mathbf{p}$. We refer to Figure 1 of Mertikopoulos and Sandholm [26] for an illustration of unit balls in this metric. The (Riemannian) gradient of the loss function $\widetilde{L}(\mathbf{p}) = -\|\mathbf{p}\|^2/2$ with respect to $d_{\mathbf{p}}$ is given by

$$\nabla_{d_{\mathbf{p}}} \widetilde{L}(\mathbf{p}) = \text{diag}(\mathbf{p})\nabla\widetilde{L}(\mathbf{p}) \in \mathcal{T}_{\mathbf{p}}\mathcal{M},$$

where we denote by $\nabla\widetilde{L}$ the Euclidean gradient of $\widetilde{L}$. The Riemannian gradient flow is called *natural gradient flow* in information geometry [3] and *Shahshahani gradient flow* in evolutionary game theory [18]. When transforming the Euclidean gradient flow for $L$ to a Riemannian gradient flow on the probability simplex, the part $+\|\mathbf{p}\|^2\mathbf{1}$ is orthogonal to $\mathcal{T}_{\mathbf{p}}\mathcal{M}$. Consequently, it does not contribute a direction on the probability simplex. Consequently, the Riemannian gradient flow of $\widetilde{L}$ and the Euclidean gradient flow of $L$ coincide.

The mirror descent algorithm [29] prescribes the discrete-time optimisation algorithm

$$\mathbf{p}(k+1) \in \underset{\mathbf{p} \in \mathcal{M}}{\text{argmin}}\left\{\mathbf{p}^\top \nabla f(\mathbf{p}(k)) + \frac{1}{\alpha}\Phi(\mathbf{p}, \mathbf{p}(k))\right\}, \quad k = 0, 1, \ldots, \qquad (14)$$

where $f : \mathcal{M} \to \mathbb{R}$ is the function to be minimised and $\Phi : \mathcal{M} \times \mathcal{M} \to \mathbb{R}_+$ is a suitable proximity function. Euclidean gradient descent is recovered by the choice $\Phi(\mathbf{p}, \mathbf{p}(k)) = \|\mathbf{p} - \mathbf{p}(k)\|^2$. It

is well-known that the natural gradient flow is the continuous-time analogue of the *exponentiated gradient descent* or *entropic mirror descent*, where $\Phi(\mathbf{p}, \mathbf{p}(k)) = \text{KL}(\mathbf{p}\|\mathbf{p}(k))$ is chosen as the Kullback–Leibler divergence between $\mathbf{p}$ and $\mathbf{p}(k)$ [2, 47, 33]. Consequently, the gradient flow Eq. (9) can also be viewed as continuous-time version of entropic mirror descent with respect to $f = \widetilde{L}$. This connection transfers to the discrete-time and noisy updating rule Eq. (3). An alternative approach for connecting our proposed discrete-time learning rule Eq. (3) to entropic mirror descent is included in Subsection A.4 of the supplementary material.

# 4 Multiple weight vectors

The learning rule Eq. (1) aligns the output neuron with the input neuron of the highest intensity, but no information about the remaining input neurons is unveiled. As a proof-of-concept, we generalise the learning algorithm Eq. (1) to estimate the order of the intensities $\lambda_1, \ldots, \lambda_d$. To this end, we consider $d$ different output neurons, which are connected to the $d$ input neurons via the weight vectors $\mathbf{w}_1, \ldots, \mathbf{w}_d \in \mathbb{R}^d$. The weights at time $k$ are combined into the matrix

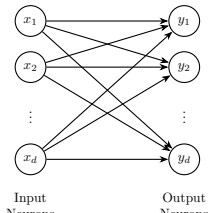

$$\mathbf{W}(k) = [\mathbf{w}_1(k) \quad \cdots \quad \mathbf{w}_d(k)] \in \mathbb{R}^{d \times d}, \quad k = 0, 1, \ldots$$

and the corresponding probabilities $\mathbf{p}_1, \ldots, \mathbf{p}_d$ are combined into the matrix $\mathbf{P}(k) = [\mathbf{p}_1(k) \quad \cdots \quad \mathbf{p}_d(k)] \in \mathbb{R}^d$. By reordering the neurons, we can achieve $\lambda_1 \le \lambda_2 \le \cdots \le \lambda_d$. If the intensities are strictly ordered, our goal is the alignment of the $j^{\text{th}}$ output neuron with the $j^{\text{th}}$ input neuron, which amounts to the convergence of $\mathbf{P}(k)$ to the identity matrix $\mathbb{I} \in \mathbb{R}^{d \times d}$ as time

Figure 4: Neural network with $d$ input/output neurons.

increases. If multiple intensities are equal, convergence is up to permutations within the group of equal intensities. We propose Algorithm 1, which constitutes an STDP rule as lines 3 - 4 can be implemented using the spike-trains and the learning rule Eq. (11).

---

**Algorithm 1:** Aligning multiple output neurons

**Input:** $K \in \mathbb{N}$: number of iterations, $\mathbf{W}(0) \in \mathbb{R}^{d \times d}$: weight initialisation, $\alpha_1, \ldots, \alpha_d$: learning rates of the output neurons.

1 **for** $k = 0, 1, \ldots, K-1$ **do**
2    **for** $j = 1, \ldots, d$ **do**
3       Receive $\mathbf{B}_j(k) \sim M(1, \mathbf{p}_j(k))$ with $\mathbf{p}_j(k) \leftarrow \boldsymbol{\lambda} \odot \mathbf{w}_j(k)/\boldsymbol{\lambda}^\top \mathbf{w}_j(k)$ and $\mathbf{Z}_j(k) \sim \text{Unif}([-1, 1]^d)$ from spike trains;
4       Compute the base change $\Delta \mathbf{w}_j(k) \leftarrow \alpha_j \mathbf{w}_j(k) \odot (\mathbf{B}_j(k) + \mathbf{Z}_j(k))$;
5       Update

$$\mathbf{w}_j(k+1) \leftarrow \Delta \mathbf{w}_j(k) - \sum_{i=1}^{j-1} \frac{(\Delta \mathbf{w}_j(k))^\top \mathbf{w}_i(k)}{\|\mathbf{w}_i(k)\|^2} \mathbf{w}_i(k);$$

6    **end**
7 **end**

**Output:** The weight evolution $\mathbf{W}(k) = [\mathbf{w}_1(k) \cdots \mathbf{w}_d(k)]$, $k = 0, \ldots, K$ and probability evolution $\mathbf{P}(k) = [\mathbf{p}_1(k) \cdots \mathbf{p}_d(k)]$, $k = 1, \ldots, K$.

---

Algorithm 1 is inspired by Sanger's rule [35] for learning $d$ principal components in streaming PCA. The first weight vector $\mathbf{w}_1(k)$ aligns with $\mathbf{e}_1$ by Theorem 2.2 since its dynamic equals the learning rule Eq. (1). By removing the components of the change $\Delta \mathbf{w}_j(k)$ in the direction of $\mathbf{w}_1(k), \ldots, \mathbf{w}_{j-1}(k)$ in line 5 of Algorithm 1, the weight vector $\mathbf{w}_j(k)$ is forced to converge to $\mathbf{e}_j$, similarly to the Gram–Schmidt algorithm.

Simulations of the corresponding probability matrix $\mathbf{P}(k)$ with varying learning rates and $\mathbf{Z}$ drawn i.i.d. from $\text{Unif}([-1, 1]^d)$ are included in Figure 5. We choose different learning rates for the $d$ vectors satisfying $\alpha_1 > \cdots > \alpha_d > 0$. This ordering ensures fast convergence of the lower order weight vectors to the correct standard basis vector and counteracts the impact of initial misalignments of the higher order weight vectors. The simulation of Algorithm 1 shown in Figure 5 displays a decrease

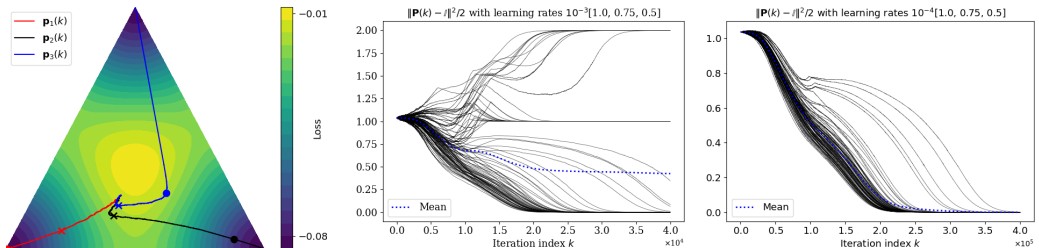

Figure 5: Probability matrix $\mathbf{P}(k)$ arising from the weight dynamic $\mathbf{W}(k)$ of Algorithm 1 for dimensions $n = d = 3$. The weights are initialised equally, and the intensities are given by $\boldsymbol{\lambda} = (10, 7.5, 5)^\top$. The resulting initial probabilities are $\mathbf{p}_1(0) = \mathbf{p}_2(0) = \mathbf{p}_3(0) = (4/9, 1/3, 2/9)^\top$. Left: A single trajectory with learning rates $10^{-3}(1, 0.75, 0.5)^\top$ and $4 \times 10^4$ iterations. The markers $\times$ and $\bullet$ correspond to the probabilities at $k = 4 \times 10^3$ and $k = 10^4$. Middle & right: The Frobenius error $\|\mathbf{P}(k) - \mathbb{I}\|^2/2$ of 100 trajectories with learning rates $10^{-3}(1, 0.75, 0.5)^\top$ and $10^{-4}(1, 0.75, 0.5)^\top$, respectively.

of the Frobenius error $\|\mathbf{P}(k) - \mathbb{I}\|^2/2$ over the iteration index $k$, when averaged (blue line). Nevertheless, we observe that for a single trajectory, the error can plateau around 1 and 2. Given that the probability vectors tend to converge to standard basis vectors $\{\mathbf{e}_1, \ldots, \mathbf{e}_d\}$, a non-vanishing error is due to an incorrect ordering or duplicates. Consequently, the error $\|\mathbf{P}(k) - \mathbb{I}\|^2/2$ corresponds to the number of output neurons aligning with the incorrect input neuron, and plateaus at 1, 2 and 3 can arise. Theorem 2.2 shows that this phenomenon can be mitigated by slower learning rates, which is corroborated by decreasing the base learning rate from $10^{-3}$ to $10^{-4}$ in the simulation. A rigorous mathematical analysis of the Algorithm 1 is challenging due to joint updates in all read-out neurons. In Section A.5 we show that Theorem 2.2 is applicable if the learning is split into disjoint learning periods, and we derive theoretical convergence guarantees.

## 5 Extensions, discussion and limitations

**Time-inhomogeneous intensities.** We considered input spike trains generated from Poisson point processes with fixed intensity. It is natural to extend this to time-inhomogeneous intensities. Here we assume that the intensities of the input neurons are constant on the interval $(k, k+1]$ and are stored in the vector $\boldsymbol{\lambda}(k)$. The intensities $\boldsymbol{\lambda}(k)$ and weights $\mathbf{w}(k)$ determine the spike-triggering-probabilities $\mathbf{p}(k) = \boldsymbol{\lambda}(k) \odot \mathbf{w}(k)/\boldsymbol{\lambda}(k)^\top \mathbf{w}(k) = \mathbb{E}[\mathbf{Y}(k)]$ and the update formula Eq. (2) becomes

$$\mathbf{p}(k+1) = \frac{\boldsymbol{\lambda}(k+1) \odot (\mathbf{w}(k) \odot (\mathbf{1} + \alpha\mathbf{Y}(k)))}{\boldsymbol{\lambda}(k+1)^\top (\mathbf{w}(k) \odot (\mathbf{1} + \alpha\mathbf{Y}(k)))} = \frac{\widetilde{\mathbf{p}}(k) \odot (\mathbf{1} + \alpha\mathbf{Y}(k))}{\widetilde{\mathbf{p}}(k)^\top (\mathbf{1} + \alpha\mathbf{Y}(k))}, \quad k = 0, 1, \ldots, \quad (15)$$

with $\widetilde{\mathbf{p}}(k) := \boldsymbol{\lambda}(k+1) \odot \mathbf{w}(k)/(\boldsymbol{\lambda}(k+1)^\top \mathbf{w}(k))$. A first order Taylor expansion yields

$$\mathbf{p}(k+1) = \widetilde{\mathbf{p}}(k) \odot \left(\mathbf{1} + \alpha(\mathbf{Y}(k) - \widetilde{\mathbf{p}}(k)^\top \mathbf{Y}(k)\mathbf{1})\right) + O(\alpha^2). \quad (16)$$

Since $\mathbb{E}[\mathbf{Y}(k)] = \mathbf{p}(k)$, this means that

$$\mathbf{p}(k+1) = \widetilde{\mathbf{p}}(k) \odot \left(\mathbf{1} + \alpha(\mathbf{p}(k) - \widetilde{\mathbf{p}}(k)^\top \mathbf{p}(k)\mathbf{1})\right) + \text{centered noise} + O(\alpha^2). \quad (17)$$

Extending the gradient flow derivation to the time-inhomogeneous case, one can identify the ODE

$$\frac{\mathrm{d}}{\mathrm{d}t}\mathbf{p}(t) = \mathbf{p}(t) \odot \left(\frac{\mathrm{d}}{\mathrm{d}t}\log(\boldsymbol{\lambda}(t)) + \mathbf{p}(t) - \mathbf{p}(t)^\top \left(\frac{\mathrm{d}}{\mathrm{d}t}\log(\boldsymbol{\lambda}(t)) + \mathbf{p}(t)\right)\mathbf{1}\right), \quad (18)$$

where the logarithm is taken componentwise, as corresponding deterministic dynamic in continuous time, see Section A.6 for a derivation. The ODE can be interpreted as a replicator equation with (time-varying) fitness $\frac{\mathrm{d}}{\mathrm{d}t}\log(\boldsymbol{\lambda}(t)) + \mathbf{p}(t)$, see e.g. Chapter 7 of [18]. An interesting scenario which lies beyond our mathematical analysis amounts to considering time-dependent mean firing rates that are piecewise constant, corresponding to the successive exposition to different input patterns [9].

**Correlated inputs.** Correlated inputs facilitate simultaneous spiking of different input spikes. The probability that input $i$ and $j$ spike at the same time is denoted by $\mathbf{\Gamma}_{ij}$, and naturally $\mathbf{\Gamma}_{ii} := 1$, for all $i \in [d]$. We introduce the $d \times d$ random symmetric matrix $\mathbf{C}(k) = (C_{i,j}(k))_{i,j \in [d]}$ with independently sampled entries

$$C_{j,i}(k) := C_{i,j}(k) \sim \text{Ber}(\mathbf{\Gamma}_{i,j}), \quad \text{for all } i \leq j \in [d].$$

$\mathbf{C}(k)$ describes the simultaneous spiking of the different inputs at the $k$-th post-synaptic spike, i.e. if $C_{i,j}(k)$ is 1 then inputs $i$ and $j$ both spike at the $k$-th post-synaptic spike if either $i$ or $j$ caused the post-synaptic spike. Compared to the original model, the random vector $\mathbf{Z}(k)$ remains the same, but the random vector $\mathbf{B}(k) = (B_1(k), \ldots, B_d(k))^\top$ that encapsulates which of the presynaptic neurons caused the postsynaptic spike is replaced by $\mathbf{S}(k) = \mathbf{C}(k)\mathbf{B}(k)$. $\mathbf{S}(k)$ encodes which of the inputs spike at a post-synaptic spike, and in particular it holds

$$\mathbb{P}\big(S_i(k) = 1 | \mathbf{p}(k)\big) = \mathbf{\Gamma}_{i,\cdot}(k)\mathbf{p}(k), \quad \text{for all } i \in [d],$$

where $\mathbf{\Gamma}_{i,\cdot}(k)$ denotes the $i$-th row of $\mathbf{\Gamma}(k)$. Since the only change in the dynamic of $\mathbf{p}(k)$ is replacing $\mathbf{B}(k)$ by $\mathbf{S}(k)$ in the definition of $\mathbf{Y}(k)$, Eq. (4) still holds true, and we obtain

$$\mathbb{E}\big[\mathbf{p}(k+1)\big|\mathbf{p}(k)\big] \approx \mathbb{E}\Big[\mathbf{p}(k) \odot \Big(\mathbf{1} + \alpha\left(\mathbf{Y}(k) - (\mathbf{p}(k))^\top \mathbf{Y}(k)\mathbf{1}\right)\Big)\Big|\mathbf{p}(k)\Big]$$

$$= \mathbf{p}(k) \odot \Big(\mathbf{1} + \alpha\left(\mathbf{\Gamma}\mathbf{p}(k) - (\mathbf{p}(k))^\top \mathbf{\Gamma}\mathbf{p}(k)\mathbf{1}\right)\Big),$$

which induces the following gradient flow

$$\frac{\mathrm{d}}{\mathrm{d}t}\mathbf{p}(t) = \mathbf{p}(t) \odot (\mathbf{\Gamma}\mathbf{p}(t) - (\mathbf{p}(t))^\top\mathbf{\Gamma}\mathbf{p}(t)\mathbf{1}). \tag{19}$$

This is again a replicator equation with fitness $\mathbf{\Gamma}\mathbf{p}$, compare Section 7 of [18]. The associated Shahshahani-loss is given by $\mathbf{x} \mapsto -\frac{1}{2}\mathbf{x}^\top\mathbf{\Gamma}\mathbf{x}$. Thus, in the correlated model, the probabilities $\mathbf{p}$ follow a flow restricted to the probability simplex aimed at maximising the quadratic form associated to the matrix $\mathbf{\Gamma}$. This property is similar to principal component analysis (PCA), as the goal there is to recover the eigenvector corresponding to the largest eigenvalue of the underlying covariance matrix. Thus, the behaviour of the correlated model can be interpreted as a form of PCA restricted to the probability simplex. Theorem A.13 in the supplementary material generalizes Theorem 2.2 to weakly correlated input neurons and is accompanied by simulations in Figure 6.

**Small biological neural network.** In this paper, we mathematically analyse the convergence behaviour of a small biological neural network with one layer composed of excitatory presynaptic/input neurons and multiple postsynaptic/output neurons. It is natural to generalise the setting to account for inhibitory neurons and more than one layer.

**Weight explosion.** The learning rule Eq. (1) for the weights $\mathbf{w}(k)$ causes them to increase without bound as the iteration index $k$ increases. When the weights exceed the spike threshold, the model becomes biologically implausible and the derivation of the probabilities Eq. (12) is no longer valid. This unstable nature is well-known to be intrinsic to Hebbian learning algorithms and is commonly countered by soft or hard bounds, or by including mean-reverting terms to the dynamic Gerstner et al. [14, pages 497-498]. We follow a different route, namely viewing Hebbian learning as a temporal phase of limited length, which is followed by a stabilising *homeostatic* learning phase. This view is corroborated by experimental results, compare Point (vi) in Morrison et al. [28, Section 2.1].

**Beyond Pair-based STDP rules.** While pair-based learning rules such as Eq. (1) only account for the relative timing of one pre- and one postsynaptic spike time, also the voltage at the location of the synapse should be taken into account [37]. A natural generalization of our framework would be to extend the results to the model proposed in [9].

## Acknowledgments and Disclosure of Funding

All authors acknowledge support from ERC grant A2B (grant agreement no. 101124751). S. G. has been partially funded by the Deutsche Forschungsgemeinschaft (DFG, German Research Foundation) CRC/TRR 388 "Rough Analysis, Stochastic Dynamics and Related Fields Project ID 516748464, Project B07. Parts of the research were carried out while the authors visited the Simons Institute in Berkeley. We thank Xiangyuan Li for pointing out some typos.

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

# A Technical Appendix

We first study the loss landscape

$$\mathbf{p} \mapsto L(\mathbf{p}) = -\frac{1}{3} \sum_{i=1}^{d} p_i^3 + \frac{1}{4} \Big( \sum_{i=1}^{d} p_i^2 \Big)^2.$$

Lemma 2.1 identifies the stationary points if we view the landscape as a function on $\mathbb{R}^d$.

*Proof of Lemma 2.1.* The formula Eq. (7) for the gradient $\nabla L(\mathbf{p})$ shows that the set of critical points is given by

$$\begin{aligned}
\mathrm{Crit} &:= \{\mathbf{p} \in \mathfrak{P} : \nabla L(\mathbf{p}) = 0\} \\
&= \{\mathbf{p} \in \mathfrak{P} : p_i \in \{0, \|\mathbf{p}\|^2\} \ \forall i \in [d]\} \\
&= \Big\{ \mathbf{p} \in \mathfrak{P} : \exists n \in \{1, \dots, d\}, S \subset [d] \text{ with } \#S = n \text{ such that } \mathbf{p} = \frac{1}{n} \sum_{j \in S} \mathbf{e}_j \Big\}.
\end{aligned}$$

To identify local the extrema, we compute the Hessian matrix

$$J(\mathbf{p}) = 2\mathbf{p}\mathbf{p}^\top + \|\mathbf{p}\|^2 \mathbb{I} - 2\,\mathrm{diag}(\mathbf{p}) \in \mathbb{R}^{d \times d}, \quad \mathbf{p} \in \mathbb{R}^d,$$

where $\mathrm{diag}(\mathbf{p}) \in \mathbb{R}^{d \times d}$ is the diagonal matrix with diagonal entries given by $\mathbf{p}$. Substituting a critical point $\mathbf{p}^*$ with $n \in [d]$ non-zero entries yields (up to permutations of rows and columns)

$$J(\mathbf{p}^*) = \frac{1}{n} \begin{pmatrix} J_n & 0 \\ 0 & \mathbf{I}_{d-n} \end{pmatrix}, \quad J_n = -\mathbf{I}_n + \frac{2}{n} \mathbf{1}_{n \times n}, \quad n \in [d],$$

where $\mathbf{1}_{n \times n}$ is the $n \times n$ matrix consisting of ones. The corresponding eigenvalues are

$$\begin{cases}
\frac{1}{n} > 0 & \text{with multiplicity 1 and eigenspace } \mathbf{E}_n := \mathrm{span}(\sum_{i=1}^{d} \mathbf{e}_i), \\
-\frac{1}{n} < 0 & \text{with multiplicity } n - 1 \text{ and eigenspace } \mathbf{E}_n^\perp, \\
\frac{1}{n} > 0 & \text{with multiplicity } d - n \text{ and eigenspace } \mathrm{span}(\mathbf{e}_{n+1}, \dots, \mathbf{e}_d),
\end{cases} \tag{20}$$

where $\mathbf{E}_n^\perp$ is the orthogonal complement of $\mathbf{E}_n$ in $\mathrm{span}(\mathbf{e}_1, \dots, \mathbf{e}_n)$. Consequently, only those critical points $\mathbf{p}^* \in \mathrm{Crit}$ are local minima, which have $n = 1$, i.e. $\mathbf{p}^* \in \{\mathbf{e}_1, \dots, \mathbf{e}_d\}$. Since all local minima attain the same loss $-1/12$ and $L(\mathbf{p}) \to \infty$ as $\|\mathbf{p}\| \to \infty$, every local minimum is also a global minimum. $\qquad\square$

*Remark* A.1. The eigenvalues of the Hessian of the loss function computed in Eq. (20) also imply that if $n \geq 2$, then $\mathbf{p}^* \in \mathrm{Crit}$ is a saddle point in $\mathbb{R}^d$. Interestingly, when restricting to directions within the probability simplex, the case $n = d$ is not a saddle point, but a maximum, since the direction $\sum_{i=1}^{d} \mathbf{e}_i$ is orthogonal to $\mathfrak{P}$.

## A.1 Proofs for Subsection 2.1

In the following we will always assume that

$$p_1(0) \geq \max_{i=2,\dots,d} p_i(0) + \Delta, \tag{21}$$

for some $\Delta \in (0, 1)$. This is a deterministic constraint. The randomness occurs because of the noise in the updates. We assume that all random variables are defined on a filtered probability space $(\Omega, \mathcal{F}, \mathbb{P})$, and denote by $\mathcal{F}_n$, $n = 0, 1, \dots$, the natural filtration of $(\mathbf{B}(n), \mathbf{Z}(n))_{n \in \mathbb{N}}$. By a slight abuse of notation we also introduce $\mathcal{F}_{-1} = \{\emptyset, \Omega\}$. In particular it then holds that $\mathbf{p}_n$ is $\mathcal{F}_{n-1}$-measurable for $n = 0, 1, \dots$. The starting point for the proof of the linear convergence of STDP is given by the following Lemma, which explicitly bounds the error term in the Taylor approximation contained in Eq. (3). Recall that $|Y_i(k)| \leq Q$, for all $i \in [d], k = 0, 1, \dots$

**Lemma A.2.** *For $i \in [d]$ and $k = 0, 1, \dots$ define*

$$\xi_i(k) := p_i(k) \Big( \mathbb{E}\Big[ Y_i(k) - \sum_{j=1}^{d} p_j(k) Y_j(k) \Big| \mathcal{F}_{k-1} \Big] - \Big( Y_i(k) - \sum_{j=1}^{d} p_j(k) Y_j(k) \Big) \Big), \tag{22}$$

*and assume $\alpha < 1/Q$. Then for any $i \in [d]$ and $k = 0, 1, \ldots$, there exists a random variable $\theta_i(k)$, satisfying*

$$|\theta_i(k)| \leq \alpha^2 \frac{2Q^2}{(1 - Q\alpha)^3} p_i(k)\big(1 - p_i(k)\big), \qquad \textit{almost surely,}$$

*such that*

$$p_i(k+1) = p_i(k) + \alpha p_i(k)\big(p_i(k) - \|\mathbf{p}(k)\|^2\big) - \alpha\xi_i(k) - \theta_i(k).$$

*Proof.* By definition

$$p_i(k+1) = p_i(k)\frac{1 + \alpha Y_i(k)}{1 + \alpha \sum_{j=1}^{d} p_j(k)Y_j(k)}, \quad k = 0, 1, \ldots, \quad i \in [d].$$

Now, for $a, b \in [-Q, Q]$ the first two derivatives of the function

$$f(x) \colon \left(0, \frac{1}{Q}\right) \to \mathbb{R}, \quad x \mapsto \frac{1 + ax}{1 + bx},$$

are given by

$$f'(x) = \frac{a(1 + bx) - b(1 + ax)}{(1 + bx)^2} = \frac{a - b}{(1 + bx)^2}$$

$$f''(x) = -2b\frac{a - b}{(1 + bx)^3}.$$

Thus, a Taylor expansion around $x = 0$ gives that there exists some $\gamma \in (0, x)$, such that

$$f(x) = 1 + (a - b)x - b\frac{a - b}{(1 + b\gamma)^3}x^2.$$

Hence we obtain that for some $\gamma \in (0, \alpha)$,

$$p_i(k+1) = p_i(k) + \alpha p_i(k)\Big(Y_i(k) - \sum_{j=1}^{d} p_j(k)Y_j(k)\Big)$$

$$- \alpha^2 p_i(k)\frac{\sum_{j=1}^{d} p_j(k)Y_j(k)\Big(Y_i(k) - \sum_{j=1}^{d} p_j(k)Y_j(k)\Big)}{\big(1 + \gamma \sum_{j=1}^{d} p_j(k)Y_j(k)\big)^3}.$$

Using that $|Y_i(k)| \leq Q$ almost surely for all $i \in [d], k = 0, 1, \ldots$, the absolute value of the error term can be bounded as follows

$$\left|\alpha^2 p_i(k)\frac{\sum_{j=1}^{d} p_j(k)Y_j(k)\Big(Y_i(k) - \sum_{j=1}^{d} p_j(k)Y_j(k)\Big)}{\big(1 + \gamma \sum_{j=1}^{d} p_j(k)Y_j(k)\big)^3}\right|$$

$$\leq \alpha^2 p_i(k)\frac{Q}{(1 - Q\alpha)^3}\Big|Y_i(k) - \sum_{j=1}^{d} p_j(k)Y_j(k)\Big|$$

$$\leq \alpha^2 p_i(k)\frac{Q}{(1 - Q\alpha)^3}\Big((1 - p_i(k))|Y_i(k)| + \sum_{j=1,j\neq i}^{d} p_j(k)|Y_j(k)|\Big)$$

$$\leq \alpha^2 \frac{2Q^2}{(1 - Q\alpha)^3} p_i(k)(1 - p_i(k)).$$

Since $p_i(k)$ is $\mathcal{F}_{k-1}$-measurable for any $i \in [d]$ and $\mathbb{E}[Y_i(k) \mid \mathcal{F}_{k-1}] = p_i(k)$, we also obtain

$$\xi_i(k) = p_i(k)\Big(p_i(k) - \|\mathbf{p}(k)\|^2 - Y_i(k) + \sum_{j=1}^{d} p_j(k)Y_j(k)\Big),$$

which concludes the proof. $\qquad\square$

For $\Delta$ given in Eq. (21), we define a sequence of benign events

$$\Omega(k) := \Big\{ p_1(u) \geq \max_{i=2,\ldots,d} p_i(u) + \frac{\Delta}{2}, \; \forall u \in [k] \Big\}, \quad k = 1, 2, \ldots,$$

and due to the assumption Eq. (21) we set $\Omega(0) = \Omega$. On the above events, the gradient is bounded away from zero and

$$p_1(k) = \frac{1}{d} \sum_{j=1}^{d} \big( p_1(k) - p_j(k) \big) + \frac{1}{d} \geq \frac{(d-1)\Delta}{2d} + \frac{1}{d}. \tag{23}$$

Using these properties, we can prove a recursive upper bound for $1 - p_1(k)$.

**Proposition A.3.** *If*

$$0 < \alpha \leq \frac{(1 - Q\alpha)^3}{8Q^2} \Delta,$$

*then, on the event $\Omega(k)$,*

$$1 - p_1(k+1) \leq \Big( 1 - \alpha \frac{\Delta}{4d} \Big( 1 + \frac{\Delta}{2}(d-1) \Big) \Big) \big( 1 - p_1(k) \big) + \alpha \xi_1(k).$$

*Proof.* By definition, we have on the event $\Omega(k)$,

$$p_1(k) - \|\mathbf{p}(k)\|^2 = \sum_{j=1}^{d} p_j(k) \big( p_1(k) - p_j(k) \big)$$

$$\geq \frac{\Delta}{2} \big( 1 - p_1(k) \big). \tag{24}$$

The constraint imposed on the learning rate implies that $\alpha < 1/Q$ and Lemma A.2 becomes applicable. Now, combining the previous inequality with the assumption on $\alpha$ and applying Lemma A.2 with $i = 1$, as well as Eq. (23), gives, on the event $\Omega(k)$,

$$1 - p_1(k+1)$$
$$\leq 1 - p_1(k) - \alpha \frac{\Delta}{2} p_1(k) \big( 1 - p_1(k) \big) + \alpha \xi_1(k) + \theta_1(k)$$
$$\leq 1 - p_1(k) - \alpha \Big( \frac{\Delta}{2} - \frac{2Q^2}{(1 - Q\alpha)^3} \alpha \Big) p_1(k) \big( 1 - p_1(k) \big) + \alpha \xi_1(k)$$
$$\leq 1 - p_1(k) - \frac{\Delta}{4} \alpha p_1(k) \big( 1 - p_1(k) \big) + \alpha \xi_1(k)$$
$$\leq \Big( 1 - \alpha \frac{\Delta}{4d} \Big( 1 + \frac{\Delta}{2}(d-1) \Big) \Big) \big( 1 - p_1(k) \big) + \alpha \xi_1(k).$$

This concludes the proof. $\qquad\square$

Having understood the dynamics of $\mathbf{p}$ on the favourable event $\Omega(k)$, we aim for a lower bound for its probability. A key step is the following Lemma, which states that $\Omega(k)$ is fulfilled as soon as

$$M_j(k) := \sum_{\ell=0}^{k} \alpha \xi_j(\ell) \mathbb{1}_{\Omega(\ell)}, \quad k = 0, 1, \ldots \tag{25}$$

with $\xi_j(\ell)$ defined in Eq. (22), exhibit a uniform concentration behaviour.

**Lemma A.4.** *Define the sets*

$$E_j(k) := \Big\{ \max_{u \in [k]} |M_j(u)| \leq \frac{\Delta}{4} \Big\}, \quad E(k) := \bigcap_{j=1}^{d} E_j(k), \quad k = 0, 1, \ldots.$$

*Then if*

$$0 < \alpha \leq \Delta^2 \frac{(1 - Q\alpha)^3}{16Q^2},$$

*the following set inclusion holds for any $k = 0, 1, \ldots$*

$$E(k) \subseteq \Omega(k+1).$$

*Proof.* Let $u \in \{2, \ldots, d\}$ be arbitrary. It follows by Lemma A.2, that on $\Omega(k)$ the bound

$$p_u(k+1) = p_u(k) + \alpha p_u(k)\big(p_u(k) - \|\mathbf{p}(k)\|^2\big) - \alpha\xi_u(k) - \theta_u(k)$$
$$\leq p_u(k) + \alpha p_u(k)\big(p_1(k) - \|\mathbf{p}(k)\|^2\big) - \alpha\xi_u(k) - \theta_u(k)$$

holds. Consequently, on $\Omega(k)$, we have

$$p_1(k+1) - p_u(k+1)$$
$$= p_1(k) - p_u(k) + \alpha\big(p_1(k) - p_u(k)\big)\big(p_1(k) - \|\mathbf{p}(k)\|^2\big) + \alpha\big(\xi_u(k) - \xi_1(k)\big) - \theta_1(k) + \theta_u(k).$$

We have $p_u(k) \leq 1 - p_1(k)$ and thus, on $\Omega(k)$,

$$-\theta_1(k) + \theta_u(k) \geq -\alpha^2 \frac{2Q^2}{(1-Q\alpha)^3}\Big(p_1(k)\big(1 - p_1(k)\big) + p_u(k)(1 - p_u(k))\Big)$$
$$\geq -\alpha^2 \frac{2Q^2}{(1-Q\alpha)^3}\Big(1 - p_1(k) + p_u(k)\Big)$$
$$\geq -\alpha^2 \frac{4Q^2}{(1-Q\alpha)^3}\big(1 - p_1(k)\big)$$
$$\geq -\alpha\frac{\Delta^2}{4}\big(1 - p_1(k)\big),$$

invoking the constraint on the learning rate in the last step. From the assumptions on $\alpha$ we deduce that on the event $\Omega(k)$,

$$p_1(k+1) - p_u(k+1) - \alpha(\xi_u(k) - \xi_1(k))$$
$$\geq p_1(k) - p_u(k) + \alpha\big(p_1(k) - p_u(k)\big)\big(p_1(k) - \|\mathbf{p}(k)\|^2\big) - \alpha\frac{\Delta^2}{4}\big(1 - p_1(k)\big)$$
$$\geq p_1(k) - p_u(k) + \alpha\frac{\Delta}{2}\big(p_1(k) - p_u(k)\big)\big(1 - p_1(k)\big) - \alpha\frac{\Delta^2}{4}\big(1 - p_1(k)\big)$$
$$\geq p_1(k) - p_u(k) + \alpha\frac{\Delta^2}{4}\big(1 - p_1(k)\big) - \alpha\frac{\Delta^2}{4}\big(1 - p_1(k)\big)$$
$$\geq p_1(k) - p_u(k),$$

where we applied Eq. (24) in the third to last inequality. Because of $\Omega(k) \subseteq \Omega(k-1)$ it then follows,

$$(p_1(k+1) - p_u(k+1))\mathbb{1}_{\Omega(k)} \geq (p_1(k) - p_u(k))\mathbb{1}_{\Omega(k)} + \alpha(\xi_u(k) - \xi_1(k))\mathbb{1}_{\Omega(k)}$$
$$= \mathbb{1}_{\Omega(k)}\Big((p_1(k) - p_u(k))\mathbb{1}_{\Omega(k-1)} + \alpha(\xi_u(k) - \xi_1(k))\mathbb{1}_{\Omega(k)}\Big).$$

This gives

$$(p_1(k+1) - p_u(k+1))\mathbb{1}_{\Omega(k)}$$
$$\geq \mathbb{1}_{\Omega(k)}\Big(p_1(0) - p_u(0) + \sum_{\ell=0}^{k} \alpha(\xi_u(\ell) - \xi_1(\ell))\mathbb{1}_{\Omega(\ell)}\Big) \quad (26)$$
$$\geq \Delta\mathbb{1}_{\Omega(k)} - |M_u(k)| - |M_1(k)|.$$

We want to prove by induction that $E(k) \subseteq \Omega(k+1)$ for all $k = 0, 1, \ldots$. For $k = 0$, this directly follows from Eq. (26), since $\Omega(0) = \Omega$ due to assumption Eq. (21). Assume the assertion holds for some $k = 0, 1, \ldots$. Hence, it holds $E(k+1) \subseteq E(k) \subseteq \Omega(k+1)$, such that for any $u \in \{2, \ldots, d\}$ it holds on $E(k+1)$ by Eq. (26)

$$(p_1(k+2) - p_u(k+2)) \geq \Delta - |M_u(k+1)| - |M_1(k+1)|$$
$$\geq \frac{\Delta}{2},$$

which proves the assertion. □

Having assembled the previous results, we are able to prove the linear convergence of STDP stated in Theorem 2.2. As Proposition A.3 already suggests the desired behaviour of $\mathbf{p}$ on $\Omega(k)$, the main

part of the proof is to show that $\Omega(k)$ is satisfied with large probability. For that we deploy Doob's submartingale inequality, which states that for a martingale $(X_n)_{n\in\mathbb{N}}$, any $p \geq 1$, and any $u > 0$,

$$\mathbb{P}\Big( \max_{i\in[n]} |X_i| \geq u \Big) \leq \frac{\mathbb{E}[|X_n|^p]}{u^p}. \tag{27}$$

This will be applied to derive lower bounds for the event $E(k)$ defined in Lemma A.4, which are also lower bounds for the probability of $\Omega(k+1)$ by the same Lemma. For the reader's convenience we restate Theorem 2.2 before giving its proof.

**Theorem 2.2.** *Given $\varepsilon \in (0,1)$, assume*

$$\Delta := p_1(0) - \max_{i=2,\dots,d} p_i(0) > 0 \quad and \quad 0 < \alpha \leq \frac{\Delta^2}{16Q^2}\Big((1-Q\alpha)^3 \wedge \frac{1}{256(1-p_1(0))}\Big(4\frac{\Delta}{d}+\Delta^2\Big)\varepsilon\Big).$$

*Then there exists an event $\Theta$ with probability $\geq 1 - \varepsilon/2$ such that*

$$\mathbb{E}\big[\|\mathbf{p}(k) - \mathbf{e}_1\|_1 \mathbb{1}_\Theta\big] \leq 2(1 - p_1(0)) \exp\Big( -\frac{\alpha}{16}\Big(4\frac{\Delta}{d} + \Delta^2\Big)k\Big), \quad for\ all\ k = 0, 1, \dots$$

*Consequently, given $\delta > 0$, it holds*

$$\mathbb{P}\Big(\|\mathbf{p}(k) - \mathbf{e}_1\|_1 \geq \delta\Big) \leq \varepsilon \quad for\ all \quad k \geq \frac{16d}{\alpha\Delta(4+d\Delta)} \log\Big(\frac{4(1-p_1(0))}{\varepsilon\delta}\Big).$$

*Proof of Theorem 2.2.* The recursive definition ensures that $\mathbf{p}(k)$ is $\mathcal{F}_{k-1}$-measurable. Thus, also $\Omega(k) \in \mathcal{F}_{k-1}$ for any $k = 0, 1, \dots$. One can check that $(M_i(k))_{k=0,1,\dots}$, defined in Eq. (25), forms a martingale for each $i \in [d]$. This allows us to apply Doob's submartingale inequality. To apply it with $p = 2$, we deduce the following bound on the second moment,

$\mathbb{E}[M_1(k)^2]$

$$= \alpha^2 \mathbb{E}\Big[\Big(\sum_{\ell=0}^k \xi_1(\ell)\mathbb{1}_{\Omega(\ell)}\Big)^2\Big]$$

$$= \alpha^2\Big(\sum_{\ell=0}^k \mathbb{E}[(\xi_1(\ell)\mathbb{1}_{\Omega(\ell)})^2] + \sum_{i,j=0, i\neq j}^k \mathbb{E}[\xi_1(i)\mathbb{1}_{\Omega(i)}\xi_1(j)\mathbb{1}_{\Omega(j)}]\Big)$$

$$= \alpha^2\Big(\sum_{\ell=0}^k \mathbb{E}[(\xi_1(\ell)\mathbb{1}_{\Omega(\ell)})^2] + \sum_{i,j=0, i\neq j}^k \mathbb{E}\Big[\xi_1(i\wedge j)\mathbb{1}_{\Omega(i\wedge j)}\mathbb{E}[\xi_1(i\vee j)\mathbb{1}_{\Omega(i\vee j)}|\mathcal{F}_{i\wedge j}]\Big]\Big)$$

$$= \alpha^2 \sum_{\ell=0}^k \mathbb{E}[(\xi_1(\ell)\mathbb{1}_{\Omega(\ell)})^2]$$

$$= \alpha^2 \sum_{\ell=0}^k \mathbb{E}\Big[(p_1(\ell))^2\Big(\mathbb{E}\Big[Y_1(\ell) - \sum_{j=1}^d p_j(\ell)Y_j(\ell)\Big|\mathcal{F}_{\ell-1}\Big] - \Big(Y_1(\ell) - \sum_{j=1}^d p_j(\ell)Y_j(\ell)\Big)\Big)^2 \mathbb{1}_{\Omega(\ell)}\Big]$$

$$\leq \alpha^2 \sum_{\ell=0}^k \mathbb{E}\Big[\mathbb{E}\Big[\Big(\mathbb{E}\Big[Y_1(\ell) - \sum_{j=1}^d p_j(\ell)Y_j(\ell)\Big|\mathcal{F}_{\ell-1}\Big] - \Big(Y_1(\ell) - \sum_{j=1}^d p_j(\ell)Y_j(\ell)\Big)\Big)^2\Big|\mathcal{F}_{\ell-1}\Big]\mathbb{1}_{\Omega(\ell)}\Big]$$

$$= \alpha^2 \sum_{\ell=0}^k \mathbb{E}\Big[\mathbb{E}\Big[\Big(Y_1(\ell) - \sum_{j=1}^d p_j(\ell)Y_j(\ell)\Big)^2\Big|\mathcal{F}_{\ell-1}\Big]\mathbb{1}_{\Omega(\ell)} - \mathbb{E}\Big[Y_1(\ell) - \sum_{j=1}^d p_j(\ell)Y_j(\ell)\Big|\mathcal{F}_{\ell-1}\Big]^2 \mathbb{1}_{\Omega(\ell)}\Big]$$

$$\leq \alpha^2 \sum_{\ell=0}^k \mathbb{E}\Big[\Big(Y_1(\ell) - \sum_{j=1}^d p_j(\ell)Y_j(\ell)\Big)^2 \mathbb{1}_{\Omega(\ell)}\Big]$$

$$= \alpha^2 \sum_{\ell=0}^k \mathbb{E}\Big[\Big((1 - p_1(\ell))Y_1(\ell) - \sum_{j=2}^d p_j(\ell)Y_j(\ell)\Big)^2 \mathbb{1}_{\Omega(\ell)}\Big]$$

$$\leq 2\alpha^2 \sum_{\ell=0}^{k} \mathbb{E}\Big[\big((1-p_1(\ell))Y_1(\ell)\big)^2 \mathbb{1}_{\Omega(\ell)} + \Big(\sum_{j=2}^{d} p_j(\ell)Y_j(\ell)\Big)^2 \mathbb{1}_{\Omega(\ell)}\Big]$$

$$\leq 2Q^2\alpha^2 \sum_{\ell=0}^{k} \mathbb{E}\Big[\big((1-p_1(\ell))^2 + \Big(\sum_{j=2}^{d} p_j(\ell)\Big)^2\big) \mathbb{1}_{\Omega(\ell)}\Big]$$

$$\leq 4Q^2\alpha^2 \sum_{\ell=0}^{k} \mathbb{E}\Big[(1-p_1(\ell)) \mathbb{1}_{\Omega(\ell)}\Big],$$

where we used that $\mathbb{E}[\xi_1(k_2) \,|\, \mathcal{F}_{k_1}] = 0$, for any $k_2 > k_1 = 0, 1, \ldots$, together with $|Y_1(k)| \leq Q$, the inequality $(a+b)^2 \leq 2(a^2+b^2)$ for $a, b \in \mathbb{R}$ and $1 - p_\ell^1 \leq 1$. Arguing similarly we obtain for $u \in \{2, \ldots, d\}$,

$$\mathbb{E}[(M_u(k)^2] = \alpha^2 \mathbb{E}\Big[\sum_{\ell=0}^{k} (\xi_u(\ell) \mathbb{1}_{\Omega(\ell)})^2\Big]$$

$$\leq \alpha^2 \sum_{\ell=0}^{k} \mathbb{E}\Big[(p_u(\ell))^2\Big(Y_u(\ell) - \sum_{j=1}^{d} p_j(\ell)Y_j(\ell)\Big)^2 \mathbb{1}_{\Omega(\ell)}\Big]$$

$$\leq 4Q^2\alpha^2 \sum_{\ell=0}^{k} \mathbb{E}\Big[p_u(\ell) \mathbb{1}_{\Omega(\ell)}\Big].$$

Hence, applying a union bound, Doob's submartingale inequality Eq. (27) with $p = 2$ gives for any $k = 0, 1, \ldots$

$$\mathbb{P}(E(k)) = 1 - \mathbb{P}\Big(\bigcup_{j=1}^{d} \max_{u \in [k]} |M_j(u)| \geq \Delta/4\Big)$$

$$\geq 1 - \sum_{j=1}^{d} \mathbb{P}\Big(\max_{u \in [k]} |M_j(u)| \geq \Delta/4\Big)$$

$$\geq 1 - 64Q^2 \frac{\alpha^2}{\Delta^2}\Big(\sum_{\ell=0}^{k} \mathbb{E}\Big[(1-p_1(\ell)) \mathbb{1}_{\Omega(\ell)}\Big] + \sum_{j=2}^{d}\sum_{\ell=0}^{k} \mathbb{E}\Big[p_j(\ell) \mathbb{1}_{\Omega(\ell)}\Big].\Big)$$

$$= 1 - 128Q^2 \frac{\alpha^2}{\Delta^2} \sum_{\ell=0}^{k} \mathbb{E}\Big[(1-p_1(\ell)) \mathbb{1}_{\Omega(\ell)}\Big].$$

Proposition A.3 gives for any $k = 0, 1, \ldots$ the bound

$$\mathbb{E}[(1-p_1(k+1)) \mathbb{1}_{\Omega(k+1)}] \leq \mathbb{E}[(1-p_1(k+1)) \mathbb{1}_{\Omega(k)}]$$

$$\leq \mathbb{E}\Big[\Big(1-\alpha\frac{\Delta}{4d}\big(1+\frac{\Delta}{2}(d-1)\big)\Big)\big(1-p_1(k)\big) \mathbb{1}_{\Omega(k)} + \alpha\xi_1(k) \mathbb{1}_{\Omega(k)}\Big]$$

$$= \Big(1-\alpha\frac{\Delta}{4d}\big(1+\frac{\Delta}{2}(d-1)\big)\Big)\mathbb{E}[(1-p_1(k)) \mathbb{1}_{\Omega(k)}],$$

which implies

$$\mathbb{E}\Big[\big(1-p_1(k)\big) \mathbb{1}_{\Omega(k)}\Big] \leq \big(1-p_1(0)\big)\Big(1-\alpha\frac{\Delta}{4d}\big(1+\frac{\Delta}{2}(d-1)\big)\Big)^k. \tag{28}$$

We set

$$\Theta := \bigcap_{k=0}^{\infty} \Omega(k) = \Big\{p_1(u) \geq \max_{i=2,\ldots,d} p_i(u) + \frac{\Delta}{2}, \forall u \in \mathbb{N}\Big\}.$$

The continuity of probability measures and Lemma A.4 then imply

$$\mathbb{P}(\Theta) = \lim_{k\to\infty} \mathbb{P}(\Omega(k))$$

$$\geq \lim_{k\to\infty} \mathbb{P}(E(k))$$

$$\geq 1 - 128 Q^2 \frac{\alpha^2}{\Delta^2} \sum_{\ell=0}^{\infty} \mathbb{E}\Big[ (1 - p_1(\ell)) \mathbb{1}_{\Omega(\ell)} \Big]$$

$$\geq 1 - 128 Q^2 (1 - p_1(0)) \frac{\alpha^2}{\Delta^2} \sum_{\ell=0}^{\infty} \Big( 1 - \alpha \frac{\Delta}{4d} \Big( 1 + \frac{\Delta}{2}(d-1) \Big) \Big)^{\ell}$$

$$= 1 - 1024 Q^2 (1 - p_1(0)) \frac{d\alpha}{\Delta^3 \big( 2 + \Delta(d-1) \big)}$$

$$\geq 1 - 2048 Q^2 (1 - p_1(0)) \frac{\alpha}{\Delta^3 \big( \frac{4}{d} + \Delta \big)}$$

$$\geq 1 - \frac{\varepsilon}{2},$$

where we used that we can assume $d \geq 2$ without loss of generality. Additionally, Eq. (28) and the elementary inequality $1 - x \leq \exp(-x)$, which is valid for any real number $x$, give

$$\mathbb{E}\big[ (1 - p_1(k)) \mathbb{1}_{\Theta} \big] \leq \mathbb{E}\big[ (1 - p_1(k)) \mathbb{1}_{\Omega(k)} \big]$$

$$\leq (1 - p_1(0)) \Big( 1 - \alpha \frac{\Delta}{4d} \Big( 1 + \frac{\Delta}{2}(d-1) \Big) \Big)^{k}$$

$$\leq (1 - p_1(0)) \exp \Big( - \alpha \frac{\Delta}{4d} \Big( 1 + \frac{\Delta}{2}(d-1) \Big) k \Big).$$

When $d = 1$, the right hand side of this inequality is 0. For $d \geq 2$, we can also use the bound $d - 1 \geq d/2$. Together with

$$\|\mathbf{p}(k) - \mathbf{e}_1\|_1 = 1 - p_1(k) + \sum_{i=2}^{d} p_i(k) = 2 \big( 1 - p_1(k) \big),$$

this concludes the proof of the first statement. For the proof of the second statement, we apply Markov's inequality to obtain

$$\mathbb{P}\Big( \|\mathbf{p}(k) - \mathbf{e}_1\|_1 \geq \delta \Big)$$

$$\leq \mathbb{P}(\Theta^{\mathbf{C}}) + \mathbb{P}\Big( \|\mathbf{p}(k) - \mathbf{e}_1\|_1 \mathbb{1}_{\Theta} \geq \delta \Big)$$

$$\leq \frac{\varepsilon}{2} + 2(1 - p_1(0)) \exp \Big( - \frac{\alpha}{16} \Big( 4 \frac{\Delta}{d} + \Delta^2 \Big) k \Big) \delta^{-1}.$$

Hence, if

$$k \geq \Big( \frac{\alpha}{16} \Big( 4 \frac{\Delta}{d} + \Delta^2 \Big) \Big)^{-1} \log \Big( \frac{4(1 - p_1(0))}{\varepsilon \delta} \Big)$$

$$= \frac{16d}{\alpha \Delta (4 + d\Delta)} \log \Big( \frac{4(1 - p_1(0))}{\varepsilon \delta} \Big),$$

then,

$$\mathbb{P}\Big( \|\mathbf{p}(k) - \mathbf{e}_1\|_1 \geq \delta \Big) \leq \varepsilon.$$

$\square$

**Lemma A.5.** *Given an initialization of the weights* $\mathbf{w}(0) = (w_1(0), \dots, w_d(0))^{\top}$ *consider two d-dimensional intensity vectors* $\boldsymbol{\lambda} = (\lambda_1, \dots, \lambda_d)^{\top}, \widetilde{\boldsymbol{\lambda}} = (\widetilde{\lambda}_1, \dots, \widetilde{\lambda}_d)^{\top}$ *with positive entries. Assume that* $\lambda_1 w_1(0) > \max_{i=2,\dots,d} \lambda_i w_i(0)$ *and* $\widetilde{\lambda}_d w_d(0) > \max_{i=1,\dots,d-1} \widetilde{\lambda}_i w_i(0)$. *Assume we run the learning rule Eq. (3) with intensity* $\boldsymbol{\lambda}$ *until time* $K^*$ *and then, for* $k > K^*$, *change the intensity to* $\widetilde{\boldsymbol{\lambda}}$ *and run the learning dynamic until time* $k \to \infty$. *If* $K^*$ *is small, in particular, if* $K^* = 0$, *the above convergence result can be applied to show that the dynamic converges to the corner* $\mathbf{e}_d$. *However, for any* $\varepsilon \in (0, 1)$ *and all sufficiently large* $K^*$ *(depending on* $\varepsilon$), *the dynamics will converge to* $\mathbf{e}_1$ *with probability* $1 - 2\varepsilon$.

This result shows that the dynamic can be primed at the beginning to end up in one regime. Despite the noise and the infinite amount of data, the dynamic is unable to escape this domain of attraction. From the proof, one can derive quantitative bounds for $K^*$.

*Proof.* Given $\varepsilon \in (0,1)$, choose $\delta \in (0,1)$ such that

$$\max_{i \neq 1} \frac{\widetilde{\lambda}_i \lambda_1 \delta}{\lambda_i \widetilde{\lambda}_1 (1-\delta)} < 1. \tag{29}$$

Let $\Delta = p_1(0) - \max_{i \neq 1} p_i(0)$. Given $\varepsilon, \delta, \Delta$ choose

$$K^* \geq \frac{16d}{\alpha \Delta (4 + d\Delta)} \log\left(\frac{4(1 - p_1(0))}{\varepsilon \delta}\right).$$

By Theorem 2.2, this guarantees that

$$\mathbb{P}\left(\|\mathbf{p}(K^*) - \mathbf{e}_1\|_1 \geq \delta\right) \leq \varepsilon.$$

On the event $\|\mathbf{p}(K^*) - \mathbf{e}_1\|_1 < \delta$, we have

$$1 - \frac{\lambda_1 w_1(K^*)}{\boldsymbol{\lambda}^\top \mathbf{w}(K^*)} < \delta, \quad \text{and} \quad \max_{i \neq 1} \frac{\lambda_i w_i(K^*)}{\boldsymbol{\lambda}^\top \mathbf{w}(K^*)} < \delta,$$

which can be combined into

$$\max_{i \neq 1} \lambda_i w_i(K^*) < \delta \boldsymbol{\lambda}^\top \mathbf{w}(K^*) < \frac{\delta}{1-\delta} \lambda_1 w_1(K^*).$$

Using the inequality Eq. (29), we obtain

$$\max_{i \neq 1} \widetilde{\lambda}_i w_i(K^*) < \frac{\widetilde{\lambda}_1 (1-\delta)}{\lambda_1 \delta} \frac{\delta}{1-\delta} \lambda_1 w_1(K^*) = \widetilde{\lambda}_1 w_1(K^*).$$

This means that restarting the learning rule Eq. (3) at time $K^*$ with intensities $\widetilde{\lambda}_1, \ldots, \widetilde{\lambda}_d$ and weights $w_1(K^*), \ldots, w_d(K^*)$, shows that $p_1(K^*) > \max_{i \neq 2} p_i(K^*)$. Applying Theorem 2.2 again shows convergence to $\mathbf{e}_1$ with probability $1 - 2\varepsilon$.

$\square$

## A.2 Proofs for Subsection 2.2

This section contains additional material on the gradient flow Eq. (9), as well as the proofs for Lemma A.6 and Theorem 2.4.

For $d = 2$, the gradient flow admits an explicit solution. In this case, $\mathbf{p}(t) = (p_1(t), p_2(t))^\top$. If $\mathbf{p}(0) = (1/2, 1/2)^\top$, then this is a stationary solution and $\mathbf{p}(t) = \mathbf{p}(0) = (1/2, 1/2)^\top$ for all $t \geq 0$. If $p_1(0) > 1/2$, then,

$$p_1(t) = \frac{1}{2} + \frac{1}{2\sqrt{Ce^{-t} + 1}}, \quad \text{with } C := \frac{1}{(2p_1(0) - 1)^2} - 1. \tag{30}$$

If $p_1(0) < 1/2$, then $p_2(t) = 1 - p_1(t) > 1/2$ follows the dynamic in Eq. (30). This formula immediately implies that $p_1(t)$ converges exponentially fast to 1.

*Proof of Formula Eq. (30).* Throughout the proof we set $p(t) := p_1(t)$ and do not use the previous notation $p_1(t), p_2(t)$ for the first and second probability. For $d = 2$, the gradient flow ODE Eq. (9) becomes

$$\frac{\mathrm{d}}{\mathrm{d}t} p(t) = p(t)^2 - p(t)\left(p(t)^2 + (1 - p(t))^2\right) = 3p(t)^2 - 2p(t)^3 - p(t). \tag{31}$$

Rewriting this in the variable $u(t) = 1 - 2p(t)$ gives the dynamic,

$$\frac{\mathrm{d}}{\mathrm{d}t} u(t) = -2\frac{\mathrm{d}}{\mathrm{d}t} p(t) = -6p(t)^2 + 4p(t)^3 + 2p(t) = \frac{1}{2}\left(1 - u(t)^2\right) u(t).$$

This is solved by $u(t) = -1/\sqrt{Ce^{-t} + 1}$ since

$$-\frac{\mathrm{d}}{\mathrm{d}t}\frac{1}{\sqrt{Ce^{-t} + 1}} = -\left(-\frac{1}{2}\right)\cdot\frac{-Ce^{-t}}{(Ce^{-t} + 1)^{3/2}} = \frac{1}{2}\left(1 - u(t)^2\right)u(t).$$

Thus, $p(t) = \frac{1}{2}(1 - u(t))$ solves Eq. (31). Finally, $C$ is determined by the initial condition $p(0) = \frac{1}{2}(1 - 1/\sqrt{C+1})$. $\qquad\square$

The following lemma summarises different properties of the gradient flow Eq. (9). In its statement, differentiable on $[0,1]$ means differentiable on $(0,1)$ and continuous on $[0,1]$.

**Lemma A.6.** *The gradient flow Eq. (9) exhibits the following properties.*

(a) *If $\phi\colon [0,1] \to \mathbb{R}$ is a convex and differentiable function, then $t \mapsto \sum_{i=1}^{d} \phi(p_i(t))$ is monotone increasing.*

(b) *Let $i, j \in [d]$ with $i \neq j$. If $p_i(0) > p_j(0)$, respectively $p_i(0) = p_j(0)$, then $p_i(t) > p_j(t)$, respectively $p_i(t) = p_j(t)$, for all $t \geq 0$. Moreover, if*

$$\Delta := p_1(0) - \max_{i=2,\dots,d} p_i(0) > 0,$$

*then*

$$p_1(t) \geq \max_{i=2,\dots,d} p_i(t) + \Delta \quad \text{for all } t \geq 0.$$

Lemma A.6 implies that the $q$-norm $t \mapsto |\mathbf{p}(t)|_q$ is monotonically increasing whenever $1 \leq q < \infty$. The result also implies that if instead $\phi$ is concave and differentiable, then $t \mapsto \sum_{i=1}^{d} \phi(p_i(t))$ is monotonically decreasing.

*Proof of Lemma A.6.*

(a) Since $\phi$ is convex, $\phi'$ is monotonically increasing. Thus, for a probability vector $\mathbf{q} = (q_1, \dots, q_d)$, we have

$$\sum_{i=1}^{d} q_i \phi'(q_i)\left(q_i - \|\mathbf{q}\|_2^2\right) = \sum_{i=1}^{d} q_i \phi'(q_i)\left(q_i \sum_{j=1}^{d} q_j - \sum_{j=1}^{d} q_j^2\right)$$

$$= \sum_{i,j=1}^{d} q_i q_j \phi'(q_i)\left(q_i - q_j\right)$$

$$= \sum_{1 \leq i < j \leq d} q_i q_j \left(\phi'(q_i) - \phi'(q_j)\right)\left(q_i - q_j\right)$$

$$\geq 0.$$

(If $\phi$ is strictly convex, then strict equality holds if and only if $\mathbf{q}$ is one of the stationary points described above.) Using this and the gradient flow formula

$$\frac{\mathrm{d}}{\mathrm{d}t}\sum_{i=1}^{d} \phi(p_i(t)) = \sum_{i=1}^{d} \phi'(p_i(t))\frac{\mathrm{d}}{\mathrm{d}t}p_i(t) = \sum_{i=1}^{d} p_i(t)\phi'(p_i(t))\left(p_i(t) - \|\mathbf{p}(t)\|_2^2\right) \geq 0,$$

proving the result.

(b) By definition it holds for $i, j \in [d]$

$$\frac{\mathrm{d}}{\mathrm{d}t}\left(p_i(t) - p_j(t)\right) = p_i(t)\left(p_i(t) - \|\mathbf{p}(t)\|^2\right) - p_j(t)\left(p_j(t) - \|\mathbf{p}(t)\|^2\right)$$

$$= \left(p_i(t) - p_j(t)\right)\left(p_i(t) + p_j(t) - \|\mathbf{p}(t)\|^2\right). \tag{32}$$

From this we deduce that

$$p_i(t) - p_j(t) = (p_i(0) - p_j(0))\exp\left(\int_0^t p_i(s) + p_j(s) - \|\mathbf{p}(s)\|^2\,\mathrm{d}s\right),$$

which concludes the proof of the first statement since the exponential function is strictly positive.

To prove the second statement, we have

$$p_1(t) = p_1(0) + \int_0^t p_1(s)\big(p_1(s) - \|\mathbf{p}(s)\|^2\big) \, ds \geq p_1(0) - t,$$

and similarly, for any $i \in \{2, \dots, d\}$,

$$p_i(t) \leq p_i(0) + t \leq p_1(0) + t - \Delta.$$

Hence, whenever $t \in [0, \Delta/2]$, we find

$$p_1(t) \geq \max_{i=2,\dots,d} p_i(t),$$

which also implies

$$p_1(t) \geq p_1(t)^2 + \max_{i=2,\dots,d} p_i(t)\big(1 - p_1(t)\big) \geq \|\mathbf{p}(t)\|^2,$$

for all $t \in [0, \Delta/2]$. Therefore for any $t \in [0, \Delta/2], i \in [d]$ it holds

$$\frac{d}{dt}\big(p_1(t) - p_i(t)\big) = \big(p_1(t) - p_i(t)\big)\big(p_1(t) + p_i(t) - \|\mathbf{p}(t)\|^2\big) \geq 0,$$

which implies for any $i \in \{2, \dots, d\}$ and $t \in [0, \Delta/2]$,

$$p_1(t) - p_i(t) \geq p_1(0) - p_i(0) \geq \Delta,$$

Applying this argument iteratively concludes the proof.

$\square$

For the reader's convenience we restate Theorem 2.4 before giving its proof.

**Theorem 2.4.** *Assume*

$$p_1(0) \geq \max_{i=2,\dots,d} p_i(0) + \Delta,$$

*for some $\Delta > 0$. Then*

$$\|\mathbf{e}_1 - \mathbf{p}(t)\|_1 \leq 2(1 - p_1(0)) \exp\left(-\frac{\Delta}{d}(1 + (d-1)\Delta)t\right),$$

*that is linear convergence of $\mathbf{p}(t) \to \mathbf{e}_1$ as $t \to \infty$.*

*Proof of Theorem 2.4.* Arguing as in Eq. (24) and Eq. (23), Lemma A.6 Eq. (b) implies

$$\frac{d}{dt}(1 - p_1(t)) = -p_1(t)(p_1(t) - \|\mathbf{p}(t)\|^2)$$
$$\leq -\mu(1 - p_1(t)),$$

where

$$\mu := \frac{\Delta}{d}((d-1)\Delta + 1).$$

Grönwall's inequality entails

$$1 - p_1(t) \leq (1 - p_1(0)) \exp(-\mu t),$$

which gives

$$\|\mathbf{p}(t) - \mathbf{e}_1\|_1 = 1 - p_1(t) + \sum_{i=2}^d p_i(t) = 2(1 - p_1(t)) \leq 2(1 - p_1(0)) \exp(-\mu t).$$

$\square$

## A.3 Proofs for Subsection 2.3

In this subsection, we heuristically derive the expression for the probabilities

$$\frac{\lambda_j w_j(\mathsf{t}_k)}{\sum_{\ell=1}^d \lambda_\ell w_\ell(\mathsf{t}_k)}, \qquad j = 1, \dots, d, \tag{33}$$

in Eq. (12). To this end, we assume that the weights are small compared to the threshold $S$, that the weights are only updated at the postsynaptic spike times, and that $\sum_{\ell=1}^d \lambda_\ell w_\ell(\mathsf{t}_k) \gg S$. For convenience, we write $w_\ell$ for $w_\ell(\mathsf{t}_k)$ and all $\ell \in [d]$. The constraint $\sum_{\ell=1}^d \lambda_\ell w_\ell \gg S$ guarantees that after the postsynaptic spike time $\mathsf{t}_k$, the membrane potential $Y_t$ will again reach $S$ and thus emit another spike at time $\mathsf{t}_{k+1}$.

Taking the expectation of the membrane potential $Y_t = \sum_{j=1}^d \sum_{\tau \in \mathcal{T}_j \cap (\mathsf{t}_k, t]} w_j e^{-(t-\tau)}$ with respect to all except the $j^{\text{th}}$ spike-train, gives

$$\begin{aligned}
Z_t &:= \sum_{\tau \in \mathcal{T}_j \cap (\mathsf{t}_k, t]} w_j e^{-(t-\tau)} + \sum_{\ell \neq j} w_\ell \lambda_\ell \int_{\mathsf{t}_k}^t e^{-(t-s)} \, \mathrm{d}s \\
&= \sum_{\tau \in \mathcal{T}_j \cap (\mathsf{t}_k, t]} w_j e^{-(t-\tau)} + \sum_{\ell \neq j} w_\ell \lambda_\ell \left(1 - e^{-(t-\mathsf{t}_k)}\right),
\end{aligned}$$

for all $\mathsf{t}_k \leq t < \mathsf{t}_{k+1}$.

Introduce $\mathsf{t}^* := \inf\{t \geq \mathsf{t}_k : Z_t \geq S - w_j\}$ and write $\mathsf{t}^+$ for the first time after $\mathsf{t}^*$ where

$$t \mapsto \underbrace{Z_{\mathsf{t}^*} + e^{-(\mathsf{t}^* - \mathsf{t}_k)} \sum_{\ell \neq j}^d w_\ell \lambda_\ell \left(1 - e^{-(t-\mathsf{t}^*)}\right)}_{=: \, V_t}$$

reaches the threshold $S$. If there are sufficiently many neurons, the probability that the $j^{\text{th}}$ presynaptic neuron spikes at time $\mathsf{t}^*$ is small and will be neglected. We have $Z_{\mathsf{t}^*} = S - w_j$ such that $V_{\mathsf{t}^+} = w_j$. Approximating $1 - e^{-(\mathsf{t}^+ - \mathsf{t}^*)} \approx \mathsf{t}^+ - \mathsf{t}^*$ gives

$$\mathsf{t}^+ - \mathsf{t}^* \approx e^{\mathsf{t}^* - \mathsf{t}_k} \frac{w_j}{\sum_{\ell \neq j} w_\ell \lambda_\ell}.$$

The $j^{\text{th}}$ presynaptic neuron causes the next postsynaptic spike if and only if it spikes in the interval $(\mathsf{t}^*, \mathsf{t}^+)$. The spike times of the $j^{\text{th}}$ presynaptic neuron are generated from a Poisson process with intensity $\lambda_j$. Thus, if $U \sim \mathrm{Poisson}(\lambda_j(\mathsf{t}^+ - \mathsf{t}^*))$, the probability that the $j^{\text{th}}$ presynaptic neuron spikes in $(\mathsf{t}^*, \mathsf{t}^+)$ is given by

$$\mathbb{P}\big(U \neq 0\big) = 1 - \mathbb{P}\big(U = 0\big) = 1 - \exp\big(-\lambda_j(\mathsf{t}^+ - \mathsf{t}^*)\big) \approx \lambda_j(\mathsf{t}^+ - \mathsf{t}^*) \approx e^{\mathsf{t}^* - \mathsf{t}_k} \frac{w_j \lambda_j}{\sum_{\ell \neq j} w_\ell \lambda_\ell}.$$

We can moreover approximate the denominator on the right hand side by the full sum $\sum_{\ell=1}^d w_\ell \lambda_\ell$. Since the probabilities add up to one, we must have $e^{\mathsf{t}^* - \mathsf{t}_k} \approx 1$. This shows that the probability of the $j^{\text{th}}$ presynaptic neuron triggering the first postsynaptic spike after $\mathsf{t}_k$ is approximately given by Eq. (33).

**Lemma A.7.** *Consider the setting outlined in Subsection 2.3. If at some time point $\mathsf{t} > 0$, all weights are the same, then the probability that the $j^{th}$ neuron triggers the next postsynaptic spike after $\mathsf{t}$ is given by*

$$\frac{\lambda_j}{\sum_{\ell=1}^d \lambda_\ell}.$$

*Proof.* Since all weights are the same, we can denote their value by $w$. The $j^{\text{th}}$ neuron causes a postsynaptic spike if and only if it is the first one to spike after the postsynaptic membrane potential $Y_t$ has reached a level $\geq S - w$. As $\mathsf{t}^* = \inf\{t \geq \mathsf{t} : Y_t \geq S - w\}$ is a jump time and a stopping

time, we can restart the process at $\mathfrak{t}^*$. As the increments of Poisson processes are independent, and the time between the jumps is exponentially distributed with parameters $\lambda_j$, the probability that the $j^{\text{th}}$ neuron causes the next presynaptic spike is given by

$$\mathbb{P}\big(X_j = \min(X_1, \ldots, X_d)\big),$$

where $(X_i)_{i \in [d]}$ are independent random variables satisfying $X_i \sim \text{Exp}(\lambda_i)$. If $U \sim \text{Exp}(\lambda)$ and $V \sim \text{Exp}(\lambda')$ are independent, then, $U \wedge V \sim \text{Exp}(\lambda + \lambda')$ and $\mathbb{P}(U \leq V) = \lambda/(\lambda + \lambda')$. Thus, $\min_{i \neq j} X_i \sim \text{Exp}(\sum_{i \neq j} \lambda_i)$, and

$$\mathbb{P}\big(X_j = \min(X_1, \ldots, X_d)\big) = \mathbb{P}\Big(X_j \leq \min_{i \neq j} X_i\Big) = \frac{\lambda_j}{\sum_{\ell=1}^d \lambda_\ell}.$$

$\square$

The above lemma and the previous discussion give a motivation for the form of the probabilities in settings, where the weights are small compared to the threshold $S$ or equal. We now give another heuristic, motivating our modelling choice. Let $Y$ be the postsynaptic membrane potential. Then, in expectation $Y$ grows linearly with slope $\boldsymbol{\lambda}^\top \mathbf{w}$. Furthermore, input $i$ causes a postsynaptic spike if, and only if, it spikes at a time at which $Y \geq S - w_i$, where $S > 0$ is the threshold level. As $Y$'s growth is approximately linear, the amount of time in which $Y \geq S - w_i$, holds is approximately equal to $w_i/\boldsymbol{\lambda}^\top \mathbf{w}$. Now the probability that input $i$ jumps in an interval of length $w_i/\boldsymbol{\lambda}^\top \mathbf{w}$ is given by $1 - \exp(-\lambda_i w_i/\boldsymbol{\lambda}^\top \mathbf{w}) \approx \lambda_i w_i/\boldsymbol{\lambda}^\top \mathbf{w}$, which is exactly our modelling choice in the independent setting.

### A.4 On the connection to entropic mirror descent

An alternative approach to connecting our proposed learning rule Eq. (3) for the probabilities $\mathbf{p}$ and the entropic mirror descent in discrete-time is as follows. The entropic mirror descent step Eq. (14) with Kullback–Leibler divergence and potential $f$ can be solved explicitly and yields

$$p_i(k+1) = \frac{p_i(k) \exp(-\alpha(\nabla f(\mathbf{p}(k)))_i)}{\sum_{j=1}^d p_j(k) \exp(-\alpha(\nabla f(\mathbf{p}(k)))_j)}, \quad i \in [d], k = 0, 1, \ldots, \tag{34}$$

see Section 5 of Beck and Teboulle [5] for details. With $f(\mathbf{p}) = \widetilde{L}(\mathbf{p}) = \|\mathbf{p}\|^2/2$ and the first order approximation $\exp(x) \approx 1 + x$ for small $x$ we deduce

$$\begin{aligned}
p_i(k+1) &= \frac{p_i(k) \exp(-\alpha(\nabla \widetilde{L}(\mathbf{p}(k)))_i)}{\sum_{j=1}^d p_j(k) \exp(-\alpha(\nabla \widetilde{L}(\mathbf{p}(k)))_j)} = \frac{p_i(k) \exp(\alpha p_i(k))}{\sum_{j=1}^d p_j(k) \exp(\alpha p_j(k))} \\
&\approx \frac{p_i(k)(1 + \alpha p_i(k))}{\sum_{j=1}^d p_j(k)(1 + \alpha p_j(k))}
\end{aligned}$$

for any $i = 1, \ldots, d$ and $k = 0, 1, \ldots$. As our proposed learning rule Eq. (3) is a noisy version of the last line, it is naturally connected to noisy entropic gradient descent.

### A.5 Theoretical results for the alignment of multiple read-out neurons

For a weight vector $\mathbf{w} \in \mathbb{R}^d$ with nonnegative entries let $i^* := \text{argmax}_{i=1,\ldots,d} \mathbf{w}^\top \mathbf{e}_i = \text{argmax}_{i=1,\ldots,d} w_i$ and define the cosine-projection

$$\mathcal{P}\mathbf{w} := \|\mathbf{w}\|\mathbf{e}_{i^*}.$$

Assume that $\lambda_1 > \cdots > \lambda_d$ and consider first learning the first weight vector $\mathbf{w}_1$ using the learning rule in Eq. (1) while the remaining weight vectors $\mathbf{w}_2, \ldots, \mathbf{w}_d$ are fixed. Theorem 2.2 implies that after $K$ iterations, we have $\mathbf{w}_1 \approx \mathbf{e}_1$. By projecting onto $\mathbf{w}_1^* := \mathcal{P}\mathbf{w}_1(K)$ and setting $\mathbf{w}_2(0) \leftarrow \mathbf{w}_2(0) - \mathbf{w}_1^* \mathbf{w}_2(0)^\top \mathbf{w}_1^*/\|\mathbf{w}_1^*\|$, we ensure that $[\mathbf{p}_2(0)]_2 = \max_{i=1,\ldots,d}[\mathbf{p}_2(0)]_i$ and Theorem 2.2 applies and yields $\mathbf{p}_2(K) \approx \mathbf{e}_2$. Proceeding successively, we arrive at Algorithm 2.

**Theorem A.8.** *Consider Algorithm 2. Assume that $\lambda_1 > \cdots > \lambda_d$ and the minimal gap $\Delta = \min_{i=1,\ldots,d-1}([\mathbf{p}_i(0)]_i - \max_{j>i}[\mathbf{p}_i(0)]_j) > 0$ is positive. Then we have*

$$\mathbb{P}(\mathbf{P}^* \neq \mathbb{I}) \to 0$$

*as $K \to \infty$. More precisely, let $\delta < \kappa/(1+\kappa)$ for $\kappa = \min_{i=1,\ldots,d} \lambda_i / \max_{i=1,\ldots,d} \lambda_i$ and $\varepsilon > 0$. Then*

$$\mathbb{P}(\mathbf{P}^* = \mathbb{I}) \geq (1-\varepsilon)^d \quad \text{for } K \geq \frac{16d}{\alpha\Delta(4+d\Delta)} \log\left(\frac{4}{\varepsilon\delta}\right).$$

---

**Algorithm 2:** Sequential alignment of multiple output neurons

---

**Input:** $K \in \mathbb{N}$: number of iterations for each learning period, $\mathbf{W}(0) \in \mathbb{R}^{d\times d}$: weight initialisation.

1 **for** $j = 1, \ldots, d$ **do**
2     **if** $j \geq 2$ **then**
3         $\mathbf{w}_j(0) \leftarrow \mathbf{w}_j(0) - \sum_{i=1}^{j-1} \frac{\mathbf{w}_j(0)^\top \mathbf{w}_i^*}{\|\mathbf{w}_i^*\|^2}\mathbf{w}_i^*$;
4     **end**
5     **for** $k = 0, 1, \ldots, d$ **do**
6         Receive $\mathbf{B}_j(k) \sim M(1, \mathbf{p}_j(k))$ with $\mathbf{p}_j(k) \leftarrow \boldsymbol{\lambda} \odot \mathbf{w}_j(k)/\boldsymbol{\lambda}^\top \mathbf{w}_j(k)$ and $\mathbf{Z}_j(k) \sim \text{Unif}([-1,1]^d)$ from spike trains;
7         Update
$$\mathbf{w}_j(k+1) \leftarrow \alpha\mathbf{w}_j(k) \odot (\mathbf{B}_j(k) + \mathbf{Z}_j(k));$$
        Set $\mathbf{w}_j^* := \mathcal{P}\mathbf{w}_j(K)$ to obtain $\mathbf{p}_j^* = \boldsymbol{\lambda}^\top \mathbf{w}_j^*$.
8     **end**
9 **end**

**Output:** The weight evolution $\mathbf{W}(k) = [\mathbf{w}_1(k) \cdots \mathbf{w}_d(k)]$, $k = 0, \ldots, K$, probability evolution $\mathbf{P}(k) = [\mathbf{p}_1(k) \cdots \mathbf{p}_d(k)]$, $k = 1, \ldots, K$ and projections $\mathbf{P}^* = [\mathbf{p}_1^* \cdots \mathbf{p}_d^*]$.

---

Before proving Theorem A.8, we start with an auxiliary result on the order of the weights when the probability vector is close to a standard unit vector.

**Lemma A.9.** *Assume that $0 < \lambda_{\min} = \min_{i=1,\ldots,d} \lambda_i \leq \lambda_{\max} = \max_{i=1,\ldots,d} \lambda_i < \infty$ and let $\kappa = \lambda_{\min}/\lambda_{\max}$. Consider a weight vector $\mathbf{w} \in \mathbb{R}^d$ with corresponding probability vector $\mathbf{p} = \mathbf{w} \odot \boldsymbol{\lambda}/\mathbf{w}^\top \boldsymbol{\lambda}$ and let $0 < \delta < 1$. Then the condition $1 - p_1 < \delta$ implies that $\max_{i=2,\ldots,d} w_i \leq w_1\kappa^{-1}\delta/(1-\delta)$.*

*Proof.* Since $1 - p_1 \leq \delta$ we know that $\sum_{i=2}^d \lambda_i w_i \leq \frac{\delta}{1-\delta}\lambda_1 w_1$. By bounding the $\lambda_i$ from above and below we find

$$\lambda_{\min}\sum_{i=2}^d w_i \leq \frac{\delta}{1-\delta}\lambda_{\max}w_1,$$

such that

$$\max_{i=2,\ldots,d} w_i \leq \frac{\delta}{1-\delta}\kappa^{-1}w_1.$$

$\square$

*Proof of Theorem A.8.* By Theorem 2.2 we know that $\mathbb{P}(\|\mathbf{p}_1(K) - \mathbf{e}_1\|_1 \leq \delta) \geq 1 - \varepsilon$ for our choice of $K$. Note that the projection picks out the direction $\arg\max_{i=1,\ldots,d} \mathbf{w}_1(K)^\top \mathbf{e}_i = \arg\max_{i=1,\ldots,d}[\mathbf{w}_1(K)]_i$. Consequently, Lemma A.9 and $\delta < \kappa/(1-\kappa)$ imply that $\mathbf{w}_1(K)$ projects in the direction of $\mathbf{e}_1$ with probability at least $1 - \varepsilon$. We find that $\mathbb{P}(\mathbf{p}_1^* \neq \mathbf{e}_1) \leq 1 - \varepsilon$. By the adjustment of the weight vector $\mathbf{w}_2(0)$ we remove its first component, such that $[\mathbf{p}_2(0)]_1 = 0$ and $[\mathbf{p}_2(0)]_2 = \max_{j=1,\ldots,d}[\mathbf{p}_2(0)]_i$. By the assumption on $[\mathbf{p}_2(0)]_2 - \max_{j=3,\ldots,d}[\mathbf{p}_2(0)]_j \geq \Delta > 0$ we find that $\mathbb{P}(\|\mathbf{p}_2(K) - \mathbf{e}_2\|_1 \leq \delta) \geq 1 - \varepsilon$. By iteration and independence of the training windows we conclude the proof. $\square$

## A.6 Additional material for the extension to time-inhomogeneous intensities

**Derivation of Eq. (18):** Eq. (17) yields the deterministic update scheme

$$\mathbf{p}(k+1) = \widetilde{\mathbf{p}}(k) \odot \Big( \mathbf{1} + \alpha\big(\mathbf{p}(k) - \widetilde{\mathbf{p}}(k)^\top \mathbf{p}(k)\mathbf{1}\big) \Big) + \mathcal{O}(\alpha^2). \tag{35}$$

Let $T > 0$ be the time horizon and assume that the time-inhomogeneous intensities $\boldsymbol{\lambda}_\alpha$ change on the same scale as the weights and are given by $\boldsymbol{\lambda}_\alpha(t) = \boldsymbol{\lambda}(t\alpha)$, $t \in [0, T/\alpha]$, where $\boldsymbol{\lambda} \in C_b^2([0, T])$ is a universal, twice differentiable function with bounded derivatives and $\boldsymbol{\lambda}(t) \geq \lambda_{\min} > 0$, $t \in [0, T]$, componentwise. For vectors $\mathbf{w}, \boldsymbol{\lambda}, \boldsymbol{\lambda}'$ that are of the same length, and $\Delta := \boldsymbol{\lambda}' - \boldsymbol{\lambda}$, we have

$$\frac{\boldsymbol{\lambda}' \odot \mathbf{w}}{\boldsymbol{\lambda}'^\top \mathbf{w}} = \frac{\Delta \odot \mathbf{w} + \boldsymbol{\lambda} \odot \mathbf{w}}{\Delta^\top \mathbf{w} + \boldsymbol{\lambda}^\top \mathbf{w}} = \frac{\Delta \odot \mathbf{w}}{\Delta^\top \mathbf{w} + \boldsymbol{\lambda}^\top \mathbf{w}} + \frac{\boldsymbol{\lambda} \odot \mathbf{w}}{\boldsymbol{\lambda}^\top \mathbf{w}}\Big(1 - \frac{\Delta^\top \mathbf{w}}{\Delta^\top \mathbf{w} + \boldsymbol{\lambda}^\top \mathbf{w}}\Big).$$

Applying this with $\boldsymbol{\lambda}' = \boldsymbol{\lambda}_\alpha(k+1), \boldsymbol{\lambda} = \boldsymbol{\lambda}_\alpha(k), \mathbf{w} = \mathbf{w}(k)$, yields

$$\widetilde{\mathbf{p}}(k) = \frac{\boldsymbol{\lambda}_\alpha(k+1) \odot \mathbf{w}(k)}{\boldsymbol{\lambda}_\alpha(k+1)^\top \mathbf{w}(k)}$$

$$= \mathbf{p}(k) + \frac{\Delta\boldsymbol{\lambda}_\alpha(k) \odot \mathbf{w}(k)}{\Delta\boldsymbol{\lambda}_\alpha(k)^\top \mathbf{w}(k) + \boldsymbol{\lambda}_\alpha(k)^\top \mathbf{w}(k)} - \mathbf{p}(k)\frac{\Delta\boldsymbol{\lambda}_\alpha(k)^\top \mathbf{w}(k)}{\Delta\boldsymbol{\lambda}_\alpha(k)^\top \mathbf{w}(k) + \boldsymbol{\lambda}_\alpha(k)^\top \mathbf{w}(k)}.$$

Our assumptions on $\boldsymbol{\lambda}_\alpha$ imply that $\widetilde{\mathbf{p}}(k) - \mathbf{p}(k) \lesssim \|\Delta\boldsymbol{\lambda}_\alpha(k)\| \lesssim \alpha$ for all $k = 0, 1, \ldots, \lfloor T/\alpha\rfloor$. This gives $\widetilde{\mathbf{p}}(k) \odot \mathbf{p}(k) - \widetilde{\mathbf{p}}(k)\widetilde{\mathbf{p}}(k)^\top \mathbf{p}(k) = \mathbf{p}(k) \odot \mathbf{p}(k) - \mathbf{p}(k)\mathbf{p}(k)^\top \mathbf{p}(k) + \mathcal{O}(\alpha)$. In combination with Eq. (35) we find

$$\begin{aligned}
\frac{\mathbf{p}(k+1) - \mathbf{p}(k)}{\alpha} &= \frac{\Delta\boldsymbol{\lambda}_\alpha(k)}{\alpha} \odot \frac{\mathbf{w}(k)}{\Delta\boldsymbol{\lambda}_\alpha(k)^\top \mathbf{w}(k) + \boldsymbol{\lambda}_\alpha(k)^\top \mathbf{w}(k)} \\
&\quad - \frac{\mathbf{p}(k)}{\Delta\boldsymbol{\lambda}_\alpha(k)^\top \mathbf{w}(k) + \boldsymbol{\lambda}_\alpha(k)^\top \mathbf{w}(k)}\frac{\Delta\boldsymbol{\lambda}_\alpha(k)^\top}{\alpha}\mathbf{w}(k) \\
&\quad + \widetilde{\mathbf{p}}(k) \odot \mathbf{p}(k) - \widetilde{\mathbf{p}}(k)\widetilde{\mathbf{p}}(k)^\top \mathbf{p}(k) \\
&= \frac{\Delta\boldsymbol{\lambda}_\alpha(k)}{\alpha} \odot \frac{\mathbf{w}(k)}{\mathcal{O}(\alpha) + \boldsymbol{\lambda}_\alpha(k)^\top \mathbf{w}(k)} \\
&\quad - \frac{\mathbf{p}(k)}{\mathcal{O}(\alpha) + \boldsymbol{\lambda}_\alpha(k)^\top \mathbf{w}(k)}\frac{\Delta\boldsymbol{\lambda}_\alpha(k)^\top}{\alpha}\mathbf{w}(k) \\
&\quad + \mathbf{p}(k) \odot \mathbf{p}(k) - \mathbf{p}(k)\mathbf{p}(k)^\top \mathbf{p}(k) + \mathcal{O}(\alpha).
\end{aligned} \tag{36}$$

Sending $\alpha \to 0$ and using that $\Delta\boldsymbol{\lambda}_\alpha(t/\alpha)/\alpha \to \frac{\mathrm{d}}{\mathrm{d}t}\boldsymbol{\lambda}(t)$ as $\alpha \to 0$ we recognize Eq. (36) as an Euler-type scheme for the ODE

$$\begin{aligned}
\frac{\mathrm{d}}{\mathrm{d}t}\mathbf{p}(t) &= \frac{\frac{\mathrm{d}}{\mathrm{d}t}[\boldsymbol{\lambda}(t)] \odot \mathbf{w}(t)}{\boldsymbol{\lambda}^\top(t)\mathbf{w}(t)} - \mathbf{p}(t)\frac{\frac{\mathrm{d}}{\mathrm{d}t}[\boldsymbol{\lambda}(t)]^\top \mathbf{w}(t)}{\boldsymbol{\lambda}^\top(t)\mathbf{w}(t)} + \mathbf{p}(t) \odot \big(\mathbf{p}(t) - \|\mathbf{p}(t)\|^2\mathbf{1}\big) \\
&= \mathbf{p}(t) \odot \Big(\frac{\mathrm{d}}{\mathrm{d}t}\log\big(\boldsymbol{\lambda}(t)\big) - \mathbf{p}^\top(t)\frac{\mathrm{d}}{\mathrm{d}t}\log\big(\boldsymbol{\lambda}(t)\big)\mathbf{1}\Big) + \mathbf{p}(t) \odot \big(\mathbf{p}(t) - \|\mathbf{p}(t)\|^2\mathbf{1}\big) \\
&= \mathbf{p}(t) \odot \Big(\frac{\mathrm{d}}{\mathrm{d}t}\log\big(\boldsymbol{\lambda}(t)\big) + \mathbf{p}(t) - \mathbf{p}(t)^\top\Big(\frac{\mathrm{d}}{\mathrm{d}t}\log\big(\boldsymbol{\lambda}(t)\big) + \mathbf{p}(t)\Big)\mathbf{1}\Big), \quad t \in [0, T],
\end{aligned}$$

where the logarithm is taken componentwise.

## A.7 Analysis of the correlated model

In the following we always assume that the off-diagonal entries of $\boldsymbol{\Gamma}$ are strictly smaller than 1, that is, $\boldsymbol{\Gamma}_{i,j} < 1$ for all $i \neq j = 1, \ldots, d$. This assumption is reasonable since a perfect correlation between input $i$ and input $j$ corresponds to one single input neuron with Poisson process intensity $\lambda_i + \lambda_j$. Under this assumption it is easy to see that the quadratic form associated to $\boldsymbol{\Gamma}$ is maximised on the probability simplex by the basis vectors $\mathbf{e}_1, \ldots, \mathbf{e}_d$. Thus, the natural question to investigate is the same as in the original model: To which basis vector does the model converge? In the following section we investigate this question and show results for the weakly dependent case.

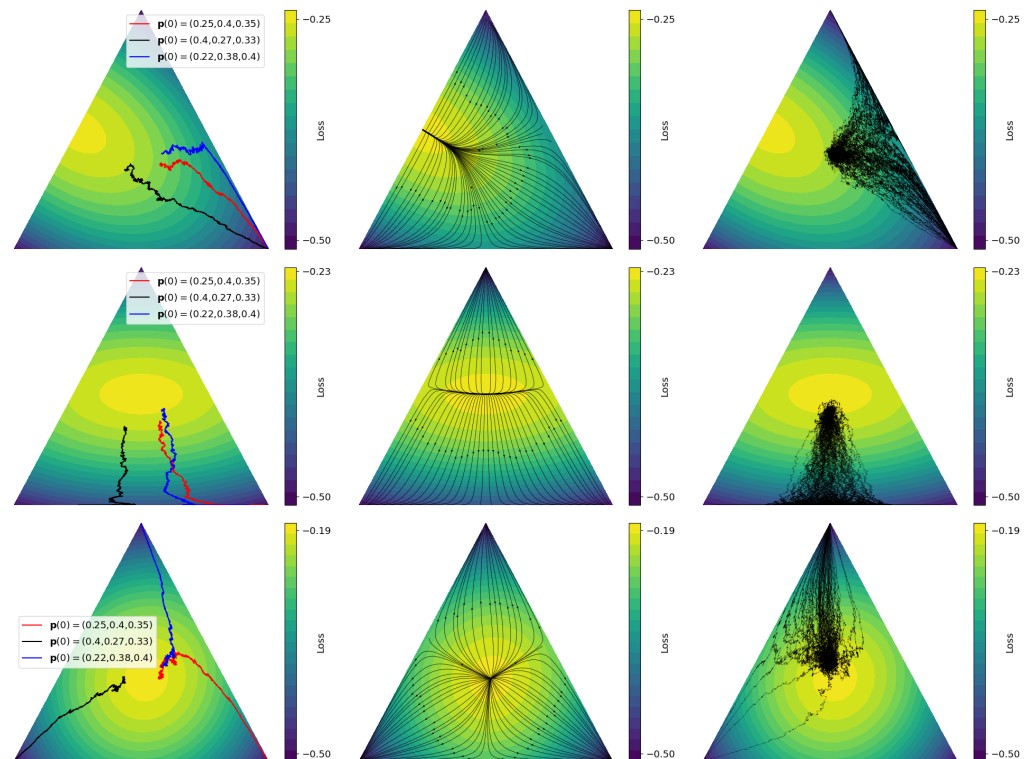

Figure 6: Correlated inputs with

$$\boldsymbol{\Gamma} = \begin{pmatrix} 1 & 1/2 & 0 \\ 1/2 & 1 & 1/2 \\ 0 & 1/2 & 1 \end{pmatrix} \text{(top)}, \boldsymbol{\Gamma} = \begin{pmatrix} 1 & 3/4 & 0 \\ 3/4 & 1 & 0 \\ 0 & 0 & 1 \end{pmatrix} \text{(middle)}, \begin{pmatrix} 1 & 1/10 & 1/10 \\ 1/10 & 1 & 0 \\ 1/10 & 0 & 1 \end{pmatrix} \text{(bottom)}.$$

Contour plot of the Shahshahani loss function $L(\mathbf{p}) = -\frac{1}{2}\mathbf{p}^\top \boldsymbol{\Gamma}\mathbf{p}$ on the probability simplex $\mathfrak{P}$ for $d = 3$ with different overlays. Left: Three sample trajectories of Eq. (3) with different initial configurations $\mathbf{p}(0)$. Middle: Stream plot of the gradient field given by Eq. (7). Right: 100 sample trajectories of Eq. (3) with $\mathbf{p}(0) = (0.3, 0.3, 0.4)^\top$. All trajectories are simulated with 2000 iteration steps, learning rate $\alpha = 0.01$ and $\mathbf{Z}(k) \sim \text{Unif}([-1, 1]^d)$.

The analysis of the correlated version of the model follows the same steps as in the independent case. For this we assume that all random variables are defined on a filtered probability space $(\Omega, \mathcal{F}, \mathbb{P})$ and denote by $\mathcal{F}_n, n = 1, 2, \ldots$ the natural filtration of $(\mathbf{B}(n), \mathbf{Z}(n), \mathbf{C}(n))_{n \in \mathbb{N}}$, and set $\mathcal{F}_{-1} = \{\emptyset, \Omega\}$. The proof follows exactly the same steps as the proof of Theorem 2.2. We start with the following result, which bounds the error of the Taylor approximation of the random dynamics.

**Lemma A.10.** *For $i \in [d]$ and $k = 0, 1, \ldots$ define*

$$\xi_i(k) := p_i(k)\left(\mathbb{E}\left[Y_i(k) - \sum_{j=1}^d p_j(k)Y_j(k)\Big|\mathcal{F}_{k-1}\right] - \left(Y_i(k) - \sum_{j=1}^d p_j(k)Y_j(k)\right)\right), \quad (37)$$

*and assume $\alpha < 1/Q$. Then for any $i \in [d]$ and $k = 0, 1, \ldots$, there exists a random variable $\theta_i(k)$, satisfying*

$$|\theta_i(k)| \le \alpha^2 \frac{2Q^2}{(1 - Q\alpha)^3} p_i(k)\big(1 - p_i(k)\big), \qquad \text{almost surely,}$$

*such that*

$$p_i(k+1) = p_i(k) + \alpha p_i(k)\big(\boldsymbol{\Gamma}_{i,.}\mathbf{p}(k) - (\mathbf{p}(k))^\top \boldsymbol{\Gamma}\mathbf{p}(k)\big) - \alpha\xi_i(k) - \theta_i(k).$$

*Proof.* As the dynamic of $\mathbf{p}$ still follows Eq. (4) as in the uncorrelated model, we can argue as in the proof of Lemma A.2 to obtain that for some $\gamma \in (0,1)$

$$p_i(k+1) = p_i(k) + \alpha p_i(k)\Big(Y_i(k) - \sum_{j=1}^{d} p_j(k)Y_j(k)\Big)$$

$$- \alpha^2 p_i(k)\frac{\sum_{j=1}^{d} p_j(k)Y_j(k)\Big(Y_i(k) - \sum_{j=1}^{d} p_j(k)Y_j(k)\Big)}{\big(1 + \gamma \sum_{j=1}^{d} p_j(k)Y_j(k)\big)^3}.$$

Then, since $|Y_i(k)| \leq Q$ we obtain the same bound on the error term as in the proof of Lemma A.2

$$\left| \alpha^2 p_i(k)\frac{\sum_{j=1}^{d} p_j(k)Y_j(k)\Big(Y_i(k) - \sum_{j=1}^{d} p_j(k)Y_j(k)\Big)}{\big(1 + \gamma \sum_{j=1}^{d} p_j(k)Y_j(k)\big)^3} \right|$$

$$\leq \alpha^2 \frac{2Q^2}{(1-Q\alpha)^3} p_i(k)(1 - p_i(k)).$$

Since the $p_i(k)$ are $\mathcal{F}_{k-1}$-measurable and $\mathbf{C}(k)$ is independent of $\mathcal{F}_{k-1}$, we finally obtain $\mathbb{E}[Y_i(k)|\mathcal{F}_{k-1}] = \mathbf{\Gamma}_{i,\cdot}\mathbf{p}(k)$. $\qquad\square$

In the following we always assume for $\Delta_p, \Delta_{\mathbf{\Gamma}} > 0$

$$p_1(0) \geq p_i(0) + \Delta_p, \quad \forall i = 2, \ldots, d,$$
$$(\mathbf{\Gamma}\mathbf{p}(0))_1 \geq (\mathbf{\Gamma}\mathbf{p}(0))_i + \Delta_{\mathbf{\Gamma}}, \quad \forall i = 2, \ldots, d, \tag{38}$$
$$c_\star := \frac{\Delta_p \Delta_{\mathbf{\Gamma}}}{4} - \nu\Big(1 + \frac{\Delta_p \Delta_{\mathbf{\Gamma}}}{4}\Big) > 0, \quad \text{where,} \quad \nu := \max_{i \neq j \in [d]} \mathbf{\Gamma}_{i,j}.$$

These assumptions are for example fulfilled in the following case

$$\mathbf{p}(0) = \begin{pmatrix} 0.8 \\ 0.1 \\ 0.1 \end{pmatrix}, \quad \mathbf{\Gamma} = \begin{pmatrix} 1 & 0.1 & 0.1 \\ 0.1 & 1 & 0 \\ 0.1 & 0 & 1 \end{pmatrix},$$

since we can choose $\Delta_p = 0.7, \Delta_{\mathbf{\Gamma}} = 0.64, \nu = 0.1$, and hence $c_\star = 8 * 10^{-4}$. In the independent case, which corresponds to $\nu = 0$, the assumptions given in Eq. (38) reduce to the original assumption given in Eq. (21). For $k = 1, 2, \ldots$, we define a sequence of benign events

$$\Omega(k) := \Big\{p_1(u) \geq \max_{i=2,\ldots,d} p_i(u) + \frac{\Delta_p}{2}, (\mathbf{\Gamma}\mathbf{p}(u))_1 \geq \max_{i=2,\ldots,d} (\mathbf{\Gamma}\mathbf{p}(u))_i + \frac{\Delta_{\mathbf{\Gamma}}}{2}, \forall u \in [k]\Big\}.$$

Due to assumption Eq. (38), $\Omega(0) = \Omega$. Additionally, since the assumptions in Eq. (38) imply the gradient to be bounded away from 0, we can prove the following recursive upper bound for $1 - p_1(k)$.

**Proposition A.11.** *If*

$$0 < \alpha \leq \frac{(1-Q\alpha)^3}{8Q^2}\Delta_{\mathbf{\Gamma}},$$

*then, on the event $\Omega(k)$,*

$$1 - p_1(k+1) \leq \Big(1 - \alpha\frac{\Delta_{\mathbf{\Gamma}}}{4d}\Big(1 + \frac{\Delta_p}{2}(d-1)\Big)\Big)(1 - p_1(k)) + \alpha\xi_i(k).$$

*Proof.* By definition, we have on the event $\Omega(k)$,

$$(\mathbf{\Gamma}\mathbf{p}(k))_1 - \mathbf{p}(k)^\top\mathbf{\Gamma}\mathbf{p}(k) = \sum_{j=1}^{d} p_j(k)\big((\mathbf{\Gamma}\mathbf{p}(k))_1 - (\mathbf{\Gamma}\mathbf{p}(k))_j\big)$$

$$\geq \frac{\Delta_{\mathbf{\Gamma}}}{2}\big(1 - p_1(k)\big). \tag{39}$$

The assumption on $\alpha$ implies $\alpha < 1/Q$ and thus Lemma A.10 becomes applicable. Now applying Lemma A.10 with $i = 1$, gives on the event $\Omega(k)$,

$$1 - p_1(k+1)$$

$$\leq 1 - p_1(k) - \alpha \frac{\Delta_\Gamma}{2} p_1(k)(1 - p_1(k)) + \alpha \xi_i(k) + \theta_i(k)$$

$$\leq 1 - p_1(k) - \alpha \Big( \frac{\Delta_\Gamma}{2} - \alpha \frac{2Q^2}{(1 - Q\alpha)^3} \Big) p_1(k)(1 - p_1(k)) + \alpha \xi_i(k)$$

$$\leq 1 - p_1(k) - \alpha \frac{\Delta_\Gamma}{4} p_1(k)(1 - p_1(k)) + \alpha \xi_i(k)$$

$$\leq \Big( 1 - \alpha \frac{\Delta_\Gamma}{4d} \Big( 1 + \frac{\Delta_p}{2}(d - 1) \Big) \Big)(1 - p_1(k)) + \alpha \xi_i(k)$$

This concludes the proof. $\qquad\square$

As in the uncorrelated setting our goal is now to derive a lower bound for the probability of the favourable event $\Omega(k)$. For this we again follow the same strategy and rely on uniform concentration inequalities for martingales. In order apply those, we require the following lemma, which states that $\Omega(k)$ is fulfilled as soon as

$$M_j(k) := \sum_{\ell=0}^{k} \alpha \xi_j(\ell) \mathbb{1}_{\Omega(\ell)}, \quad k = 0, 1, \dots \tag{40}$$

with $\xi_j(\ell)$ defined in Eq. (22), exhibits a uniform concentration behaviour. In the following we denote by $\|\Gamma\|_\infty$ the row-sum norm of $\Gamma$, i.e. $\|\Gamma\|_\infty = \max_{i \in [d]} \sum_{j=1}^{d} \Gamma_{i,j}$.

**Lemma A.12.** *Define the sets*

$$(E)_j(k) := \Big\{ \max_{u \in [k]} |M_j(u)| \leq \frac{1}{4} \Big( \Delta_p \wedge \frac{\Delta_\Gamma}{\|\Gamma\|_\infty} \Big) \Big\}, \quad E(k) := \bigcap_{j=1}^{d} (E)_j(k), \quad k = 0, 1, \dots.$$

*Then if*

$$0 < \alpha \leq \frac{(1 - Q\alpha)^3}{4Q^2} c_\star,$$

*the following set inclusion holds for any $k = 0, 1, \dots$*

$$E(k) \subseteq \Omega(k+1).$$

*Proof.* Let $u \in \{2, \dots, d\}$ be arbitrary. By Lemma A.10, on the event $\Omega(k)$,

$$p_1(k+1) - p_u(k+1)$$
$$\geq p_1(k) - p_u(k) + \alpha \big( p_1(k) - p_u(k) \big) \big( (\Gamma \mathbf{p}(k))_1 - \mathbf{p}(k)^\top \Gamma \mathbf{p}(k) \big)$$
$$\quad + \alpha \big( \xi_j(k) - \xi_1(k) \big) - \theta_1(k) + \theta_u(k).$$

Since $p_u(k) \leq 1 - p_1(k)$ we also obtain on $\Omega(k)$,

$$-\theta_1(k) + \theta_u(k) \geq -\alpha^2 \frac{2Q^2}{(1 - Q\alpha)^3} \Big( p_1(k)\big(1 - p_1(k)\big) + p_u(k)(1 - p_u(k)) \Big)$$

$$\geq -\alpha^2 \frac{4Q^2}{(1 - Q\alpha)^3} \big( 1 - p_1(k) \big).$$

The assumptions on $\alpha$ then imply that on the event $\Omega(k)$,

$$p_1(k+1) - p_u(k+1) - \alpha(\xi_j(k) - \xi_1(k))$$

$$\geq p_1(k) - p_u(k) + \alpha \big( p_1(k) - p_u(k) \big) \big( (\Gamma \mathbf{p}(k))_1 - \mathbf{p}(k)^\top \Gamma \mathbf{p}(k) \big) - \alpha^2 \frac{4Q^2}{(1 - Q\alpha)^3} \big( 1 - p_1(k) \big)$$

$$\geq p_1(k) - p_u(k).$$

Because of $\Omega(k) \subseteq \Omega(k-1)$ it then follows,

$$(p_1(k+1) - p_u(k+1))\mathbb{1}_{\Omega(k)} \geq (p_1(k) - p_u(k))\mathbb{1}_{\Omega(k)} + \alpha(\xi_j(k) - \xi_1(k))\mathbb{1}_{\Omega(k)}$$
$$= \mathbb{1}_{\Omega(k)}\Big((p_1(k) - p_u(k))\mathbb{1}_{\Omega(k-1)} + \alpha(\xi_j(k) - \xi_1(k))\mathbb{1}_{\Omega(k)}\Big).$$

This gives

$$(p_1(k+1) - p_u(k+1))\mathbb{1}_{\Omega(k)}$$
$$\geq \mathbb{1}_{\Omega(k)}\Big(p_1(0) - p_u(0) + \sum_{\ell=0}^{k} \alpha(\xi_j(\ell) - \xi_1(\ell))\mathbb{1}_{\Omega(\ell)}\Big) \tag{41}$$
$$\geq \Delta_p \mathbb{1}_{\Omega(k)} - |M_j(k)| - |M_1(k)|.$$

Now again let $u \in \{2, \ldots, d\}$ be given. Then it holds, on $\Omega(k)$

$$(\mathbf{\Gamma}\mathbf{p}(k+1))_1 - (\mathbf{\Gamma}\mathbf{p}(k+1))_u$$
$$= (\mathbf{\Gamma}\mathbf{p}(k))_1 - (\mathbf{\Gamma}\mathbf{p}(k))_u + \alpha \sum_{j=1}^{d} \mathbf{\Gamma}_{1,j} p_j(k)((\mathbf{\Gamma}\mathbf{p}(k))_j - \mathbf{p}(k)\mathbf{\Gamma}\mathbf{p})$$
$$- \alpha \sum_{j=1}^{d} \mathbf{\Gamma}_{u,j} p_j(k)((\mathbf{\Gamma}\mathbf{p}(k))_j - \mathbf{p}(k)\mathbf{\Gamma}\mathbf{p}) - \sum_{j=1}^{d}(\mathbf{\Gamma}_{1,j} - \mathbf{\Gamma}_{u,j})(\alpha\xi_j(k) + \theta_j(k))$$
$$\geq (\mathbf{\Gamma}\mathbf{p}(k))_1 - (\mathbf{\Gamma}\mathbf{p}(k))_u + \alpha(1 - \mathbf{\Gamma}_{1,u})(p_1(k) - p_u(k))((\mathbf{\Gamma}\mathbf{p}(k))_1 - \mathbf{p}(k)\mathbf{\Gamma}\mathbf{p})$$
$$+ \alpha \sum_{j=2,j\neq u}^{d}(\mathbf{\Gamma}_{1,j} - \mathbf{\Gamma}_{u,j})p_j(k)((\mathbf{\Gamma}\mathbf{p}(k))_j - \mathbf{p}(k)\mathbf{\Gamma}\mathbf{p})$$
$$- \sum_{j=1}^{d}(\mathbf{\Gamma}_{1,j} - \mathbf{\Gamma}_{u,j})(\alpha\xi_j(k) + \theta_j(k))$$
$$\geq (\mathbf{\Gamma}\mathbf{p}(k))_1 - (\mathbf{\Gamma}\mathbf{p}(k))_u + \alpha(1 - \mathbf{\Gamma}_{1,u})\frac{\Delta_p\Delta_{\mathbf{\Gamma}}}{4}(1 - p_1(k)) - \alpha\nu(1 - p_1(k))$$
$$- \sum_{j=1}^{d}(\mathbf{\Gamma}_{1,j} - \mathbf{\Gamma}_{u,j})(\alpha\xi_j(k) + \theta_j(k))$$
$$\geq (\mathbf{\Gamma}\mathbf{p}(k))_1 - (\mathbf{\Gamma}\mathbf{p}(k))_u + \alpha\Big((1 - \nu)\frac{\Delta_p\Delta_{\mathbf{\Gamma}}}{4} - \nu - \alpha\frac{4Q^2}{(1 - Q\alpha)^3}\Big)(1 - p_1(k))$$
$$- \alpha \sum_{j=1}^{d}(\mathbf{\Gamma}_{1,j} - \mathbf{\Gamma}_{u,j})\xi_j(k)$$
$$\geq (\mathbf{\Gamma}\mathbf{p}(k))_1 - (\mathbf{\Gamma}\mathbf{p}(k))_u - \alpha \sum_{j=1}^{d}(\mathbf{\Gamma}_{1,j} - \mathbf{\Gamma}_{u,j})\xi_j(k).$$

Hence, arguing as in the derivation of Eq. (41) gives

$$\Big((\mathbf{\Gamma}\mathbf{p}(k+1))_1 - (\mathbf{\Gamma}\mathbf{p}(k+1))_u\Big)\mathbb{1}_{\Omega(k)}$$
$$\geq \mathbb{1}_{\Omega(k)}\Big((\mathbf{\Gamma}\mathbf{p}(0))_1 - (\mathbf{\Gamma}\mathbf{p}(0))_u - \alpha \sum_{\ell=0}^{k}\sum_{j=1}^{d}(\mathbf{\Gamma}_{1,j} - \mathbf{\Gamma}_{u,j})\xi_j(\ell)\mathbb{1}_{\Omega(\ell)}\Big)$$
$$= \mathbb{1}_{\Omega(k)}\Big((\mathbf{\Gamma}\mathbf{p}(0))_1 - (\mathbf{\Gamma}\mathbf{p}(0))_u - \sum_{j=1}^{d}(\mathbf{\Gamma}_{1,j} - \mathbf{\Gamma}_{u,j})M_j(k)\Big) \tag{42}$$
$$\geq \mathbb{1}_{\Omega(k)}\Big((\mathbf{\Gamma}\mathbf{p}(0))_1 - (\mathbf{\Gamma}\mathbf{p}(0))_u - 2\|\mathbf{\Gamma}\|_\infty \max_{j\in[d]}|M_j(k)|\Big).$$

With the above results we can now begin proving that $E(k) \subseteq \Omega(k+1)$ for all $k = 0, 1, \ldots$. We do this by induction. For $k = 0$, this directly follows from Eq. (41) and Eq. (42), since $\Omega(0) = \Omega$

by assumption. Now, assume the assertion holds for some $k = 0, 1, \ldots$. This implies $E(k+1) \subseteq E(k) \subseteq \Omega(k+1)$, such that for any $u \in \{2, \ldots, d\}$ it holds on $E(k+1)$ by Eq. (41)

$$(p_1(k+2) - p_u(k+2)) \geq \Delta_p - |M_j(k+1)| - |M_1(k+1)|$$

$$\geq \frac{\Delta_p}{2},$$

and additionally by Eq. (42) it holds on $E(k+1)$

$$(\mathbf{\Gamma p}(k+2))_1 - (\mathbf{\Gamma p}(k+2))_u \geq (\mathbf{\Gamma p}(0))_1 - (\mathbf{\Gamma p}(0))_u - 2\|\mathbf{\Gamma}\|_\infty \max_{j \in [d]} |M_j(k+1)|$$

$$\geq \frac{\Delta_{\mathbf{\Gamma}}}{2},$$

which proves the assertion. $\qquad\square$

With the above results we are now able to prove and state the main theorem for the correlated case.

**Theorem A.13.** *Given $\varepsilon \in (0,1)$, assume*

$$\Delta_p := p_1(0) - \max_{i=2,\ldots,d} p_i(0) > 0, \quad \Delta_{\mathbf{\Gamma}} := (\mathbf{\Gamma p}(0))_1 - \max_{i=2,\ldots,d}(\mathbf{\Gamma p}(0))_1 > 0,$$

$$c_\star := \frac{\Delta_p \Delta_{\mathbf{\Gamma}}}{4} - \nu\left(1 + \frac{\Delta_p \Delta_{\mathbf{\Gamma}}}{4}\right) > 0, \quad where, \quad \nu := \max_{i \neq j \in [d]} \mathbf{\Gamma}_{i,j},$$

*and*

$$0 < \alpha \leq \frac{1}{4Q^2}\left((1 - Q\alpha)^3 c_\star \wedge \frac{1}{1024(1 - p_1(0))}\left(\Delta_p \wedge \frac{\Delta_{\mathbf{\Gamma}}}{\|\mathbf{\Gamma}\|_\infty}\right)^2 \Delta_{\mathbf{\Gamma}}\left(\frac{4}{d} + \Delta_p\right)\right).$$

*Then there exists an event $\Theta$ with probability $\geq 1 - \varepsilon/2$ such that*

$$\mathbb{E}\big[\|\mathbf{p}(k) - \mathbf{e}_1\|_1 \mathbb{1}_\Theta\big] \leq 2(1 - p_1(0)) \exp\left(-\alpha \frac{\Delta_{\mathbf{\Gamma}}}{16}\left(\frac{4}{d} + \Delta_p\right)k\right), \quad for\ all\ k = 0, 1, \ldots$$

*Consequently, given $\delta > 0$, it holds*

$$\mathbb{P}\left(\|\mathbf{p}(k) - \mathbf{e}_1\|_1 \geq \delta\right) \leq \varepsilon \quad for\ all \quad k \geq \frac{16d}{\alpha \Delta_{\mathbf{\Gamma}}(4 + d\Delta_p)} \log\left(\frac{4(1 - p_1(0))}{\varepsilon\delta}\right).$$

Before giving the proof of the above theorem, we want to remark that Theorem A.13 exactly recovers the result of Theorem 2.2 in the independent case. Indeed, in the independent case $\mathbf{\Gamma}$ is equal to the identity matrix and thus $\Delta_p = \Delta_{\mathbf{\Gamma}}, \nu = 0$ and $c_\star = \Delta_p^2/4$ hold true, which gives the result of Theorem 2.2.

*Proof of Theorem A.13.* As in the uncorrelated case, the recursive definition ensures that $\mathbf{p}(k)$ is $\mathcal{F}_{k-1}$-measurable. Thus, also $\Omega(k) \in \mathcal{F}_{k-1}$ for any $k = 0, 1, \ldots$. Then $(M_i(k))_{k=0,1,\ldots}$, defined in Eq. (40), is a martingale for each $i \in [d]$. This allows us to apply Doob's submartingale inequality. For this, we deduce the following bound on the second moment,

$$\mathbb{E}[M_1(k)^2]$$

$$= \alpha^2 \sum_{\ell=0}^{k} \mathbb{E}[(\xi_1(\ell)\mathbb{1}_{\Omega(\ell)})^2]$$

$$= \alpha^2 \sum_{\ell=0}^{k} \mathbb{E}\left[(p_1(\ell))^2 \left(\mathbb{E}\left[Y_1(\ell) - \sum_{j=1}^{d} p_j(\ell)Y_j(\ell)\Big|\mathcal{F}_{\ell-1}\right] - \left(Y_1(\ell) - \sum_{j=1}^{d} p_j(\ell)Y_j(\ell)\right)\right)^2 \mathbb{1}_{\Omega(\ell)}\right]$$

$$\leq \alpha^2 \sum_{\ell=0}^{k} \mathbb{E}\left[\mathbb{E}\left[\left(\mathbb{E}\left[Y_1(\ell) - \sum_{j=1}^{d} p_j(\ell)Y_j(\ell)\Big|\mathcal{F}_{\ell-1}\right] - \left(Y_1(\ell) - \sum_{j=1}^{d} p_j(\ell)Y_j(\ell)\right)\right)^2\Big|\mathcal{F}_{\ell-1}\right]\mathbb{1}_{\Omega(\ell)}\right]$$

$$\leq \alpha^2 \sum_{\ell=0}^{k} \mathbb{E}\left[\left(Y_1(\ell) - \sum_{j=1}^{d} p_j(\ell)Y_j(\ell)\right)^2 \mathbb{1}_{\Omega(\ell)}\right]$$

$$= \alpha^2 \sum_{\ell=0}^{k} \mathbb{E}\Big[\Big((1 - p_1(\ell))Y_1(\ell) - \sum_{j=2}^{d} p_j(\ell)Y_j(\ell)\Big)^2 \mathbb{1}_{\Omega(\ell)}\Big]$$

$$\leq 2Q^2\alpha^2 \sum_{\ell=0}^{k} \mathbb{E}\Big[\Big((1 - p_1(\ell))^2 + \Big(\sum_{j=2}^{d} p_j(\ell)\Big)^2\Big)\mathbb{1}_{\Omega(\ell)}\Big]$$

$$\leq 4Q^2\alpha^2 \sum_{\ell=0}^{k} \mathbb{E}\Big[(1 - p_1(\ell))\mathbb{1}_{\Omega(\ell)}\Big],$$

where we argued similarly as for the independent case. Furthermore, we obtain for $u \in \{2, \ldots, d\}$,

$$\mathbb{E}[(M_u(k))^2] = \alpha^2 \mathbb{E}\Big[\sum_{\ell=0}^{k}(\xi_u(\ell)\mathbb{1}_{\Omega(\ell)})^2\Big]$$

$$\leq \alpha^2 \sum_{\ell=0}^{k} \mathbb{E}\Big[(p_u(k))^2\Big(Y_u(\ell) - \sum_{j=1}^{d} p_j(\ell)Y_j(\ell)\Big)^2 \mathbb{1}_{\Omega(\ell)}\Big]$$

$$\leq 4Q^2\alpha^2 \sum_{\ell=0}^{k} \mathbb{E}\Big[p_u(\ell)\mathbb{1}_{\Omega(\ell)}\Big].$$

Hence, applying a union bound, Doob's submartingale inequality Eq. (27) with $p = 2$ gives for any $k = 0, 1, \ldots$

$$\mathbb{P}(E(k)) = 1 - \mathbb{P}\Big(\bigcup_{j=1}^{d} \max_{u \in [k]} |M_j(u)| \geq \frac{1}{4}\Big(\Delta_p \wedge \frac{\Delta_\Gamma}{\|\Gamma\|_\infty}\Big)\Big)$$

$$\geq 1 - \sum_{j=1}^{d} \mathbb{P}\Big(\max_{u \in [k]} |M_j(u)| \geq \frac{1}{4}\Big(\Delta_p \wedge \frac{\Delta_\Gamma}{\|\Gamma\|_\infty}\Big)\Big)$$

$$\geq 1 - 64Q^2\alpha^2\Big(\Delta_p \wedge \frac{\Delta_\Gamma}{\|\Gamma\|_\infty}\Big)^{-2}\Big(\sum_{\ell=0}^{k} \mathbb{E}\Big[(1 - p_1(\ell))\mathbb{1}_{\Omega(\ell)}\Big] + \sum_{j=2}^{d}\sum_{\ell=0}^{k} \mathbb{E}\Big[p_j(\ell)\mathbb{1}_{\Omega(\ell)}\Big]\Big)$$

$$= 1 - 128Q^2\alpha^2\Big(\Delta_p \wedge \frac{\Delta_\Gamma}{\|\Gamma\|_\infty}\Big)^{-2} \sum_{\ell=0}^{k} \mathbb{E}\Big[(1 - p_1(\ell))\mathbb{1}_{\Omega(\ell)}\Big].$$

Proposition A.11 gives for any $k = 0, 1, \ldots$ the bound

$$\mathbb{E}\Big[(1 - p_1(k+1))\mathbb{1}_{\Omega(k+1)}\Big]$$

$$\leq \mathbb{E}\Big[(1 - p_1(k+1))\mathbb{1}_{\Omega(k)}\Big]$$

$$\leq \mathbb{E}\Big[\Big(1 - \alpha\frac{\Delta_\Gamma}{4d}\Big(1 + \frac{\Delta_p}{2}(d-1)\Big)\Big)(1 - p_1(k))\mathbb{1}_{\Omega(k)} + \alpha\xi_1(k)\mathbb{1}_{\Omega(k)}\Big]$$

$$= \Big(1 - \alpha\frac{\Delta_\Gamma}{4d}\Big(1 + \frac{\Delta_p}{2}(d-1)\Big)\Big)\mathbb{E}\Big[(1 - p_1(k))\mathbb{1}_{\Omega(k)}\Big],$$

which implies

$$\mathbb{E}\Big[(1 - p_1(k))\mathbb{1}_{\Omega(k)}\Big] \leq (1 - p_1(0))\Big(1 - \alpha\frac{\Delta_\Gamma}{4d}\Big(1 + \frac{\Delta_p}{2}(d-1)\Big)\Big)^k. \tag{43}$$

We set

$$\Theta := \bigcap_{k=0}^{\infty} \Omega(k).$$

The continuity of probability measures and Lemma A.12 then imply

$$\mathbb{P}(\Theta) = \lim_{k \to \infty} \mathbb{P}(\Omega(k))$$

$$\geq \lim_{k \to \infty} \mathbb{P}(E(k))$$

$$\geq 1 - 128 Q^2 \alpha^2 \Big( \Delta_p \wedge \frac{\Delta_{\mathbf{\Gamma}}}{\|\mathbf{\Gamma}\|_\infty} \Big)^{-2} \sum_{\ell=0}^{\infty} \mathbb{E}\Big[ (1 - p_1(\ell)) \mathbb{1}_{\Omega(\ell)} \Big]$$

$$\geq 1 - 128 Q^2 \alpha^2 (1 - p_1(0)) \Big( \Delta_p \wedge \frac{\Delta_{\mathbf{\Gamma}}}{\|\mathbf{\Gamma}\|_\infty} \Big)^{-2} \sum_{\ell=0}^{\infty} \Big( 1 - \alpha \frac{\Delta_{\mathbf{\Gamma}}}{4d} \Big( 1 + \frac{\Delta_p}{2}(d-1) \Big) \Big)^{\ell}$$

$$= 1 - 1024 Q^2 \alpha (1 - p_1(0)) \Big( \Delta_p \wedge \frac{\Delta_{\mathbf{\Gamma}}}{\|\mathbf{\Gamma}\|_\infty} \Big)^{-2} \frac{d}{\Delta_{\mathbf{\Gamma}} \big( 2 + \Delta_p(d-1) \big)}$$

$$= 1 - 2048 Q^2 \alpha (1 - p_1(0)) \Big( \Delta_p \wedge \frac{\Delta_{\mathbf{\Gamma}}}{\|\mathbf{\Gamma}\|_\infty} \Big)^{-2} \frac{1}{\Delta_{\mathbf{\Gamma}} (4/d + \Delta_p)}$$

$$\geq 1 - \varepsilon/2,$$

where we used that we can assume $d \geq 2$ without loss of generality. Additionally, Eq. (43) and the elementary inequality $1 - x \leq \exp(-x)$, which is valid for any real number $x$, give

$$\mathbb{E}\Big[ (1 - p_1(k)) \mathbb{1}_\Theta \Big] \leq \mathbb{E}\Big[ (1 - p_1(k)) \mathbb{1}_{\Omega(k)} \Big]$$

$$\leq (1 - p_1(0)) \Big( 1 - \alpha \frac{\Delta_{\mathbf{\Gamma}}}{4d} \Big( 1 + \frac{\Delta_p}{2}(d-1) \Big) \Big)^{k}$$

$$\leq (1 - p_1(0)) \exp\Big( - \alpha \frac{\Delta_{\mathbf{\Gamma}}}{4d} \Big( 1 + \frac{\Delta_p}{2}(d-1) \Big) k \Big).$$

When $d = 1$, the right hand side of this inequality is 0. For $d \geq 2$, we can also use the bound $d - 1 \geq d/2$. Together with

$$\|\mathbf{p}(k) - \mathbf{e}_1\|_1 = 1 - p_1(k) + \sum_{i=2}^{d} p_i(k) = 2 \big( 1 - p_1(k) \big),$$

this concludes the proof of the first statement. For the proof of the second statement, we apply Markov's inequality to obtain

$$\mathbb{P}\Big( \|\mathbf{p}(k) - \mathbf{e}_1\|_1 \geq \delta \Big)$$

$$\leq \mathbb{P}((\Theta)^{\mathbf{C}}) + \mathbb{P}\Big( \|\mathbf{p}(k) - \mathbf{e}_1\|_1 \mathbb{1}_\Theta \geq \delta \Big)$$

$$\leq \frac{\varepsilon}{2} + 2(1 - p_1(0)) \exp\Big( - \alpha \frac{\Delta_{\mathbf{\Gamma}}}{16} \Big( \frac{4}{d} + \Delta_p \Big) k \Big) \delta^{-1}.$$

Hence, if

$$k \geq \Big( \frac{\alpha \Delta_{\mathbf{\Gamma}}}{16} \Big( \frac{4}{d} + \Delta_p \Big) \Big)^{-1} \log \Big( \frac{4(1 - p_1(0))}{\varepsilon \delta} \Big)$$

$$= \frac{16d}{\alpha \Delta_{\mathbf{\Gamma}} (4 + d \Delta_p)} \log \Big( \frac{4(1 - p_1(0))}{\varepsilon \delta} \Big),$$

then,

$$\mathbb{P}\Big( \|\mathbf{p}(k) - \mathbf{e}_1\|_1 \geq \delta \Big) \leq \varepsilon.$$

$\square$

