# OpenReview forum: "Spike-timing-dependent Hebbian learning as noisy gradient descent"
_NeurIPS.cc/2025/Conference — NeurIPS 2025 poster_

### Official Review · Reviewer_2zhK · 2025-06-26

**Clarity:** 4
**Significance:** 2
**Originality:** 2
**Rating:** 4
**Confidence:** 5

**Summary:**

In this manuscript, the authors return to classic pair-based STDP models, in which the change of a synapse depends on the relative timing of the spikes of the pre- and postsynaptic neurons. The authors prove mathematically that this type of STDP model identifies the pre-neuron with the highest triggering probability. Moreover, they show that it is connected to noisy mirror descent.

**Questions:**

Please see "Strengths And Weaknesses" for my concern and questions.

**Ethical Concerns:**

["NO or VERY MINOR ethics concerns only"]

**Final Justification:**

As noted in my review, I recommend borderline acceptance to NeurIPS.

**Limitations:**

No. The authors have not adequately addressed the limitations of their work. While the theoretical analysis is presented clearly, the manuscript overlooks several critical limitations that should be acknowledged and discussed:

- Connection to Prior Work: The authors are not aware of important, closely related literature, which limits the novelty of their results. (Please see "Strengths And Weaknesses" for more detail.)

- Narrow Scope of Input Models: The work assumes independent, homogeneous Poisson inputs—a highly restrictive and biologically unrealistic setting. This limits the applicability of the results. The potential for generalization to inhomogeneous or correlated inputs should be addressed.

- Simplifying Assumptions in STDP Rule: The analysis is restricted to a specific, highly idealized form of pair-based STDP—likely the balanced, antisymmetric case. The authors should clarify whether their results hold for more general STDP methods.

- Biological Plausibility: The model assumes pair-based STDP, which has been shown to be an incomplete description of biological plasticity. Experimental studies (e.g., Sjöström et al., 2001; Clopath et al.) have demonstrated the importance of triplet interactions and voltage dependence. These factors should at least be acknowledged as limitations of the modeling framework.

**Quality:**

2

**Strengths And Weaknesses:**

In the literature (reviewed in the cited paper of Morrison et al.), there are many versions of such a pair-based STDP rule. If I understand correctly, the authors focus on the one where, for the sequence post-before-pre and pre-before-post, only the closest postsynaptic spike matters, whereas all presynaptic spikes between two postsynaptic spikes are taken into account. This assumption is well accepted, and I don't criticize it.


Existing analytical models of STDP (e.g. Kempter et al. 1999 as cited) have mostly assumed stochastic spike arrival with Poisson rate $\lambda_j$  and stochastic output spiking with instantaneous rate (conditional stochastic intensity)  $\lambda_{post} = \sum_j w_j \int_0^\infty \epsilon(s) \lambda_j(t-s) ds$ with some integration kernel $\epsilon$. They found that pair-based STDP performs PCA. For uncorrelated Poisson input, there is a positive feedback loop proportional to the overlap
$\int_0^\infty  W(s) \epsilon(s) w_j \lambda_j ds$ where $W(s)$ is the causal (=pre-before-post) section of the STDP window.

The authors modify this standard paradigm a little bit in that they assume that the postsynaptic neuron has no intrinsic noise. Instead, output spikes are defined by a deterministic threshold crossing process (leaky integrate-and-fire, LIF). Under the assumption (i) that the kernel $\epsilon$ is a jump followed by exponential decay, and (ii) that the stochastic firing intensity of presynaptic neurons is constant, the authors correctly observe that the threshold crossing occurs at the moment of (or arbitrarily shortly after) presynaptic spike arrival and that the relative probability that a presynaptic neuron j is the one that triggers the postsynaptic spike is (if all weights are small)

$$p(j) = \frac{\lambda_j w_j}{\sum_l \lambda_l w_l}.  (Eq. 12)$$

I like this observation, and it looks novel to me!
From this observation, together with the assumption that a sequence pre-before-post leads to weight increase, it follows immediately that the input weight vector becomes in the limit parallel to $p(j)$. Indeed, the resulting self-feedback term is very close to that of Kempter et al. (1999). The difference is that in the noisy neuron model of Kempter, the full time course of $\epsilon$ plays a role, whereas in the deterministic LIF model, only the initial moment of the rapid rise of $\epsilon$ plays a role. Indeed, it is known in computational neuroscience that the threshold process of an LIF generates correlations that are sensitive to the derivative of $\epsilon$.

Starting from Eq. 12 (which is unfortunately only given in Section 2.3), the authors have arrived at a formal mathematical model that distinguishes the input spike that triggers the postsynaptic spike from all other input spikes. This model enables mathematical analysis. As expected from earlier work, weights explode, and the weight vector becomes aligned with the strongest input.

In the formal mathematical model (from line 225), the algorithm "selects a presynaptic spike time $\tau^* > t_k$ and then sets $t_{k+1} = \tau^∗$. This is done in two stages by first choosing a presynaptic neuron $j^∗(k + 1) \in [d]$ and, in a second step, picking a spike time $τ^∗$ among the spike times of the $j^∗(k + 1)^{th}$ presynaptic neuron." I would have preferred to see first the biological motivation of Section 2.3. before the abstract mathematical model, but this is a matter of taste and only a minor point.

----

**Major**:

1. The authors are not aware of important closely related literature  which limits the novelty of their results. Between 1996 and 2007, (pair-based) STDP attracted a lot of research interest.

1.1 Kempter et al. (1999, cited) already analyzed the evolution of synaptic weights in a mathematical model where the (stochastic) generation of output spikes depend linearly on the input spikes. Even though the noise model is different from that of the current manuscript, the results are essentially the same. Importantly, Kempter et al. (1999) cover a much broader range of input paradigms that go well beyond the independent homogeneous Poisson processes considered in the current manuscript.

1.2. As part of the efforts 20 years ago, several loss functions have been proposed that yield STDP via gradient descent. Noteworthy examples are Chechik 2003, Bohte and Moser 2005, Bell and Para 2005, Toyoizumi et al. 2005; Toyoizumi et al. 2005a, Pfister et al. 2006, Toyoizumi et al. 2007.

1.3. The general theoretical framework of gradient descent in computatonal neuroscience has been reviewed by Surace et al. 2020 who highlight the importance of a metric in order to fully define gradient descent. The metric of mirror descent is one example. A rescaling of parameters from a to b=exp(a) to guarantee positive values of b is a standard trick in the field.

2. It is unfortunate that the authors consider only a very specialized stimulation paradigm and a very specific case of STDP.

2.1. I believe it would be possible to generalize the approach (i)  from homogeneous to inhomogeneous Poisson input using the time rescaling theorem and (ii) from independent to correlated Poisson inputs. For the latter, several possibilities exist to generate correlations in the input. Note that Kempter et al. (1999) already analyzed for their neuron model a rather general case of inputs. They indeed found that STDP generated PCA as conjectured in the manuscript.

2.2 In standard pair based STDP there are four parameters, i.e., two time constants and two amplitude parameters. There is a well-known special case of STDP which is called balanced STDP. In this special case the total area under the STDP window is zero. If, in addition you require that the time constants for pre-before-post and post-before-pre are identical, then we get an exactly antisymmetric STDP window.  If I am not mistaken, the authors make both assumptions (integral zero and same time constant) to get their Eq. (11). How would the results generalize to other cases?

2.3. Unfortunately, pair-based STDP is NOT a good representation of biological data as shown by Sjostrom et al. (2001). First, more than two spikes interact ('triplet effect') and second, the voltage at the location of the synapse also plays a role (for both, see e.g.  the cited paper of Clopath et al). These observations  might have contributed to the reduced interest in computational neuroscience for standard pair-based STDP models even though they are still widely used.


Additional references:


Sjostrom, P., G. Turrigiano, and S. Nelson (2001). Rate, timing, and cooperativity jointly determine cortical synaptic plasticity. Neuron 32, 1149–1164.

Chechik, G. (2003). Spike-timing-dependent plasticity and relevant mututal information
maximization. Neural Computation 15, 1481–1510

Bohte, S. M. and M. C. Mozer (2005). Reducing spike train variability: A computational
theory of spike-timing dependent plasticity. In L. K. Saul, Y. Weiss, and L. Bottou
(Eds.), Advances in Neural Information Processing Systems 17, pp. 201–208. Cam-
bridge, MA: MIT Press

Bell, A. J. and L. C. Parra (2005). Maximising sensitivity in a spiking network. In L. K.
Saul, Y. Weiss, and L. Bottou (Eds.), Advances in Neural Information Processing
Systems 17, pp. 121–128. Cambridge, MA: MIT Press.


T. Toyoizumi, J.-P. Pfister, K. Aihara, and W. Gerstner (2007)
Optimality Model of Unsupervised Spike-Timing Dependent Plasticity: Synaptic Memory and Weight Distribution
Neural Computation, 19: 639-671


T. Toyoizumi, J.-P. Pfister, K. Aihara, and W. Gerstner (2005)
Generalized Bienenstock-Cooper-Munro rule for spiking neurons that maximizes information transmission
Proc. Natl. Acad. Sci. USA, 102:5239-5244

Toyoizumi, T., J.-P. Pfister, K. Aihara, and W. Gerstner (2005a). Spike-timing dependent
plasticity and mutual information maximization for a spiking neuron model. In L. K. Saul, Y. Weiss, and L. Bottou (Eds.), Advances in Neural Information Processing Systems 17, pp. 1409–1416. Cambridge, MA: MIT Press.

J.-P. Pfister, T. Toyoizumi, D. Barber, and W. Gerstner (2006)
Optimal Spike-Timing Dependent Plasticity for Precise Action Potential Firing in supervised learning
Neural Computation 18:1309-1339

T. Toyoizumi, J.-P. Pfister, K. Aihara, and W. Gerstner (2007)
Optimality Model of Unsupervised Spike-Timing Dependent Plasticity: Synaptic Memory and Weight Distribution
Neural Computation, 19: 639-67

S.C. Surace, J.-P. Pfister, W. Gerstner, and J. Brea (2020)
On the choice of metric in gradient-based theories of brain function
PLoS ComputBiol 16(4):e1007640,
doi: 10.1371/journal.pcbi.1007640

---

> ### Author Rebuttal · Authors · 2025-07-30
>
> This is a truly impressive report. The level of your expert knowledge of the field and the careful analysis of our submission is outstanding and has helped us a lot to position our work. We are especially grateful for the suggestions regarding related articles and will revise the paper to include a broader comparison with the existing literature. Overall, we do believe that we could address your comments.
> We would be delighted to receive more feedback during the forthcoming discussion process.
>
> - **1.1 Comparison with Kempter et al. 1999:**
>
>     Kempter et al. 1999 start with a noisy spike-time dependent dynamic which is transformed into a deterministic ODE by imposing a slow learning rate and using the self-averaging effect of the system. The important correlations between input and output spike sequences are encapsulated via a covariance type matrix. The ODE is analysed. The advantage of this approach is that a wide range of choices regarding the structure of the input can be made. In particular, one can include time-inhomogeneous intensities and correlations.
>
>     One major difference is the influence of the noise. In their work, the variance grows linearly in $t$ and a careful comparison of time scales is required. In our manuscript, despite a constant injection of noise into the system, the dynamic converges to a deterministic limit. (The comparison is more subtle as they study the weights and we study the dynamic of the triggering probabilities.) The second major difference is that we track the influence of the realised noise in every step and use recent ideas from the analysis of noisy SGD to prove convergence. We do acknowledge that a more detailed comparison was missing. This will be added to the revised version.
>
> - **1.2 Previously considered loss functions that yield STDP via gradient descent:**
>
>     As mentioned, this line of work starts with a loss function and then derives a form of gradient descent, while our approach starts with pair based STDP and derives from there a stochastic learning rule and an associated loss function. We will cite these articles in the revision and acknowledge that the connection between learning and gradient descent has been integral to prior work. In the revision, we will also compare to Scellier and Bengio '17.
>
> - **1.3 Comparison with Surace et al. 2020:** Thank you for pointing us to this article that we overlooked. While we already cite Cornford et al., in the revision we will emphasize that there has been more previous work in neuroscience highlighting that the Euclidean gradient might not be the most natural choice and mentioning the Fisher information metric in this context. We will moreover cite Neumann et al.\ "Intrinsic plasticity via natural gradient descent with application to drift compensation" (2013). The sentence "We also discover an intrinsic connection to noisy mirror descent" will be removed from the abstract.
>
> - **2.1 Generalization to non-homogeneous Poisson process and correlated inputs:** Thank you for asking about time-dependent intensities. In the revised version, we will add a small section deriving the following results:
>
>
>     **(a)**  Small perturbations of the intensity will not affect the convergence result. In fact in every step the intensities can vary by (small constant)$\times \alpha \Delta$ ($\alpha$ the learning rate, $\Delta$ the separation quantity) without changing the limit.
>
>
>      **(b)** Large changes in the intensities can of course lead to different limits and it is important in which order the intensities are presented to the learning rule. The next result (which we include in the revision) shows that the dynamic can be primed at the beginning to end up in one regime. Despite the noise and the infinite amount of data, the dynamic is unable to escape the domain of attraction.
>
>     **New result:** Given an initialization of the weights $(w_1(0),\ldots,w_d(0))^\top$ consider two $d$-dimensional intensity vectors $(\lambda_1,\ldots,\lambda_d)^\top, (\widetilde \lambda_1,\ldots, \widetilde \lambda_d)^\top$ with positive entries. Assume that $\lambda_1 w_1(0)> \max_{i=2,\ldots,d} \lambda_i w_i(0)$ and $\widetilde \lambda_d w_d(0)> \max_{i=1,\ldots,d-1} \widetilde \lambda_i w_i(0).$ Assume we run the learning rule (3) with intensity $\boldsymbol{\lambda}$ until time $K^* $ and then, for $k>K^* $, change the intensity to $\widetilde{\boldsymbol{\lambda}} $ and run the learning dynamic until time $k\to \infty.$ For any $\epsilon\in (0,1)$ and all sufficiently large $K^*$, the dynamics will converge to $\mathbf{e}_1$ with probability $1-2\epsilon.$
>
>
>     From the proof, one can derive quantitative bounds for $K^*.$
>
>
>     **(c)** We will also derive the gradient flow version in the time-inhomogeneous case
>         \begin{align*}
>         \dot{\mathbf{p}} =\mathbf{p}\odot\left(\dot{\log(\boldsymbol{\lambda})}-\mathbf{p}^T \dot{\log(\boldsymbol{\lambda})}\mathbf{1}\right)+\mathbf{p}\odot(\mathbf{p}-\Vert\mathbf{p}\Vert^2\mathbf{1}).
>     \end{align*}
>     Here, $\boldsymbol{\lambda}$ is a vector-valued function, $\log()$ has to be interpreted componentwise, and the dot denotes the time derivative.
>
>     **(d)**  We agree that the setting of independent inputs is too restrictive. In order to introduce correlations between the inputs, we propose to allow the inputs to jump simultaneously. The probability that input $i$ and $j$ jump at the same time is denoted by $\rho_{ij},$ and naturally $\rho_{ii}=1,$ for all $i\in[d]$. For $i,j\in[d],k\in\mathbb{N},$ we introduce independent random variables $c_{i,j}(k),$ such that $c_{i,j}(k) \sim \operatorname{Ber}(\rho_{i,j}),\  i\leq j\in[d],$ and we set $c_{j,i}:=c_{i,j}\  i\leq j\in[d],$ and $\mathbf{C}(k):=(c_{i,j}(k))_{ij}.$
>
>     The random variables $Z$ then stay centered, however the behaviour of the $\mathbf{B}$'s changes, because multiple inputs can jump at a postsynaptic spike. The probability of $B_i(k)$ being one is now given by $\boldsymbol{\rho}\_{i,\cdot} \mathbf{p}(k), $ where $\boldsymbol{\rho}=(\rho_{ij})\_{ij} $ is a $d\times d$ matrix and $\boldsymbol{\rho}_{i,\cdot}$ denotes the $i$-th row. Since multiple input neurons can spike simultaneously we adjust the modelling of the probabilities $\mathbf{p}$ to
>
>   \begin{align*}
>     p_i(k)=\frac{\lambda_i(w_i(k) + \sum_{j,j\neq i}c_{i,j}(k)w_j(k))}{\boldsymbol{\lambda}^\top \mathbf{w}(k) + \sum_{i,j,i\neq j}\lambda_ic_{i,j}(k)w_j(k)} = \frac{\lambda_i\boldsymbol{C}_{i,}(k)\mathbf{w}}{\boldsymbol{\lambda}^\top \boldsymbol{C}(k) \mathbf{w}},
>   \end{align*}
>    that is, the weight corresponding to each jump in the independent setting is replaced by the effective weight in which all inputs are taken into account. It holds
>
>   \begin{equation*}
>     \mathbf{p}(k+1) = \frac{\boldsymbol{\lambda} \odot (\mathbf{C}(k+1) \mathbf{w}(k+1))}{\boldsymbol{\lambda}^\top \mathbf{C}(k+1)\mathbf{w}(k+1)} = \frac{\boldsymbol{\lambda} \odot (\mathbf{C}(k+1) (\mathbf{w}(k)\odot(\mathbf{1}+\alpha\mathbf{Y}(k))))}{\boldsymbol{\lambda}^\top \mathbf{C}(k+1)(\mathbf{w}(k)\odot(\mathbf{1}+\alpha\mathbf{Y}(k))}.
>   \end{equation*}
>   Since $\mathbb{E}[\mathbf{Y}(k)]=\boldsymbol{\rho}\mathbf{p}(k), \mathbb{E}[\mathbf{C}(k)]=\boldsymbol{\rho}$, we obtain the gradient flow
>
>   \begin{equation*}
>     \frac{\mathbf{d}}{\mathbf{d} t}\mathbf{p}(t) = \mathbf{p}(t)\odot (\boldsymbol{\rho}\mathbf{p}(t) - \mathbf{p}(t)^\top \boldsymbol{\rho}\mathbf{p}(t)\mathbf{1}),
>   \end{equation*}
>   which again is a replicator equation, and the Shahshahani-loss is $\mathbf{x}\mapsto \frac{1}{2}\mathbf{x}^\top \boldsymbol{\rho}\mathbf{x}$. A rigorous mathematical analysis of this correlated model follows the steps in the independent case.
>
> - **2.2 Extension to non-balanced STDP:**
>
>     Our approach can be adjusted to non-balanced STDP in the following way. Let $W(x)$, $x\in\mathbb{R}$, be the learning window let $\Delta\tau$ be a generic difference between a pre- and a postsynaptic spike. Define the ensemble average $\bar{W}=\mathbb{E}[W(\tau)]$, for example $\bar{W}=\int W(x)d x$ for uniformly distributed jump times within the support of $W$. We adjust the original learning rule (Eq. 1) such that
>     \begin{equation*}
>         \mathbf{w}(k+1) = \mathbf{w}(k)\odot \left(1 + \alpha(\mathbf{B}(k) + \mathbf{Z}(k) - \bar{W}\mathbf{1})\right).
>     \end{equation*}
>     Since $\mathbb{E}[\mathbf{Z}(k)]=\bar{W}\mathbf{1}$, we have for $\tilde{\mathbf{Y}}=\mathbf{B}+\mathbf{Z}-\bar{W}\mathbf{1}$ that $\tilde{\mathbf{Y}}$ is bounded and that $\mathbb{E}[\tilde{\mathbf{Y}}\vert\mathbf{p}(k)]=\mathbf{p}(k)$. These two properties allow us to extend the convergence guarantees as in Theorem 2.1 to non-balanced STDP. The subtle interplay between the integral of the learning window and the linear term $\bar{W}$ has already been observed by Kempter et al 2001.
>
> - **2.3 Beyond pair based STDP:** In the revised version, we will highlight the limitations of pair-based STDP rules in the discussion section and cite Sjostrom et al. (2001). We will also briefly discuss extension to triplet and voltage-dependent rules.
>
> **Questions:** The questions are linked to the previous comments.
>
> - **Connection to Prior Work:** While we tried to provide a summary of the previous work, we sometimes only mentioned articles such as the related Kempter et al. paper without a more detailed comparison. We do acknowledge that this needs to be improved in the revised version.
>
>     The comment refers to the lines of work mentioned in 1.1-1.3 above. Please see the specific answers to those comments.
>
> - **Narrow scope of input models:** Please see the response above on 2.1 generalization to non-homogeneous Poisson processes and correlated inputs.
>
> - **Simplifying assumptions in STDP rule/ Extension to non-balanced STDP:** Please see our response to 2.2.
>
> - **Biological plausibility/ Beyond pair-based STDP:** Please see our response to 2.3.
>
> Kempter et. al (2001) Neural Comput
>
> Scellier, Bengio (2017) Front Comput Neurosci

---

> > ### Comment · Reviewer_2zhK · 2025-08-04
> >
> > I thank the authors for their thorough reply. I have no further questions.
> > With the extra work they did on the correlated input and time-dependent input, I go to 4.

---

> > > ### Author Response · Authors · 2025-08-05
> > >
> > > Thank you — we are glad that we were able to provide satisfactory responses to your remarks.

---

### Official Review · Reviewer_vBKN · 2025-07-01

**Clarity:** 2
**Significance:** 3
**Originality:** 3
**Rating:** 5
**Confidence:** 3

**Summary:**

In this work, the authors establish a mathematical connection between spike-timing-dependent plasticity (STDP) and noisy gradient descent on a probability simplex. The nature of this connection allows for mathematical analysis which goes beyond previous techniques which utilized ensemble averages. The main finding of their analysis is that their STDP learning rule converges exponentially fast to identify the presynaptic neuron with the highest firing rate. Additionally, they establish a connection to noisy mirror descent, specifically entropic gradient descent.

**Questions:**

Regarding the selection probabilities (section beginning line 238), I would assume that, under most biological conditions, a handful of presynaptic spikes (on the order of tens) would be sufficient to trigger a postsynaptic spike. Is there any perturbation or corrective term which can be added to move even slightly away from the two limit cases (“weights are very small” or “weights are large enough to for any presynaptic activity to trigger a postsynaptic spike”)?

I lost the plot a bit around theorem 2.2. I was able to answer the questions I had by going to the appendix, but it broke up the flow of the paper a bit for me. Perhaps the authors can consider adding more “connective tissue” in the main text to guide the reader, some bits are currently limited to an expert audience.

I would be curious to see a more direct comparison to the previous methods of STDP analysis which used ensemble averages.

**Ethical Concerns:**

["NO or VERY MINOR ethics concerns only"]

**Final Justification:**

This paper provides a new take on STDP with strong theoretical backing.

**Limitations:**

Yes

**Paper Formatting Concerns:**

No issues.

**Quality:**

3

**Strengths And Weaknesses:**

This work provides a novel analysis of STDP, one that is certainly significant to the fields of computational neuroscience and AI. The mathematical formulation linking local spike-based updates to global optimization is valuable. The  mirror descent perspective is of particular interest, connecting STDP to an intriguing framework from Cornford et al. 2024. The assumptions and proofs are well described, though it may benefit the authors to target a slightly wider audience. While the analysis is novel and interesting, it does require some assumptions which limit its applicability to STDP in biological conditions (the authors acknowledge these limitations).

---

> ### Author Rebuttal · Authors · 2025-07-30
>
> Thank you for your positive assessment of our work. We fully agree that for NeurIPS the manuscript should be readable by a wide audience.
>
>
> - **"it may benefit the authors to target a slightly wider audience":**
>
>     As mentioned also in the response to Reviewer jhz1,  our aim is to contribute an article that is appealing to the whole NeurIPS community and in particular also researchers working on theoretical guarantees for SGD.
>
>     Based on all the received feedback, we will be able to add some discussion that is beneficial to a general readership. If you have specific suggestions, we would of course be very happy to learn about them.
>
> - **"Is there any perturbation or corrective term which can be added to move even slightly away from the two limit cases":**
>
>     In general it is hard to find an exact representation of the probabilities beyond the cases already stated in the paper, see Novikov et al. 2005 for a broader discussion.
>
>     However, the formula for the probabilities can also be motivated by the following heuristic. The postsynaptic membrane potential $Y$ grows, in expectation, at an approximately linear rate $\mathbf{\boldsymbol{\lambda}^\top \mathbf{w}}$. Furthermore, input $i$ causes a postsynaptic spike if, and only if, it spikes at a time at which $Y\geq S-w_i,$ where $S>0$ is the threshold level. As $Y$'s growth is approximately linear with rate $\mathbf{\boldsymbol{\lambda}^\top \mathbf{w}}$, the amount of time in which $Y\geq S-w_i,$ holds is approximately $w_i/\mathbf{\boldsymbol{\lambda}^\top \mathbf{w}}$. Now the probability that input $i$ jumps in an interval of length $w_i/\mathbf{\boldsymbol{\lambda}^\top \mathbf{w}}$ is given by $1-\exp(-\lambda_iw_i/\mathbf{\boldsymbol{\lambda}^\top \mathbf{w}})\approx \lambda_iw_i/\mathbf{\boldsymbol{\lambda}^\top \mathbf{w}},$ which is our modelling choice in the independent setting.
>
>     We will add this heuristic to the revised manuscript.
>
> - **Adding more material around Theorem 2.2:** Of course, we could add more discussion here in the revised version. For us it would be very valuable to learn what specifically you missed in the text.
>
> * **A more direct comparison to the previous methods of STDP analysis which used ensemble averages:** This is closely related to the second comment of the first referee and one comment of the second referee. We reproduce the answer below:
>
>     It is known that rate-based learning rules will do some form of principal component analysis. While we take the stochasticity into account, rate-based models only capture the mean behaviour. This makes, however, a huge difference and can lead to different behaviour. An example is a zero-mean stochastic process: The rate-based behaviour is just the zero function (the mean of the process), whereas the sample path can be very wild and also diverge to infinity.
>
>     For instance, the rightmost plot in Figure 2 shows that due to randomness, the dynamic can end up in a different local minimum. Rate-based models are obtained already by taking a limit $t\to \infty.$ This means that they operate effectively on a slower time scale. Thus, the exponential fast decay of the STDP rule suggests a much faster convergence than an exponential decay in a rate-based model.
>
>     An exception to the analysis of rate-based models is Kempter et al. '99 which also includes noise but rewrites the dynamic into a deterministic ODE. The analysis and the results are both different. For more details on that, please see the first point in the reply to 2zhK.
>
>     We will add the argument above to the revised version.
>
>
>
>
> Novikov et al. (2005) Prob Eng Mech doi:10.1016/j.probengmech.2004.04.005

---

> > ### Comment · Reviewer_vBKN · 2025-08-04
> > **reply**
> >
> > Thank you we will keep our score.
> >
> > Regarding theorem 2.2, it would be helpful to a) repeat the definition of some parameters (i.e. Q), which were not referenced in 2.1. It would also be helpful to provide a description in words before each step, e.g. 'we define the difference between the largest and second largest initial probabilities as \Delta' and assume an upper bound on the learning rate alpha which scales with (blank). ' It would further help to connect these to visuals. For example, you might choose an initial condition from Figure 2a and determine what this upper bound for alpha would be, and how that compares to your chosen rate of alpha = 0.01.  I think that directly communicating to the reader a) the scaling and b) reasonable values for each equation helps build intuition without needing to see the derivation. Additionally, the ultimate point of  2.2 is to demonstrate that  'the theorem shows linear convergence of the first component to 1 on an event Θ in expectation.'  Point out explicitly (in words) what part of the theorem this convergence hinges on, and under what alternative assumptions it might fail. All this might seem like overkill, but can greatly improve readability and accessibility to a broader audience.

---

> > > ### Author Response · Authors · 2025-08-05
> > > **Thank you**
> > >
> > > Dear Referee,
> > > Thank you very much for your time and effort in reviewing our submission. We sincerely appreciate your valuable comments and recommendations, and we will carefully incorporate them into the revised version.

---

### Official Review · Reviewer_jhz1 · 2025-07-02

**Clarity:** 2
**Significance:** 3
**Originality:** 3
**Rating:** 4
**Confidence:** 3

**Summary:**

The paper studies spike-timing dependent plasticity (STDP) in a feedforward network of d input neurons projecting on a single postsynaptic neuron. Rather than analyzing  the classical STDP model (Gerstner 1996), which the authors argue is complex to analyze, the authors consider a simplified version of it, which is described in section 2.3. The authors make a thorough analysis of this simplified model, showing convergence by casting its dynamics as noisy gradient descent of a specific loss function. Notably, this simplified model behaves as a winner-take-all selection mechanism, with the postsynaptic neuron aligning with the highest firing input neuron. This is a departure from the rate case (e.g. Oja’s rule) in which the post-synaptic neuron aligns with the first principal component of the input layer.

**Questions:**

1/ Could the authors explain how their simplified model incorporates the integration behavior of standard STDP? E.g. how are the postsynaptic spike times dependent on the threshold S?

2/ A classic result for rate models, that the authors point to, is that rate models under with Hebbian plasticity learn to extract the principal components of the input data. Often there is a somewhat natural mapping between spike and rate models, specifically STDP and Hebbian learning, that would bridge the existing PC results to the authors’ results regarding the alignment of the output neuron to the highest firing input neuron direction. This is interesting. Do the authors have any suggestions about why that does not appear to be the case here?

**Ethical Concerns:**

["NO or VERY MINOR ethics concerns only"]

**Final Justification:**

The authors have clarified several points that made this work initially hard for me to appreciate, such as the connection to existing work and novelty. As a result, I have increased my score to 4.

**Limitations:**

Yes

**Quality:**

3

**Strengths And Weaknesses:**

Strengths:

- The math and presentation of the derivations are very rigorous and well-explained.
- Convergence of this model is rigorously established by casting it as noisy gradient descent.

Broadly, the paper was a valiant effort at using sophisticated mathematical tools to analyse a system of neuroscientific interest. It performs a thorough theoretical analysis of a simplified model that is appreciated.

Weaknesses:

However, the value of the contribution is hindered slightly by the clarity of the presentation for neuroscientist, but majorly by the large leap required between STDP and the stated model.

- Relevance of the simplified model to neuroscience: The jump from the classical STDP type model to the simplified model is worrying. A crucial part of the integrate-and-fire model is the fact that it integrates inputs in order to choose when to spike. On first reading, this appears to be completely missing from the simplified model. Further, the classic components of STDP, such as spike-time dependent kernels are also missing. These details all seem to be wrapped up in the procedure for selecting the presynaptic spike at which the postsynaptic neuron fires. However, this seemingly crucial selection procedure is somewhat swept under the carpet (line 236). The best version of this claim would be that you can choose this selection procedure in order to model a classical integrate-and-fire neuron with STDP. Yet no evidence is presented that this is possible. Providing such evidence would significantly strengthen the article. More broadly, more discussion regarding these differences between the toy and classical model is strongly recommended.

- High level presentation: The paper seems formatted backwards in some ways. For example, the easier to understand analysis, on gradient flow, arrives after the complex discrete time analysis. Further, the movement from the classical integrate-and-fire model to the more toy model was presented after a long discussion of the toy model. For a neuroscientist the natural place to start would be the classical model, then explain the simplifications, before ending with the resulting analysis of the model.

More minor points:

- Some figures are not informative: e.g. Fig 1: shows a feedforward network of spiking neurons, which is very common in the field, but gives no insights on the plasticity rule modeling. A cartoon would greatly help to ease the reader into the authors’ way of considering STDP.

- The links to mirror descent could be interesting, but it is unclear how it is used to create actionable insights. It feels like it is another mathematical framework presented regarding dynamics that have the same equilibria, and the neuroscientifically interesting conclusion is the structure of these equilibria. What then do we gain via an alternative characterization of the dynamics? It would be good to signpost the role it is hoped that this additional analysis plays.

- Typo in the appendix: the loss on line 769 is missing a square.

---

> ### Author Rebuttal · Authors · 2025-07-30
>
> Thank you very much for your assessment of our manuscript. We do think that despite the idealized model, this submission is of sufficient significance and we believe that we could address all your concerns. Feel free to provide us with additional feedback in case something is unclear or needs additional clarification. Thank you again!
>
> - **How does the model incorporate the integration behavior of standard STDP? E.g. how are the postsynaptic spike times dependent on the threshold S?**
>
>     We do argue that the postsynaptic spike times that we consider follow a similar, and in some cases even the same distribution, as the spike times determined by the threshold model. In particular, they depend on the threshold $S.$ Below, we sketch this argument. To clarify this point, we will also add it to the revised article.
>
>     The common model is to assume that the postsynaptic neuron fires at the moment when the membrane potential exceeds the threshold $S.$ According to this rule, the postsynaptic spikes can only happen at moments when one of the presynaptic neurons spike. For convenience, let us assume that the previous postsynaptic spike happened at time $0.$ Denote by $\tau_{j1},\tau_{j2},\ldots$ the spike times of the $j$-th presynaptic neuron after time $0$ in increasing order.
>
>     As the next postsynaptic spike $\mathfrak{t}$ can only happen at a presynaptic spike time, the distribution of the postsynaptic spikes is completely determined by the probabilities
>
>     $$P\big(\mathfrak{t}=\tau_{j\ell}\big), \quad j=1,\ldots,d, \quad \ell=1,2,\ldots$$
>     These probabilities do depend on the threshold $S.$
>
>     In the derivation of our learning rule, we select the postsynaptic spike times to match this distribution as closely as possible. For that, we rewrite the probability into
>
>     $$P\big(\mathfrak{t}=\tau_{j\ell}\big)
>         = P\big(\mathfrak{t}=\tau_{j\ell} \big|\mathfrak{t}\in (\tau_{jk})\_{k\geq 1} \big)
>         P\big(\mathfrak{t}\in (\tau_{jk})\_{k\geq 1}\big) $$
>
>     This means, in a first step, we need to select the probability $P(\mathfrak{t}\in (\tau_{jk})\_{k\geq 1})$ that the $j$-th presynaptic spike train causes the next postsynaptic spike. In Section A.3, we show that the probabilities $\widetilde P(\mathfrak{t}\in (\tau_{jk})_{k\geq 1})$ that we select nearly match the probabilities of the original dynamic. In particular, Lemma A.5 describes a setting, where this is even exact.
>
>     Our argument does not require a specific choice of the second probability $P(\mathfrak{t}=\tau_{j\ell} |\mathfrak{t}\in (\tau_{jk})_{k\geq 1}).$ This means, here we can take the one from the original dynamic. This probability will also incorporate most of the dependence on $S.$
>
>     That the considered model is of integrate-and-fire type has also been acknowledged by Referee 2zhK.
>
> - **Spike-time dependent kernels are missing:** Indeed, we work with a kernel that is a jump followed by an exponential decay (see also the report of Reviewer 2zhK). This is a common choice. The derivation of the learning rule (3) does not exploit the specific structure of the kernel. If you have a specific class of kernels in mind, we would be happy to include it in the revision.
>
> - **"The paper seems formatted backwards in some ways... For a neuroscientist the natural place to start would be the classical model, then explain the simplifications, before ending with the resulting analysis of the model":**  While the paper has been mainly evaluated from the neuroscience perspective, our aim is to contribute an article that is appealing to the whole NeurIPs community and in particular also researchers working on theoretical guarantees for SGD. Even in its current form, Referee vBKN suggests "to target a slightly wider audience". We do acknowledge that for neuroscience, a different structuring of the material might be more natural. To make the paper more accessible to the SGD community motivated the way we have presented the results, stating the idealized learning rule right at the beginning and deferring the rather technical derivation from standard STDP rules to a later chapter.
>
>     If our reasoning on this point is unclear or unconvincing, we are very open to suggestions. Should you and the other referees feel that a different structure would be more appropriate, we would of course be happy to make the necessary changes.
>
>
> - **Mapping between spike and rate models:**
>
>     It is known that rate-based learning rules will do some form of principal component analysis. While we take the stochasticity into account, rate-based models only capture the mean behaviour. This makes, however, a huge difference and can lead to different behaviour. An example is a zero-mean stochastic process: The rate-based behaviour is just the zero function (the mean of the process), whereas a sample path can be very wild and also diverge to infinity.
>
>     For instance, the rightmost plot in Figure 2 shows that due to randomness, the dynamic can end up in a different local minimum.
>
>
>     Rate-based models are obtained already by taking a limit $t\to \infty.$ This means that they operate effectively on a slower time scale. Thus, the exponential fast decay of the STDP rule suggests a much faster convergence than an exponential decay in a rate-based model.
>
>     An exception to the analysis of rate-based models is Kempter et al. '99 which also includes noise but rewrites the dynamic into a deterministic ODE. The analysis and the results are both different. For more details on that, please see the first point in the reply to 2zhK.
>
>     We will add the argument above to the revised version.
>
> - **More minor points:** In the revised version, we will add some information on the learning dynamic to Figure 1. Regarding the connection to mirror descent, please see the reply to Referee 9Yjy. Thank you for spotting the typo in line 769.

---

> > ### Comment · Reviewer_jhz1 · 2025-08-06
> >
> > I thank the authors for their thorough response which has brought many welcome clarifications, I will increase my score.

---

### Official Review · Reviewer_9Yjy · 2025-07-03

**Clarity:** 3
**Significance:** 3
**Originality:** 3
**Rating:** 4
**Confidence:** 3

**Summary:**

The authors provide a new theoretical analysis of the convergence behavior of Hebbian STDP rules. First, they show that a Hebbian STDP rule can be reformulated as gradient descent on a loss function defined over the presynaptic neurons spike-triggering probability simplex. Then, they prove that such spike-triggering probability converges to the basis vector of the probability simplex pointing to the presynaptic neuron with highest initial spike-triggering probability. Lastly, they discuss connections to noisy mirror descent.

**Questions:**

Currently, the argument in Section 5 of the paper is not rigorous enough. Can you prove convergence of p(k) for an arbitrary number of readout neurons?

Can you provide an intuitive interpretation for the loss function in Eq. 5?

Which is the practical significance of Theorem 2.2 for training SNNs with STDP rules?

Which known STDP rules fall under the formulation of Eq. (1), and which do not?

**Ethical Concerns:**

["NO or VERY MINOR ethics concerns only"]

**Final Justification:**

I appreciated the authors detailed response. The additional emphasis on the novel contributions, along with the new analytical results on convergence with multiple output neurons, improved my understanding and appreciation of the work. As a result, I have updated my score to 4.

**Limitations:**

Currently, it is not clear which STDP rules belong to the Eq. (1) family. The authors should specify that Theorem 2.2 is limited to STDP rules belonging to the family in Eq. (1).

**Paper Formatting Concerns:**

I have no paper formatting concerns.

**Quality:**

3

**Strengths And Weaknesses:**

Strength: The mathematical derivations are theoretically sound. The rigorous convergence analysis of p(k) in STDP is a potentially important theoretical result. Viewing STDP as gradient descent on the presynaptic neurons' spike-triggering probability simplex is, to the best of my knowledge, a novel and compelling idea that may inspire further theoretical developments.

Limitations:
Despite its mathematical rigor, the manuscript lacks clarity in key areas, and in its current form, the novelty does not appear sufficient to meet the standards expected at NeurIPS.
First, it is unclear whether the authors introduce a new Hebbian STDP rule or analyze a family of existing ones. Lines 12–13 state they “provide an analysis of a STDP rule,” while line 83 says they “introduce a STDP rule,” suggesting novelty without clarification. Yet, Section 2.3 presents a biologically plausible STDP rule that has already been studied in the literature (e.g., Ref. 11), and that can be reformulated to match Eq. (1). This suggests that Eq. (1) generalizes a known class of STDP rules, with Eq. (11) as a specific instantiation. However, the manuscript does not clearly state which rules fall under this general form.
The paper's structure further obscures this point: Eq. (1) is introduced early, without sufficient context, and its connection to established STDP formulations only emerges later, in Section 2.3. This non-linear narrative makes it difficult to follow the paper’s logic.
The authors should more precisely specify the generality of Eq. (1) in relation to known STDP variants that go beyond the brief description in lines 27–28. Experimental studies support the existence of multiple types of STDP rules, varying across brain regions and cell types (Fino et al. J Physiol 2008, doi: 10.1113/jphysiol.2007.144501; Perez et al. Cell Rep 2022, doi: 10.1016/j.celrep.2022.110521). While this diversity is not yet fully leveraged in SNNs, the authors should acknowledge it and specify the generality of Eq. (1).
It is also unclear what results could be derived using the previous methods mentioned in lines 35–37, and which results uniquely stem from the gradient-descent interpretation introduced here.
While the derivations appear theoretically sound, their novelty and significance remain uncertain. As the authors also point out, convergence to a winner-take-all regime has been previously observed for Hebbian STDP models. The authors do not sufficiently discuss the added value of their convergence result, which appears to be the main theoretical contribution of the paper. The connection to noisy mirror descent is also only briefly mentioned and not clearly motivated or explored.
Overall, proving the convergence of a specific family of STDP rules to an already known regime – within the simplified setting of a single-readout neuron - constitutes, in my view, a limited theoretical contribution that does not meet the standard expected at NeurIPS. Rigorously extending their theoretical result to the scenario of an arbitrary number of readout neurons would significantly strengthen the contribution.
Minor comments:
•	Eq. (1): The quantity Z(k) should be defined immediately after the equation. Also, explicitly stating that k indexes postsynaptic spikes would aid clarity.
•	The writing style should be improved throughout. For instance, the sentence “We continue by relating the learning rule (3)” (line 101) appears before Eq. (3) is introduced and is difficult to follow.
•	To avoid confusion, I recommend consistently prefixing equation references with “Eq.” to distinguish them from citation numbers.

---

> ### Author Rebuttal · Authors · 2025-07-30
>
> Thank you for your careful reading of the manuscript and your suggestions. The feedback did help us to improve the article considerably.
>
> - **On the class of considered STDP rules:** We excuse for any ambiguity that might have occurred during reading. As mentioned by you, we start with a widely considered pair-based STDP rule that is stated in the standard reference [11]. We now make one modification. In the original STDP rule, the postsynaptic spike timings are determined by the moments when the postsynaptic membrane potential exceeds the threshold. According to this rule, the postsynaptic spikes can only happen at moments when one of the presynaptic neurons spike. While the formula for the postsynaptic spike timings is mathematically easy to formulate, a direct analysis seems extremely challenging and has to be replaced by a surrogate. To mimic the distribution of the original spike timings, we instead consider a two-stage rule that first selects a presynaptic spike train and then, in a second step, chooses the next postsynaptic spike time among the presynaptic spike times of the selected presynaptic spike train. We put considerable effort to show that the distributions of both rules are close, see Section A.3 for details. In particular, in the setting of Lemma A.5 they are even the same. For more details, please see the first response to Reviewer jhz1. This leads then to the learning rule (3). In the revised version, we will make this point more clear.
>
>     We want to emphasise that for a rigorous mathematical treatment of the learning rules, one needs to consider a tractable model.
>
>     Thank you for pointing us to (Fino et al. 2008 and Perez et al. 2022). In the revised version, we will emphasise the fact that there is a large variety of STDP rules and also cite these articles.
>
> - **"It is also unclear what results could be derived using the previous methods mentioned in lines 35–37":**
>
>     This is closely related to the comment of Referee jhz1 on "Mapping between spike and rate models". We refer to this comment for a detailed response.
>
>
> -  **Novelty and significance** ("Convergence of a specific family of STDP rules to an already known regime ... constitutes ... a limited theoretical contribution"):
>
>     We feel that the original article did not put sufficient emphasis on the main findings and the key contributions of this submission.
>
>
>     * **Surprising convergence behaviour:**
>     Despite the constant injection of noise, the considered Hebbian learning dynamic converges to a point. This is non-standard and surprising as typically noisy gradient descent with fixed noise level only converges to a stationary regime where the noise causes the dynamic to fluctuate around a minimiser. This point will be stressed more in the revision.
>
>         Based on the analysis, we can show an exponentially fast convergence to the limit and quantify the influence of individual parameters such as the number of input neurons and the learning rate. The exponentially fast convergence is surprising as noisy stochastic gradient descent typically only converges with at most a polynomial rate, see e.g.  Mertikopoulos et al. (reference [23]) [for gradient descent exponentially fast convergence holds in some regimes].
>
>     * **Mathematical analysis of non-convex noisy SGD:** For convex loss functions, (noisy) SGD  is well-understood. The hardness of the considered learning rule is the underlying non-convex landscape with several stationary points. Analysing noisy SGD in non-convex settings is a notoriously hard problem. Our analysis is inspired by the recent work by Mertikopoulos et al.\ (reference [23]) that previously appeared in NeurIPS. (In the past years several papers on theory for SGD in non-convex settings appeared in NeurIPS, including Patel et al. "Global convergence and stability of stochastic gradient descent" and Hu et al. "Efficiency ordering of stochastic gradient descent.")
>
>      - **Connecting recent theory of SGD with Hebbian learning rules:** To the best of our knowledge, this is the first article that provides a rigorous mathematical convergence analysis of a stochastic Hebbian learning rule. (For a comparison with Kempter et al. '99, see the response to Reviewer 2zhK). Cross-fertilisation between the theoretical learning community and practitioners has been proven to be extremely useful for the development of deep learning (highlights include the neural tangent kernel (NTK) that allows to understand gradient descent (GD) for wide networks, the fact that GD converges to weight balanced solutions, and the benign overfitting phenomenon that rigorously proves why GD learned fits can still generalise well in highly overparametrised models]). We envision a similar development  can likely generate new insights into Hebbian learning.
>
>     Same as for SGD analysis for artificial neural networks, a rigorous theory requires to work in idealised models. In deep learning theory, the theoretical findings still align with empirical findings.
>
>     To extend the setting, we will also discuss several extensions to time-inhomogeneous intensities and correlated input neurons in the revised version. For details, please see the response "2.1 Generalisation to non-homogeneous Poisson process" (Reviewer 2zhK).
>
> - **Connection to noisy mirror descent:** We do believe that the Riemannian gradient flow, as derived in Section 3, is the more natural setting, although it leads to similar results. The loss function becomes, for instance, much simpler, see also the response to the next comment.
>
>     Stochastic mirror descent is well-explored in the optimisation literature, both from its theoretical foundations and its implementation. The connection laid out in Chapter 4 enables the transfer of these insights to STDP. Potential implications for STDP and training SNNs include optimal choice of the hyperparameters (e.g., learning rate), but it is also conceivable that new variants of STDP can be introduced based on stochastic mirror descent insights.
>
>     That the choice of the metric matters, has also been observed in the neuroscience literature, see e.g. Surace et al. (2020). (This point has been  also highlighted by Reviewer 2zhK).
>
> - **Intuitive interpretation for the loss function in Eq. 5:** We do not have a full answer to this question. However, here the connection to the natural gradient descent in Chapter 3 helps. Indeed in the natural geometry, the loss is the negative of a least squares loss and thus an energy term.
>
>
> - **Extension to several readout neurons:**
>
>     We can provide convergence guarantees for a slightly modified learning algorithm for multiple output neurons, where the output neurons are trained in consecutive episodes, each with $K$ training iterations. The first output neuron is trained as in Eq. (1), such that the convergence guarantee of Theorem 2.1 holds. After $K$ training steps, the weight $\mathbf{w}_1$ is projected in the direction of the closest unit vector, measured in cosine similarity, and the learning is stopped. This yields the weight vector $\mathbf{w}_1^\ast =\Vert\mathbf{w}_1^\ast\Vert\mathbf{e}_1$ with high probability. In the second episode, the second output neuron is trained. In the first step, we remove the component of the initial weight vector $\mathbf{w}_2(0)$ into the direction of $\mathbf{w}_1^\ast$  ($\mathbf{w}_2(0)\gets \mathbf{w}_2(0)-\mathbf{w}_1^\ast\mathbf{w}_2(0)^T\mathbf{w}_1^\ast/\Vert\mathbf{w}_1^\ast\Vert$), effectively setting its first component to zero. The training proceeds as in Eq. (1), with now the second component dominating.  Again Theorem 2.1 applies. Proceeding iteratively for $d$ episodes, we obtain the weight matrix $\mathbf{W}^\ast$ and the corresponding probability matrix $\mathbf{P}^\ast$ with the following mathematical guarantee:
>
>     **New result:** For positive minimal gap at initialisation $\Delta =\min_{i=1,\dots, d-1}([\mathbf{p}\_i(0)]\_i - \max_{j> i}[\mathbf{p}_i(0)]\_j)>0,$
>
>   \begin{equation*}
>     \mathbb{P}(\mathbf{P}^\ast \ \text{is identity})\to 1
>   \end{equation*}
>   as $K\to\infty$.
>   More precisely, for $\kappa := \min_{i=1,\dots, d}\lambda_i/\max_{i=1,\dots, d}\lambda_i$ and $\epsilon>0$,
>
>   \begin{equation*}
>         \mathbb{P}(\mathbf{P}^\ast \ \text{is identity})\ge (1-\epsilon)^d \quad\text{for all }K> \frac{16d}{\alpha\Delta(4+d\Delta)}\log\Big(\frac{4(1+\kappa)}{\epsilon\kappa} \Big).
>   \end{equation*}
>
>
> - **Practical significance of Theorem 2.2 for training SNNs with STDP rules:**
>
>     Theorem 2.2 is the main result of this manuscript, establishing exponentially fast convergence ($=$linear  convergence in the optimisation literature) for the learning rule (3).
>
>     As mentioned above, there are two surprising findings, namely the fact that despite the constant injection of noise, the dynamic converges to a deterministic limit and secondly, the exponentially fast decay that is entirely unexpected for noisy SGD schemes.
>
>     The practical significance is that one does not have to worry about the noise and that one can expect an extremely fast convergence. The result also allows to pick a constant learning rate, quantifies the range of possible learning rates, and helps to understand the influence of other parameters in the model.
>
> - **Which known STDP rules fall under the formulation of Eq. (1), and which do not?**
>
>     The considered learning rules are based on the pair-based STDP models for excitatory neurons as discussed in [11], Section 19.2.2 with amplitude parameters $A_+(w)=A_-(w)=w.$ As detailed in the first response to Reviewer jhz1, we closely mimic the distribution of the original postsynaptic firing times.
>
>     We will make this very clear in the revised version.
>
> - **Minor comments:** They will be addressed in the revised article. Thank you for your suggestions.

---

> > ### Comment · Reviewer_9Yjy · 2025-08-04
> >
> > I thank the authors for their efforts in addressing my questions and for the clarifications provided. The additional emphasis on the novel contributions, along with the new analytical results on convergence with multiple output neurons, improved my understanding and appreciation of the work. As a result, I have updated my score to 4.

---

> > > ### Author Response · Authors · 2025-08-05
> > >
> > > Thank you for your thoughtful feedback and for acknowledging our clarifications and additions.

---

### Decision · Program_Chairs · 2025-09-17

**Decision:**

Accept (poster)

**Comment:**

This paper gives a rigorous, optimization-theoretic account of a class of spike-timing–dependent Hebbian rules by recasting them as noisy gradient descent on an associated loss. Within the setting of feedforward network with a single output neuron, the authors prove convergence of the spike-triggering probability to a winner-take-all solution that selects the most active presynaptic neuron, with linear rate despite persistent noise; they also elucidate a connection to noisy mirror descent. Reviewers found the derivations careful and the result conceptually clarifying, contrasting it with rate-based analyses that typically yield PCA-like behavior. The work’s principal strengths are (i) a clean formulation that links local STDP updates to global optimization geometry; (ii) a rigorous convergence guarantee for a stochastic, non-convex learning dynamic; and (iii) a bridge between computational neuroscience and modern optimization language that could seed follow-ups in SNN training.

The main weaknesses raised were scope and positioning: reliance on an idealized pair-based STDP and simplified spike-selection mechanism; initially narrow input assumptions (independent, homogeneous Poisson); ambiguity about what existing STDP variants are covered; limited discussion of prior STDP-as-optimization literature; and presentation/ordering that made accessibility difficult for non-specialists. During rebuttal, the authors substantially strengthened the case: they clarified the mapping from classical integrate-and-fire STDP to their two-stage selection rule and its dependence on threshold, specified the family of pair-based rules encompassed, and extended the analysis to multiple readout neurons via episodic training, to time-inhomogeneous and correlated inputs, and to non-balanced windows; they also broadened related-work coverage and committed to clearer exposition. These responses led all reviewers to settle at borderline-accept or accept. I recommend acceptance.